# General Analysis of LMO-based Optimizers: Beyond Bounded Variance

Egor Shulgin [1 2 †]  Mohamed Awad [2]  Peter Richtárik [1]  Eduard Gorbunov [2]

## Abstract

We study a broad family of momentum *Linear Minimization Oracle* (LMO) methods that includes normalized SGD with momentum, sign-based (Adam-like) directions, and Muon (spectral) updates. Our focus is on subsampling regimes where the classical uniformly-bounded-variance model can be fragile even for finite-sum objectives on unbounded domains. To obtain subsampling-faithful guarantees, we analyze this LMO family under *expected smoothness* (ABC condition), which captures common sampling schemes. We establish a unified nonconvex convergence theory via a new self-bounding closure that handles the history-coupling induced by momentum under ABC. Our bounds recover known bounded-variance results as a special case and simplify in strong-growth regimes. Specializing to $\tau$-nice sampling, we derive explicit batch-size scaling laws, predicting that the optimal momentum must increase with the batch size to maximize sample efficiency. We further identify a theoretical 'optimal batch size' that minimizes total sample complexity. Experiments on linear and matrix regression corroborate these predictions, showing a distinct diagonal shift in the optimal momentum-batch landscape that matches our theoretical scaling.

## 1. Introduction

Mini-batch stochastic gradient descent (SGD) remains the workhorse of large-scale optimization in machine learning. Its behavior is now well understood under classical stochastic-gradient noise models such as uniformly bounded variance (Ghadimi & Lan, 2013). However, modern training pipelines are fundamentally *subsampling-driven* (finite-sum objectives + mini-batches), and in this regime bounded-variance-type modeling can be misleading and can obscure batch-size effects.

**From Bounded Variance to Expected Smoothness.** A key issue is that bounded variance (and similarly uniformly bounded $p$-th moments) can fail even for simple finite-sum problems on unbounded domains, e.g., subsampled quadratic objectives. This motivates analyses under *expected smoothness* models. In particular, Khaled & Richtárik (2023) proposed the *ABC condition*, which controls the second moment of the stochastic gradient via a combination of function suboptimality, gradient norm, and a residual term. Beyond subsampling, closely related "expected smoothness" frameworks have also been adapted to capture broader effects such as distributed/communication settings with compression.

**Why This Matters for Modern Optimizers.** Since the rise of deep learning, practical training has increasingly relied on *normalized* (You et al., 2017; 2020), *sign-based* (e.g. Adam (Kingma & Ba, 2015; Orvieto & Gower, 2025)), and more generally *LMO-based* (steepest-descent-in-a-norm) updates (Pethick et al., 2025a), including Muon-style methods (Jordan et al., 2024). Existing nonconvex guarantees for this family are largely derived under bounded-variance or bounded-moment assumptions, which are precisely the abstractions that can break under subsampling and that suppress nontrivial batch-size phenomena. This leaves a gap between the empirically observed batch-size sensitivity of SGD/Adam/Muon-type optimizers (Marek et al., 2025) and existing theory.

To bridge the gap between theoretical assumptions and practical behavior, we develop a unified convergence theory for the broad momentum *LMO-family* under the general ABC condition. Unlike prior analyses restricted by bounded variance, our framework explicitly models the effects of subsampling in finite-sum regimes. This approach allows us to overcome the technical challenges of history-coupled noise and derive rigorous scaling laws that link batch size, momentum, and sample complexity. Our main contributions are as follows:

---

[†]Most of this work was done when Egor Shulgin was a visiting student in the group of Prof. Eduard Gorbunov at MBZUAI, UAE. [1]King Abdullah University of Science and Technology (KAUST), Thuwal, Saudi Arabia [2]Mohammed Bin Zayed University of Artificial Intelligence, Abu Dhabi, United Arab Emirates. Correspondence to: Egor Shulgin <shulgin.yegor@gmail.com>.

*Proceedings of the 43rd International Conference on Machine Learning*, Seoul, South Korea. PMLR 306, 2026. Copyright 2026 by the author(s).

- **ABC-based theory for momentum LMO methods.** We provide a unified nonconvex analysis under the ABC condition for a broad LMO-family that subsumes normalized SGDM, signSGD with momentum, and Muon-style updates.

- **A self-bounding closure for ABC-induced history coupling.** We introduce a closure technique that converts the history-coupled terms induced by ABC into explicit rates.

- **Critical batch size and momentum scaling.** We derive the critical mini-batch size $b^\star$ that minimizes sample complexity under $\tau$-nice sampling (Corollary 6.2). Furthermore, our theory predicts a precise scaling law: the optimal momentum $\alpha^\star(b)$ and step size $\gamma^\star(b)$ grow with the batch size.

- **Empirical validation of hyperparameter shifts.** We confirm these theoretical predictions on linear and matrix regression tasks. The experiments reveal the predicted "U-shaped" sample complexity and verify the necessity of increasing momentum as batch size grows, a phenomenon that classical bounded-variance analysis cannot explain.

## 2. Related Work and Background

### 2.1. LMO-Based Optimization and Muon-Style Updates

Linear-minimization-oracle (LMO) updates compute a descent direction by solving a linear problem over a unit ball in a chosen geometry, i.e., they implement steepest descent in a non-Euclidean norm. This umbrella includes sign-based directions (the $\ell_\infty$ geometry), normalized SGD (the $\ell_2$ geometry), and the spectral-/orthogonalization-based update underlying Muon. Most existing convergence analyses for these methods are derived under classical noise models for stochastic gradients – *uniformly* bounded variance or, more generally, uniformly bounded central $p$-th moments[1] for $p \in (1, 2]$ – or under stronger growth-type conditions.

**Sign and Normalized Directions.** Bernstein et al. (2018) analyze SignSGD and its momentum variant (Signum), and provide nonconvex convergence guarantees in an $\ell_\infty$-type geometry; however, the guarantees require large mini-batches (or additional distributional assumptions on the noise). The line of work on biased gradient compression shows that the sign operator can be "repaired" by *error feedback* (Seide et al., 2014): Karimireddy et al. (2019) show that error feedback fixes SignSGD (and more general

---

[1] That is, there exist $p \in (1, 2]$ and $\sigma \geq 0$ such that $\mathbb{E}\|g(x) - \nabla f(x)\|_2^p \leq \sigma^p$ holds uniformly for all $x \in \mathbb{R}^d$. As with uniformly bounded variance, this assumption may fail under subsampling.

compression schemes), recovering SGD-type convergence rates under standard smoothness/second-moment assumptions. For normalized directions, Cutkosky & Mehta (2020) analyze normalized SGD with Polyak momentum (a special case of the LMO-family studied here) and obtain optimal nonconvex rates under uniformly bounded variance, showing in particular that momentum removes the need for large batch sizes. Sun et al. (2023) provide analogous convergence results for sign-based updates with momentum. More recently, Hübler et al. (2025) extend normalized/momentum-style methods to heavy-tailed noise and establish convergence under uniformly bounded central $p$-th moments.

**Spectral Descent and Muon-Style Updates.** Early precursors of modern Muon-style updates are the *(preconditioned) spectral descent* methods of Carlson et al. (2015c;a), where the update can also be interpreted as a steepest-descent/LMO step in a matrix norm and deterministic convergence is proved for convex smooth objectives. Carlson et al. (2015b) analyze a stochastic spectral descent variant for convex and nonconvex smooth problems under a multiplicative noise model of the form $\|g^k\|_* \leq t_k\|\nabla f(x^k)\|_*$ almost surely for some $t_k \in [0, 1)$, which is closely related to strong-growth/interpolation assumptions (Schmidt & Roux, 2013; Vaswani et al., 2019).

Motivated by the recent practical success of Muon and related norm-constrained updates in deep learning, several papers develop convergence guarantees under bounded-variance-type assumptions. Li & Hong (2025) derive (to our knowledge) the first nonconvex convergence bounds for Muon under uniformly bounded variance. Pethick et al. (2025a) analyze a general LMO-family (including Muon as a special case) via the stochastic conditional gradient / stochastic Frank–Wolfe viewpoint (Frank & Wolfe, 1956; Clarkson, 2010; Jaggi, 2013; Mokhtari et al., 2020), covering both constrained and unconstrained settings. Kovalev (2025) obtain related bounds from a stochastic trust-region perspective and further analyze variants with weight decay (under star-convexity) and, assuming second-order smoothness, extrapolation.

A number of follow-up works consider algorithmic variants and refinements, but still rely on uniformly bounded variance (Riabinin et al., 2025; Pethick et al., 2025b; Gruntkowska et al., 2025; Khaled et al., 2026; Takezawa et al., 2026; Shulgin et al., 2026; Gruntkowska et al., 2026; Kovalev & Borodich, 2025; Qian et al., 2025) or uniformly bounded $p$-th moments (Sfyraki & Wang, 2025).

**Stochastic Frank-Wolfe.** Finally, there is a substantial literature on stochastic Frank-Wolfe (SFW) methods for *constrained* optimization. These methods also rely on an LMO, but use it to maintain feasibility via convex combinations, which makes their analyses and update structure different

from the momentum-based LMO-family considered here (e.g., Normalized SGDM, signSGD with momentum, and Muon). Representative results include nonconvex analyses under bounded stochastic gradients/variance (Reddi et al., 2016; Qu et al., 2018) as well as convex analyses (Mokhtari et al., 2020).

## 2.2. Beyond Bounded Variance

Uniformly bounded variance (or uniformly bounded central $p$-th moments) is known to be restrictive in subsampling/mini-batching regimes for finite-sum objectives; see Section 3 for an example, when this assumption fails. This has motivated a range of alternative "noise-growth" assumptions in the SGD literature, including maximal strong growth $\|g(x)\|_2^2 \le B\|\nabla f(x)\|_2^2$ (Tseng, 1998; Schmidt & Roux, 2013), relaxed strong growth $\mathbb{E}\|g(x)\|_2^2 \le B\|\nabla f(x)\|_2^2 + C$ (Bottou et al., 2018; Vaswani et al., 2019), gradient confusion $\langle \nabla f_i(x), \nabla f_j(x) \rangle \ge -\eta$ (Sankararaman et al., 2020), and expected smoothness $\mathbb{E}\|g(x)\|_2^2 \le A(f(x) - f_{\inf}) + C$ (Gower et al., 2019). Khaled & Richtárik (2023) proposed the more general *ABC condition* $\mathbb{E}\|g(x)\|_2^2 \le A(f(x) - f_{\inf}) + B\|\nabla f(x)\|_2^2 + C$, showed that it subsumes many of the above assumptions (including bounded variance), verified that it holds for common subsampling schemes, and derived the first nonconvex SGD rates under this model. Li & Richtárik (2020) later adjusted this condition to obtain a unified analysis of a broad class of SGD-type (distributed) methods with and without variance reduction and communication compression. Beyond being technically convenient, the ABC condition enables explicit reasoning about batch-size effects; for instance, Khaled & Richtárik (2023) derive (theoretically) optimal, non-trivial mini-batch sizes for SGD under importance sampling and $b$-nice sampling.

In the context of LMO-based methods, the closest work to ours is Nazykov et al. (2024), which proposes a unified analysis of a broad class of variance-reduced SFW methods and provides explicit convergence bounds for both convex and nonconvex objectives. Their framework is based on recursion-type conditions controlling the tracking error of gradient surrogates and an auxiliary variance sequence (similar in spirit to unified analyses for SGD-type methods such as Gorbunov et al. (2020)):

$$\mathbb{E}[\|g^k - \nabla f(x^k)\|^2 \mid x^k] \le (1 - \rho_1)\|g^{k-1} - \nabla f(x^{k-1})\|^2$$
$$+ A\sigma_{k-1}^2 + \eta_{k-1}^2 BD^2 + C,$$
$$\mathbb{E}[\sigma_k^2 \mid x^k] \le (1 - \rho_2)\sigma_{k-1}^2 + \eta_{k-1}^2 ED^2.$$

Under smoothness of each summand in the finite-sum, they verify these conditions for several variance-reduced and distributed SFW variants and derive general complexity bounds that match or improve previously known results (Reddi et al., 2016; Hazan & Luo, 2016; Yurtsever et al.,

2019; Négiar et al., 2020; Lu & Freund, 2021; Weber & Sra, 2022; Beznosikov et al., 2024). However, to the best of our knowledge, there has been no subsampling-friendly (ABC-/expected-smoothness-type) analysis for the momentum-based LMO-family considered here (in particular, Normalized SGDM, signSGD with momentum, and Muon; cf. (11)). The SFW results above do not cover this algorithmic family, which is the gap addressed by our work.

## 3. Preliminaries and Assumptions

We study unconstrained minimization

$$\min_{x \in \mathbb{R}^d} f(x), \quad (1)$$

where $f : \mathbb{R}^d \to \mathbb{R}$ is differentiable and bounded below by $f_{\inf} > -\infty$. We equip $\mathbb{R}^d$ with a (possibly non-Euclidean) norm $\|\cdot\|$ and its dual norm $\|v\|_\star := \max_{\|d\| \le 1}\langle v, d\rangle$, where $\langle \cdot, \cdot \rangle$ denotes the standard inner product.

We assume $f$ is $L$-smooth with respect to $\|\cdot\|$, meaning that for all $x, y \in \mathbb{R}^d$,

$$\|\nabla f(x) - \nabla f(y)\|_\star \le L\|x - y\|. \quad (2)$$

**Stochastic Oracle and the ABC Condition.** At each iterate, the algorithm queries an unbiased stochastic first-order oracle: for every $x \in \mathbb{R}^d$, the oracle returns a random vector $g(x)$ such that $\mathbb{E}[g(x)|x] = \nabla f(x)$. Our main analysis is based on the ABC condition introduced by Khaled & Richtárik (2023), which controls the second moment of the stochastic gradient by a combination of function suboptimality, gradient norm, and a residual term: for all $x \in \mathbb{R}^d$,

$$\mathbb{E}\|g(x)\|_2^2 \le 2A(f(x) - f_{\inf}) + B\|\nabla f(x)\|_2^2 + C, \quad (3)$$

where $A, B, C \ge 0$ and $\|\cdot\|_2$ is the Euclidean norm. This condition strictly generalizes the classical bounded-variance model (which corresponds to $A = 0$, $B = 1$, $C = \sigma^2$ under unbiasedness).

Since our update directions are defined via $\|\cdot\|$ while (3) is stated in $\|\cdot\|_2$, we will occasionally use standard norm-compatibility constants (finite-dimensional norm equivalence): there exist $\rho, \bar{\kappa} \ge 1$ such that for all $v \in \mathbb{R}^d$,

$$\|v\|_\star \le \rho\|v\|_2, \qquad \|v\|_2 \le \bar{\kappa}\|v\|_\star. \quad (4)$$

**Limitation of Bounded Variance.** A key motivation for adopting (3) is that uniformly bounded variance (and, similarly, uniformly bounded central $p$-th moment) may fail even in simple finite-sum problems on unbounded domains. Consider a quadratic finite sum

$$f(x) = \frac{1}{n}\sum_{i=1}^{n} f_i(x), \qquad f_i(x) = \frac{1}{2}x^\top H_i x, \quad (5)$$

with symmetric matrices $H_i \succeq 0$. If we sample $I$ uniformly from $\{1, \ldots, n\}$ and use the subsampled gradient $g(x) = \nabla f_I(x) = H_I x$, then $\nabla f(x) = \bar{H} x$ with $\bar{H} = \frac{1}{n} \sum_{i=1}^n H_i$, and variance

$$\mathbb{E}\|g(x) - \nabla f(x)\|_2^2 = x^\top \Sigma x, \qquad (6)$$

where $\Sigma = \frac{1}{n} \sum_{i=1}^n (H_i - \bar{H})^2$. Unless $\Sigma = 0$ (i.e., all $H_i$ coincide), the variance in (6) grows proportionally to $\|x\|_2^2$ along directions of positive curvature of $\Sigma$, and hence cannot be uniformly bounded by a constant $\sigma^2$ over $\mathbb{R}^d$.

At the same time, the ABC condition can hold for this example. If each $f_i$ is $L_i$-smooth and bounded below by $f_i^{\text{inf}}$, then the self-bounding inequality implies $\|\nabla f_i(x)\|_2^2 \leq 2L_i(f_i(x) - f_i^{\text{inf}})$, and therefore

$$\mathbb{E}\|g(x)\|_2^2 = \frac{1}{n} \sum_{i=1}^n \|\nabla f_i(x)\|_2^2 \leq \frac{2}{n} \sum_{i=1}^n L_i(f_i(x) - f_i^{\text{inf}})$$
$$\leq 2L_{\max}(f(x) - f_{\inf}) + 2L_{\max}\Delta_{\inf}, \qquad (7)$$

where $L_{\max} = \max_i L_i$ and

$$\Delta_{\inf} := f_{\inf} - \frac{1}{n} \sum_{i=1}^n f_i^{\text{inf}} \geq 0. \qquad (8)$$

Thus, even when bounded variance fails, (7) provides an ABC-type control with explicit problem-dependent constants.

**$\tau$-Nice Mini-Batching.** In the finite-sum setting (5), we also consider mini-batch gradients constructed by sampling a subset $S \subseteq \{1, \ldots, n\}$ of size $b$ uniformly without replacement ($\tau$-nice sampling with $\tau = b$) and setting

$$g_b(x) = \frac{1}{b} \sum_{i \in S} \nabla f_i(x). \qquad (9)$$

This oracle is unbiased, and the ABC constants can be chosen to explicitly reflect the batch size (Khaled & Richtárik, 2023):

$$A_b = \frac{n-b}{b(n-1)} L_{\max}, \qquad B_b = \frac{n(b-1)}{b(n-1)} \in [0,1],$$
$$C_b = 2A_b \Delta_{\inf}. \qquad (10)$$

Two qualitative features are worth noting: (i) the finite-population correction factor $(n-b)/(n-1)$ ensures $A_b \to 0$ and $C_b \to 0$ as $b \to n$, recovering the full-gradient regime; (ii) for $b \ll n$, the scaling $A_b \approx L_{\max}/b$ resembles the classical $1/b$ variance reduction, but (10) also captures the vanishing-noise limit at $b = n$ and the role of heterogeneity through $\Delta_{\inf}$. These batch-dependent ABC parameters will be the basis for our complexity and batch-size discussions.

## 4. Algorithm Family

We consider the following generic momentum-based update built around a *linear minimization oracle* (LMO):

$$m^{k+1} = (1 - \alpha)m^k + \alpha g^k,$$
$$d^k \in \text{LMO}(m^{k+1}), \qquad (11)$$
$$x^{k+1} = x^k + \gamma_k d^k,$$

where $\alpha \in (0,1)$ is the momentum parameter, $\gamma_k > 0$ is the step size, and $g^k = g(x^k)$ is an unbiased stochastic gradient estimate at $x^k$. The LMO is defined as

$$\text{LMO}(v) \in \arg \min_{\|d\| \leq 1} \langle v, d \rangle,$$

so that, by definition of the dual norm,

$$\langle v, d \rangle = -\|v\|_\star \qquad \text{for any } d \in \text{LMO}(v). \qquad (12)$$

This template encompasses a broad family of optimizers. Different choices of the norm $\|\cdot\|$ induce different geometries and recover several popular algorithms as special cases.

**Normalized SGD (Euclidean Norm).** If $\|\cdot\| = \|\cdot\|_2$ is the Euclidean norm, then the LMO has the explicit form

$$\text{LMO}(v) = -\frac{v}{\|v\|_2},$$

and the update (11) becomes

$$x^{k+1} = x^k - \gamma_k \frac{m^{k+1}}{\|m^{k+1}\|_2},$$

which corresponds to *normalized SGD with momentum* studied, for example, by Cutkosky & Mehta (2020). In this case, the method performs steepest descent with respect to the Euclidean geometry.

**SignSGD (Infinity Norm).** If $\|\cdot\| = \|\cdot\|_\infty$, then the unit ball is the hypercube $\{d : \|d\|_\infty \leq 1\}$ and the LMO reduces to

$$\text{LMO}(v) = -\text{sign}(v),$$

applied coordinate-wise. The resulting update is SignSGD with momentum, where only the signs of the momentum components are used to determine the update. This choice corresponds to steepest descent under the $\ell_\infty$ geometry.

**Muon (Spectral Norm).** When the parameter $x$ represents a matrix and $\|\cdot\| = \|\cdot\|_{\text{sp}}$ is the spectral norm, the LMO takes the form

$$\text{LMO}(V) \in \arg \min_{\|D\|_{\text{sp}} \leq 1} \langle V, D \rangle,$$

where $\langle V, D\rangle = \text{trace}(V^\top D)$. The solution is given by the orthogonal polar factor of $-V$: if $V$ is square matrix and $V = U\Sigma W^\top$ is a singular value decomposition, then $\text{LMO}(V) = -UW^\top$. For the general (not necessarily square) matrices, we have

$$\text{LMO}(V) = -\left(VV^\top\right)^{\frac{\dagger}{2}} V,$$

where the exponent $\frac{\dagger}{2}$ refers to the the square root of the Moore-Penrose pseudoinverse. Substituting this into (11) yields the *Muon* update, which performs steepest descent with respect to the spectral-norm geometry. This interpretation was formalized by Bernstein & Newhouse (2024) and connects Muon to non-Euclidean trust-region and LMO-based optimization frameworks (e.g., Pethick et al. (2025a); Kovalev (2025)).

**Unified Viewpoint.** Across all these examples, the update direction $d^k$ is the steepest descent direction for the current momentum $m^{k+1}$ under the chosen norm. Thus, (11) can be seen as a *norm-adaptive steepest descent method with momentum*. Our analysis treats this entire LMO family in a unified manner and establishes convergence guarantees under the general ABC condition, with bounded variance appearing as a special case.

# 5. Main Result

Our main result is a master bound for the momentum-based LMO family under the ABC condition. The full statement is given in Appendix A, see Theorem A.1. A horizon-tuned specialization is stated in Corollary A.2.

**Theorem 5.1 (Informal).** *Consider the method (11) with parameters*

$$\alpha = \frac{\alpha_0}{\sqrt{K}}, \qquad \gamma = \frac{\gamma_0}{K^{3/4}},$$

*for $\alpha_0 \in (0, \sqrt{K}]$ and $\gamma_0 > 0$. Let $\Delta_0 := f(x^0) - f_{\inf}$ and $E_0 := \mathbb{E}\|e^{-1}\|_\star$. Define the effective ABC constant $A_{\text{eff}} := A + L\bar\kappa^2(B-1)_+$ and the $K$-independent scale*

$$D_0 := \Delta_0 + \frac{L\gamma_0^2}{\alpha_0} + \gamma_0\sqrt{C\alpha_0}.$$

*Then the iterates of (11) satisfy the following bound on $\frac{1}{K}\sum_{k=0}^{K-1}\mathbb{E}\|\nabla f(x^{k+1})\|_\star$:*

$$\widetilde{\mathcal{O}}\Bigg(\frac{E_0}{\alpha_0}K^{-1/2} + L\gamma_0 K^{-3/4} + \left[\frac{\Delta_0}{\gamma_0} + \frac{L\gamma_0}{\alpha_0}\right]K^{-1/4}$$

$$+ \left[A_{\text{eff}}\gamma_0\alpha_0 + \sqrt{C\alpha_0} + \sqrt{A_{\text{eff}}\alpha_0 D_0}\right]K^{-1/4}\Bigg), \quad (13)$$

*where $\widetilde{\mathcal{O}}(\cdot)$ hides absolute constants, mild logarithmic factors, and norm-equivalence constants. In particular, taking*

$\alpha_0, \gamma_0 = \Theta(1)$ *yields the canonical $\widetilde{\mathcal{O}}(K^{-1/4})$ stochastic envelope (and hence $\widetilde{\mathcal{O}}(\varepsilon^{-4})$ iterations for $\varepsilon$-stationarity).*

The decomposition (13) isolates three qualitatively different effects:

- The $K^{-1/4}$ term is the *dominant* stochastic term induced by the ABC-controlled noise and it also contains the contribution of $C$ (which decays under the horizon-tuned $\alpha$ choice).

- The $K^{-1/2}$ term captures the effect of the *initial momentum mismatch* $e^{-1}$. This term can dominate for moderate horizons, which is why we keep it explicit.

- The $K^{-3/4}$ term is a higher-order *smoothness bias* contribution coming from the $L\gamma$ components of the descent inequality.

## 5.1. Proof Sketch

The proof has the standard two-layer structure (descent + tracking error), but under ABC an additional *history coupling* appears and must be closed via a self-bounding bootstrap.

**Step 1: Descent Reduces Everything to the Momentum Lag.** Let $e^k := m^{k+1} - \nabla f(x^k)$ be the one-step momentum lag. A smoothness argument combined with the LMO property yields

$$f(x^{k+1}) \leq f(x^k) - \gamma\|\nabla f(x^{k+1})\|_\star + 2\gamma\|e^k\|_\star + O(L\gamma^2). \tag{14}$$

Telescoping over $k = 0, \ldots, K-1$ gives

$$\frac{1}{K}\sum_{k=0}^{K-1}\mathbb{E}\|\nabla f(x^{k+1})\|_\star \leq \frac{\Delta_0}{\gamma K} + O(L\gamma) + \frac{2}{K}\sum_{k=0}^{K-1}\mathbb{E}\|e^k\|_\star. \tag{15}$$

**Step 2: Unroll the Momentum Recursion.** Unrolling $e^k$ splits it into (i) a drift term controlled by smoothness and momentum memory, and (ii) a geometrically weighted sum of centered noise terms $g^i - \nabla f(x^i)$:

$$\frac{1}{K}\sum_{k=0}^{K-1}\mathbb{E}\|e^k\|_\star = O\left(\frac{L\gamma}{\alpha} + \frac{E_0}{\alpha K}\right) + O(\sqrt{C\alpha})$$

$$+ O\left(\sqrt{A_{\text{eff}}\alpha}\sqrt{\frac{1}{K}\sum_{i=0}^{K-1}\Delta^i}\right). \quad (16)$$

where $\Delta^i := f(x^i) - f_{\inf}$ and $A_{\text{eff}} := A + L\bar\kappa^2(B-1)_+$.

**Step 3 (Novelty): The ABC Closure.** The term $\sqrt{A_{\text{eff}}\alpha}\sqrt{\frac{1}{K}\sum_i\Delta^i}$ is the key difference from bounded-variance analyses: under BV one has $A_{\text{eff}} = 0$, so this

history-coupled term vanishes and the proof closes by summing a geometric series. Under general ABC with $A_{\text{eff}} > 0$, we instead derive a crude upper bound on $U := \frac{1}{K} \sum_{i=0}^{K-1} \Delta^i$ from (14) (dropping the negative descent term), and combine it with (16) to obtain an inequality of the form $U \leq D + E\sqrt{U}$. Solving it gives $\sqrt{U} \leq E + \sqrt{D}$, which closes the loop when substituted back into (16) and then into (15).

**Step 4: Horizon Tuning.** Choosing $\alpha = \alpha_0/\sqrt{K}$ and $\gamma = \gamma_0/K^{3/4}$ makes all ABC-driven contributions scale as $K^{-1/4}$, yielding the explicit horizon-tuned bound in Corollary A.2 and its informal summary (13).

### 5.2. "Optimal" Parameters Under $\tau$-Nice Sampling.

In the finite-sum model with $\tau$-nice mini-batching of size $b \in \{1, \ldots, n-1\}$ (sampling without replacement), the ABC constants satisfy $B_b \leq 1$ and $A_{\text{eff}} = A_b = \frac{n-b}{b(n-1)} L_{\max}$. Minimizing the leading ($K^{-1/4}$) coefficient in the horizon-tuned bound over $\alpha = \alpha_0/\sqrt{K}$ and $\gamma = \gamma_0/K^{3/4}$ yields the parameter scalings

$$\boxed{\begin{aligned} &\alpha^\star(b, K) \asymp \min\left\{1, \sqrt{\frac{b(n-1)}{n-b}} \cdot \frac{1}{\sqrt{K}}\right\}, \\ &\gamma^\star(b, K) \asymp \left(\frac{b(n-1)}{n-b}\right)^{1/4} \cdot \frac{1}{K^{3/4}}, \end{aligned}} \quad (17)$$

where $\asymp$ hides only problem-dependent constants (e.g., $\Delta_0, L, L_{\max}, \rho$) tracked in the Appendix.

**Effect of Batch Size.** Equation (17) makes the finite-population correction explicit:

- **Small-batch regime ($b \ll n$).** Then $\frac{b(n-1)}{n-b} \approx b$, so $\alpha^\star \propto b^{1/2} K^{-1/2}$ and $\gamma^\star \propto b^{1/4} K^{-3/4}$, i.e., larger batches allow larger momentum and larger steps.

- **Large-batch regime ($b \uparrow n$).** The factor $\frac{b(n-1)}{n-b}$ diverges as $n - b \to 0$, reflecting that the stochastic component vanishes; the expression for $\alpha^\star$ saturates at 1 and the method transitions to the deterministic regime.

## 6. Complexity Analysis

**Theorem 6.1** (Iteration complexity (concise $\varepsilon$-splitting; ABC-coupled case)). *Assume the setup of Theorem A.1 and suppose the (effective) history-coupling level is nonzero, i.e., $A_{\text{eff}} > 0$ (equivalently $Q > 0$ below). Define*

$$\Delta_0 := f(x^0) - f_{\inf}, \qquad E_0 := \mathbb{E}\|m^0 - \nabla f(x^0)\|_\star,$$

*and introduce the two stochastic scales*

$$Q := \sqrt{2A_{\text{eff}}}, \qquad S := \max\{\sqrt{C}, Q\sqrt{\Delta_0}\},$$

*where $A_{\text{eff}} := A + L\bar{\kappa}^2(B-1)_+$*

*Then there exist absolute constants $c_\alpha, c_\gamma, c'_\gamma > 0$ such that the following holds. If one chooses the momentum and step size as*

$$\alpha = \min\left\{1, \frac{\varepsilon^2}{c_\alpha S^2}\right\}, \gamma = \min\left\{\frac{\varepsilon}{c_\gamma L\left(1 + \frac{1}{\alpha}\right)}, \frac{\varepsilon}{c'_\gamma Q^2 \alpha K}\right\},$$

*and runs the method* (11) *for*

$$K = \mathcal{O}\left(\frac{L\Delta_0}{\varepsilon^2} \vee \frac{E_0}{\varepsilon}\left(1 \vee \frac{S^2}{\varepsilon^2}\right) \vee \frac{L}{Q^2} \cdot \frac{S^4}{\varepsilon^4}\right), \quad (18)$$

*then it guarantees*

$$\frac{1}{K} \sum_{k=0}^{K-1} \mathbb{E}\|\nabla f(x^{k+1})\|_\star \leq \varepsilon.$$

**Remarks (Interpretation and Regimes).**

- **What the scales mean.** The quantity $S = \max\{\sqrt{C}, Q\sqrt{\Delta_0}\}$ combines (i) the *pure-noise* level $\sqrt{C}$ and (ii) the *history-coupling* level $Q\sqrt{\Delta_0}$ induced by the ABC term (effective expected smoothness). That is, one can treat $S$ as the single stochastic scale.

- **Deterministic regime ($S \lesssim \varepsilon$).** Then $\alpha$ saturates at 1, and the burn-in term becomes linear, $K = \mathcal{O}(E_0/\varepsilon)$; the baseline $\mathcal{O}(L\Delta_0/\varepsilon^2)$ is recovered with $\gamma \simeq \varepsilon/L$.

- **Stochastic regime ($S \gtrsim \varepsilon$).** Then $\alpha = \Theta(\varepsilon^2/S^2)$ is required to suppress the noise floor, and the canonical $\varepsilon^{-4}$ envelope appears through the last term in (18). The intermediate $\varepsilon^{-3}$ term reflects the initial momentum mismatch $E_0$ (burn-in).

- **Initialization effects.** If $m^0$ is well-aligned (e.g., $m^0 = \nabla f(x^0)$ so $E_0 = 0$), the burn-in terms vanish and the leading stochastic complexity is driven by the quartic term.

- **The case $Q = 0$ (no history coupling).** When $A_{\text{eff}} = 0$ the ABC-coupled terms disappear; one should instead use the bounded-variance specialization (Corollary A.4), which yields the familiar stochastic quartic term driven by $C$ (e.g., $K = \mathcal{O}(L\Delta_0 C/\varepsilon^4)$ up to norm-compatibility factors).

### 6.1. Bounded Variance

Assume a mini-batch oracle with uniformly bounded variance in Euclidean norm, $\mathbb{E}\|g_b(x) - \nabla f(x)\|_2^2 \leq \sigma^2/b$. This corresponds to the ABC parameters $(A, B, C) = (0, 1, \sigma^2/b)$, hence $A_{\mathrm{eff}} = 0$ (and the history-coupled ABC terms vanish). In this regime our bounds reduce to the standard bounded-variance envelope for normalized/LMO methods, matching prior BV analyses (Pethick et al., 2025a; Kovalev, 2025) up to norm-compatibility factors.

We measure mini-batch performance via the *sample complexity* $T(b, \varepsilon) := b\, K(b, \varepsilon)$, i.e., the total number of component-gradient evaluations. Suppressing problem-dependent constants (e.g., $L, \Delta_0, \rho, E_0, \sigma$), our BV specialization yields

$$T_{\mathrm{BV}}(b, \varepsilon) = \widetilde{\mathcal{O}}\left( \frac{b}{\varepsilon^2} \vee \frac{\sqrt{b}}{\varepsilon} \vee \frac{1}{\varepsilon^3} \vee \frac{1}{\varepsilon^4} \right). \quad (19)$$

**Batch-Size Implication Under BV.** Every $b$-dependent branch in (19) is nondecreasing in $b$, while the dominant stochastic $\varepsilon^{-4}$ term is independent of $b$; hence BV predicts that $T_{\mathrm{BV}}(b, \varepsilon)$ is minimized at $b = 1$ and mini-batching cannot yield a nontrivial sample-complexity optimum. This mirrors the analogous BV prediction for mini-batch SGD, but it fails to explain the strong optimizer-dependent batch-size sensitivity observed in modern deep learning/LLM training (e.g., the widening SGD–Adam gap at large batch sizes).

### 6.2. $\tau$-Nice Sampling (Finite-Sum Mini-Batching)

Consider the finite-sum model and $\tau$-nice sampling without replacement with batch size $b \in \{1, \ldots, n\}$. In this case the ABC parameters satisfy $A(b) = \frac{n-b}{b(n-1)} L_{\max}$ and $C(b) = 2A(b)\Delta_{\inf}$ (and $B(b) \leq 1$), so that $A(b), C(b) \to 0$ as $b \to n$. Plugging these constants into our general bound yields the sample complexity $T_\tau(b, \varepsilon)$

$$\widetilde{\mathcal{O}}\left( \frac{b}{\varepsilon^2} \vee \frac{b}{\varepsilon} \vee \frac{n-b}{n-1} \cdot \frac{1}{\varepsilon^3} \vee \frac{n-b}{n-1} \cdot \frac{1}{\varepsilon^4} \right), \quad (20)$$

where we again suppress problem-dependent constants.

**Batch-Size Implication Under $\tau$-Nice Sampling.** Unlike BV, the dominant stochastic branches in (20) retain the finite-population correction $(n-b)/(n-1)$ and therefore *decrease* with $b$, vanishing in the full-gradient limit $b = n$. Consequently, $T_\tau(b, \varepsilon)$ exhibits a genuine trade-off between an *increasing* deterministic branch ($b$ gradients per step) and *decreasing* stochastic branches (noise shrinks with $b$), implying a nontrivial critical-batch phenomenon.

**Corollary 6.2** (Critical Batch Size). *The sample complexity bound $T_\tau(b, \varepsilon)$ in (20) is minimized at a unique critical*

mini-batch size $b^\star$. *Define the heterogeneity ratio*

$$H_{\mathrm{norm}}(\varepsilon) := \frac{\rho^2 L_{\max} \Delta_*^2}{\Delta_0\,(n-1)\,\varepsilon^2},$$

*where* $\Delta_* = \max\{\Delta_0, \Delta_{\inf}\}$. *The critical batch size is given by*

$$b^\star(\varepsilon) \approx \frac{H_{\mathrm{norm}}(\varepsilon)}{1 + H_{\mathrm{norm}}(\varepsilon)} \cdot n. \quad (21)$$

*Remark* 6.3 (Interpretation of Regimes). Equation (21) governs the transition between stochastic and deterministic regimes:

- **Noise-Dominated** ($H \ll 1$)**:** When targeting moderate accuracy (large $\varepsilon$), $b^\star \approx nH \ll n$. In this regime, small batches are sample-efficient because the noise reduction from larger batches does not justify their higher per-step cost.

- **Curvature-Dominated** ($H \gg 1$)**:** As $\varepsilon \to 0$, $H$ diverges and $b^\star \to n$. High-precision training eventually favors full-batch (deterministic) updates to eliminate the $\varepsilon^{-4}$ stochastic variance term entirely.

- **Sweet Spot:** For intermediate $\varepsilon$, $b^\star$ lies strictly in $(1, n)$, predicting the "U-shaped" sample complexity curves observed in Figure 1.

## 7. Experiments

### 7.1. Convex Objective

We consider linear and matrix regression problems; the technical details are deferred to Appendix C.

We study the sample complexity of LMO-family methods by measuring the minimum number of stochastic gradient evaluations required to reach an $\varepsilon$-first-order stationary point, as a function of the mini-batch size. For each choice of the momentum parameter $\alpha$, we sweep the batch size $b$ over $\{1, \ldots, 1000\}$, where $b = 1000$ corresponds to full-batch updates. For every $(\alpha, b)$ pair, we tune the stepsize $\gamma$ to find the minimum number of iterations to reach the $\varepsilon$-first-order stationary point. For fair comparison, the number of iterations is converted into the total gradient evaluation done as the number of epochs. Figures 1 and 2 summarize the dependence of the (best-tuned) convergence speed on the mini-batch size. Each point in all figures reports the mean over three independent runs (with error band) to account for stochasticity.

In the easier instance (larger $\varepsilon = 10^{-2}$) of linear regression (Figure 1a), the optimal regime for small momentum is attained at the smallest possible batch size: the critical batch size is $b^* = 1$ for $\alpha \leq 0.5$. As $\alpha$ increases, the empirically

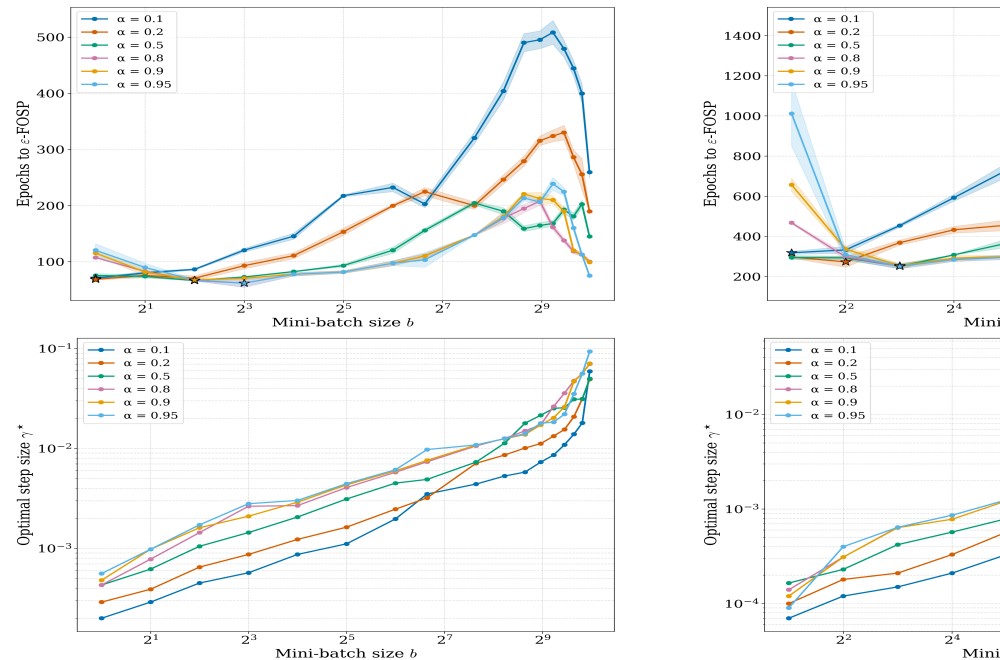

*(a)* The effect of varying $\alpha$ under fixed $\varepsilon = 10^{-2}$ (top) and the optimal step size $\gamma^*$ found with grid search (bottom).

*(b)* Different $\varepsilon = 5 \times 10^{-3}$ (top) and the corresponding optimal step size $\gamma^*$ (bottom).

*Figure 1.* Linear regression via NSGD with momentum.

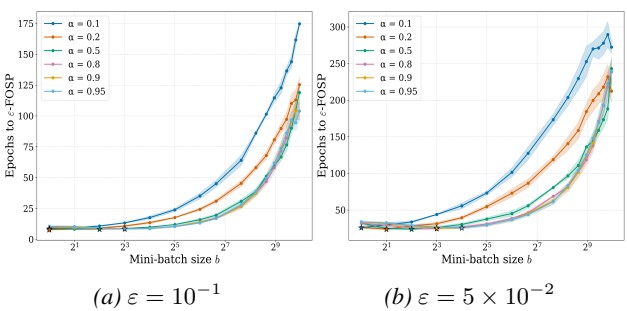

*(a)* $\varepsilon = 10^{-1}$        *(b)* $\varepsilon = 5 \times 10^{-2}$

*Figure 2.* Matrix regression via Muon.

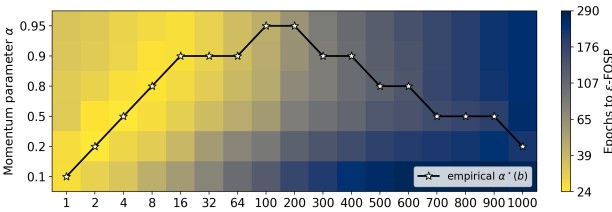

*Figure 3.* Matrix regression via Muon with $\varepsilon = 5 \times 10^{-2}$

optimal batch size shifts to larger values. This qualitative trend is consistent with our $\tau$-nice analysis in (17), which predicts that the optimal momentum and effective stepsize parameters (denoted $\alpha^\star$ and $\gamma^\star$ in our theory) increase with the batch size $b$.

For the harder instance (smaller $\varepsilon = 5 \times 10^{-3}$) (Figure 1b), the presence of a critical batch size becomes more pronounced; however, the best performance is achieved in the deterministic regime (i.e., at full-batch updates). Interestingly, we also observe a secondary (pseudo-)critical batch size at relatively large $b$ when $\alpha$ is small. This suggests that the common practice of LLM training (large mini-batch size and small momentum) lies in this pseudo-critical batch size which aligned with recent observations in LLM training (Marek et al., 2025).

For matrix regression (Figure 2), we use the idealized Muon (with update based on exact SVD) variant. In the easier instance (Figure 2a), the behavior closely matches the linear-regression case: the empirical critical batch size increases from $b_{\text{crit}} = 1$ to $b_{\text{crit}} = 16$ as $\alpha$ increases from 0.1 to 0.9. For large mini-batches in this easy setting, we do not observe a pronounced "climax" (i.e., a deterioration at intermediate $b$ followed by improvement at full batch). However, this non-monotone pattern reappears in the harder instance (Figure 2b, $\alpha = 0.1$), suggesting that the presence of such a climax is tied to problem difficulty.

As the instance becomes harder, Muon and NSGD with momentum exhibit increasingly similar critical-batch behavior: the optimal regime occurs at $b^* = 1$ for small $\alpha$, and $b^*$ shifts to larger values as $\alpha$ increases. This trend is consistent with our theoretical predictions. Furthermore, the behavior of empirically optimal stepsize $\gamma^\star$ for Muon is essentially

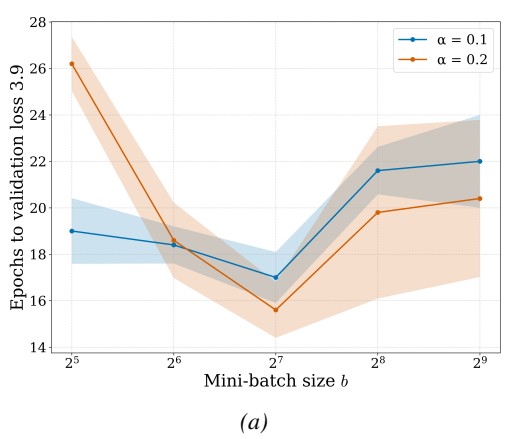
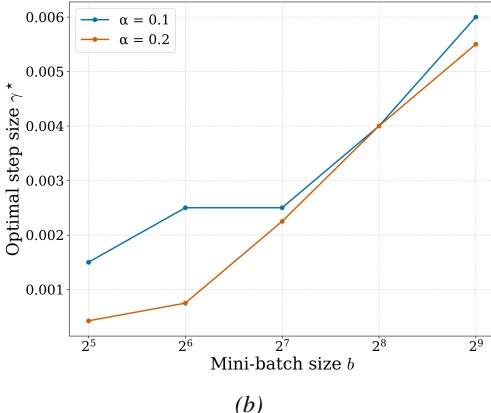

*(a)*                                          *(b)*

*Figure 4.* Text prediction via practical mix of Muon/AdamW.

very similar to that observed for NSGD with momentum across our sweep, for more details, see Appendix C.

A clearer visualization of the agreement between theory and empirics is provided by the heatmap in Figure 3. Moving along the main diagonal, the colors shift toward green, indicating that the empirically optimal momentum $\alpha^\star$ increases with the mini-batch size $b$, in line with our theoretical prediction. For full picture and full experimental details, see Appendix C.

### 7.2. Non-Convex Objective

In this experiment, we examine whether our theoretical insights extend to a realistic deep learning task, namely next-token prediction. We conduct the experiment on the WikiText-103 dataset (Merity et al., 2017), using approximately 120M training tokens. Following the Chinchilla scaling law (Hoffmann et al., 2022), we train a Nano-GPT model (Karpathy, 2022) with approximately 6M parameters on A100 GPU. Additional technical details are provided in Appendix C.

We study the interaction among the momentum parameter, step size, and batch size by measuring the minimum number of epochs required to reach a target loss for this complex nonconvex objective. For each momentum value $\alpha \in \{0.1, 0.2\}$, we consider batch sizes $b \in \{32, 64, 128, 256, 512\}$. For every $(\alpha, b)$ pair, we tune the step size $\gamma$ to minimize the number of epochs needed to achieve a validation cross-entropy loss below 3.9.

We report the average number of epochs required to reach the target validation loss over five runs with different random seeds, as we observed high variance at larger learning rates. We also report the optimal step size selected from the grid $10^{-3}, \ldots, 8 \times 10^{-3}$ with spacing $5 \times 10^{-4}$, except for batch sizes 32 and 64, for which the optimal value lies below this range.

As shown in Fig. 4, we observe the same critical-batch-size behavior as in the regression experiment, particularly the U-shaped trend in the small-batch regime. As the momentum parameter $\alpha$ increases from 0.1 to 0.2, this U-shape becomes more pronounced, mirroring the behavior observed for linear regression with NSGD at smaller values of $\varepsilon$ in Fig. 1b. The optimal step size $\gamma^*$ follows a similar pattern to that in the regression experiment, increasing with the batch size.

## 8. Conclusion

We developed an ABC (expected smoothness) convergence theory for a broad family of momentum LMO-based optimizers, unifying normalized, sign-type, and Muon-style updates. The key technical ingredient is a self-bounding closure that controls the momentum-induced history coupling under ABC, yielding explicit nonconvex master bounds and iteration/sample-complexity guarantees beyond bounded-variance analyses. Specializing to $\tau$-nice mini-batching, we obtain explicit batch-size tradeoffs (including critical-batch behavior) and scaling laws for near-optimal momentum and step size as functions of batch size. Experiments on linear and matrix regression corroborate the predicted batch-size sensitivity and hyperparameter trends.

## Acknowledgements

The research reported in this publication was supported by funding from King Abdullah University of Science and Technology (KAUST): i) KAUST Baseline Research Scheme, ii) Center of Excellence for Generative AI, under award number 5940, iii) SDAIA-KAUST Center of Excellence in Artificial Intelligence and Data Science.

## Impact Statement

This paper presents work whose goal is to advance the field of Machine Learning. There are many potential societal consequences of our work, none which we feel must be specifically highlighted here.

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

# Contents

## A. Analysis

### A.1. Further Special Cases of ABC

Let $f(x) = \frac{1}{n} \sum_{i=1}^n f_i(x)$, with each $f_i$ being $L_i$-smooth and bounded below by $f_i^{\inf}$. Let $\Delta_{\inf} := \frac{1}{n} \sum_{i=1}^n (f_{\inf} - f_i^{\inf}) \geq 0$. Then under standard sampling schemes the ABC constants are (all statements assume unbiasedness of the estimator):

- **Independent sampling with replacement** (minibatch size $\tau$, probabilities $q_i > 0$, $\sum_i q_i = 1$):

$$A = \max_i \frac{L_i}{\tau \, n \, q_i}, \qquad B = 1 - \frac{1}{\tau}, \qquad C = 2A \, \Delta_{\inf}.$$

    For uniform sampling $q_i = \frac{1}{n}$, this simplifies to $A = \frac{1}{\tau} \max_i L_i$, $B = 1 - \frac{1}{\tau}$.

- **Independent sampling without replacement** (inclusion probabilities $p_i \in (0, 1]$):

$$A = \max_i \frac{(1 - p_i) \, L_i}{p_i \, n}, \qquad B = 1, \qquad C = 2A \, \Delta_{\inf}.$$

- **$\tau$-nice sampling without replacement** (uniform over all subsets of size $\tau$):

$$A = \frac{n - \tau}{\tau \, (n - 1)} \, \max_i L_i, \qquad B = \frac{n(\tau - 1)}{\tau \, (n - 1)} \in [0, 1], \qquad C = 2A \, \Delta_{\inf}.$$

### A.2. Proof Sketch

This section provides a high-level proof sketch of Theorem A.1 and highlights the main technical differences compared to prior analyses of normalized/LMO methods under bounded variance, e.g., Cutkosky & Mehta (2020); Kovalev (2025); Pethick et al. (2025a). Our analysis works under the more general *ABC* condition of Khaled & Richtárik (2023). To keep the sketch readable, we hide universal numerical constants and norm-equivalence constants (e.g., $\rho$, $\bar{\kappa}$) inside $\widetilde{\mathcal{O}}(\cdot)$ notation.

**Step 1: One-Step Descent Reduces the Problem to Bounding the Momentum Lag.** Using $L$-smoothness and the defining property of the LMO direction, the update admits a deterministic descent inequality of the form

$$f(x^{k+1}) \leq f(x^k) - \gamma \|\nabla f(x^{k+1})\|_\star + 2\gamma \|e^k\|_\star + \widetilde{\mathcal{O}}(L\gamma^2), \tag{22}$$

where the (one-step) momentum lag is $e^k = m^{k+1} - \nabla f(x^k)$. Summing (22) over $k = 0, \dots, K - 1$ and telescoping the function values yields the *pre-master* bound

$$\frac{1}{K} \sum_{k=0}^{K-1} \mathbb{E}\|\nabla f(x^{k+1})\|_\star \leq \frac{\Delta^0}{\gamma K} + \widetilde{\mathcal{O}}(L\gamma) + \frac{2}{K} \sum_{k=0}^{K-1} \mathbb{E}\|e^k\|_\star. \tag{23}$$

Thus, the convergence analysis reduces to bounding the *average* lag $K^{-1} \sum_{k=0}^{K-1} \mathbb{E}\|e^k\|_\star$.

**Step 2: Momentum-Lag Decomposition (Noise + Drift).** The momentum recursion implies

$$e^k = (1 - \alpha)e^{k-1} + \alpha(g^k - \nabla f(x^k)) + (1 - \alpha)\big(\nabla f(x^{k-1}) - \nabla f(x^k)\big), \tag{24}$$

which can be unrolled into

$$e^k = (1 - \alpha)^{k+1} e^{-1} + V_k + D_k, \tag{25}$$

where $V_k$ is a geometrically weighted sum of the centered stochastic gradients $g^i - \nabla f(x^i)$ and $D_k$ is a geometrically weighted sum of gradient differences $\nabla f(x^{i-1}) - \nabla f(x^i)$. The drift term is controlled using smoothness and the unit-length LMO step:

$$\mathbb{E}\|D_k\|_\star = \widetilde{\mathcal{O}}\left(\frac{L\gamma}{\alpha}\right). \tag{26}$$

**Step 3: Where ABC Breaks the Bounded-Variance Argument.** Under bounded variance (BV), one has $\mathbb{E}\|g(x) - \nabla f(x)\|_2^2 \leq \sigma^2$, so the noise component satisfies the standard estimate

$$\mathbb{E}\|V_k\|_\star = \widetilde{\mathcal{O}}\left(\sigma\sqrt{\frac{\alpha}{2-\alpha}}\right), \tag{27}$$

and the rest of the argument proceeds by summing geometric series, as in Cutkosky & Mehta (2020); Kovalev (2025); Pethick et al. (2025a).

In contrast, under ABC (Khaled & Richtárik, 2023), the second moment of the stochastic gradient depends on the current function value and (potentially) on $\|\nabla f(x)\|_2^2$:

$$\mathbb{E}\|g(x)\|_2^2 \leq 2A(f(x) - f_{\mathrm{inf}}) + B\|\nabla f(x)\|_2^2 + C.$$

Consequently,

$$\mathbb{E}\|g(x) - \nabla f(x)\|_2^2 \leq 2A(f(x) - f_{\mathrm{inf}}) + (B-1)_+\|\nabla f(x)\|_2^2 + C,$$

and therefore $\mathbb{E}\|V_k\|_\star$ is no longer controlled by a constant (such as $\sigma$) but is instead coupled to the *entire optimization history* through the evolving gaps $f(x^i) - f_{\mathrm{inf}}$. This coupling is the main obstacle to extending BV-style momentum analyses to ABC.

**Step 4: Self-Bounding Closes the $(B-1)_+$ Term and Yields an Effective $A_{\mathrm{eff}}$.** To handle the additional $(B-1)_+\|\nabla f(x)\|_2^2$ contribution, we use the self-bounding inequality implied by smoothness in the LMO geometry:

$$\|\nabla f(x)\|_\star^2 \leq 2L(f(x) - f_{\mathrm{inf}}). \tag{28}$$

Combined with norm compatibility (hidden in $\widetilde{\mathcal{O}}(\cdot)$), this converts the $(B-1)_+\|\nabla f(x)\|_2^2$ term into an additional multiple of $f(x) - f_{\mathrm{inf}}$, resulting in an *effective* ABC constant

$$A_{\mathrm{eff}} = A + \widetilde{\mathcal{O}}\big(L(B-1)_+\big). \tag{29}$$

This step is crucial: it shows precisely how the "bad" regime $B > 1$ gets absorbed into a larger $A_{\mathrm{eff}}$, while when $B \leq 1$ there is no inflation beyond $A$.

**Step 5: Averaging Over Time Produces a History Term $S = \sum_{i=0}^{K-1} \Delta^i$.** After applying the martingale-difference property to $V_k$ and Jensen/Cauchy–Schwarz, one obtains an average lag bound of the schematic form

$$\frac{1}{K}\sum_{k=0}^{K-1} \mathbb{E}\|e^k\|_\star = \widetilde{\mathcal{O}}\left(\frac{L\gamma}{\alpha} + \frac{E_0}{\alpha K} + \sqrt{\frac{\alpha}{K}}\sqrt{A_{\mathrm{eff}}\sum_{i=0}^{K-1}\Delta^i + \sqrt{\alpha C}}\right), \tag{30}$$

where $E_0 = \mathbb{E}\|e^{-1}\|_\star$ and $\Delta^i$ denotes the function gaps along the trajectory. Compared to BV, the novelty is the third term in (30), which depends on the history sum $S = \sum_{i=0}^{K-1}\Delta^i$. Plugging (30) into (23) yields a master bound that depends on $\sqrt{S/K}$; hence we must also upper bound $S$.

**Step 6: The "Self-Bounding Closure" for $S$ (Bootstrapping Argument).** A direct bound on $S$ is not available under ABC because the noise magnitude depends on the gaps. Instead, we obtain a *crude* recursion for the gaps by dropping the negative descent term in (22), which gives

$$\Delta^{k+1} \leq \Delta^k + \widetilde{\mathcal{O}}\big(\gamma\mathbb{E}\|e^k\|_\star + L\gamma^2\big).$$

Summing this recursion over $k$ and then over time indices yields an inequality of the form

$$U \leq D + E\sqrt{U}, \qquad U := \frac{1}{K}\sum_{i=0}^{K-1}\Delta^i,$$

where $D$ collects the deterministic and $C$-dependent terms, and $E$ is proportional to $\gamma K\sqrt{\alpha A_{\mathrm{eff}}}$ (up to hidden constants). Solving the quadratic inequality in $\sqrt{U}$ yields $\sqrt{U} \leq E + \sqrt{D}$, which is then substituted back into (30) and finally into (23). This bootstrapping step is the main technical novelty compared to BV analyses: it closes the loop created by the ABC coupling between variance and function values.

**Step 7: Horizon-Tuned Parameters and the Multi-Term Rate.** Choosing constant (across iterations) but horizon-dependent parameters $\alpha = \alpha_0 K^{-1/2}$ and $\gamma = \gamma_0 K^{-3/4}$ balances the deterministic term $\Delta^0/(\gamma K)$ against the ABC-driven terms originating from the closure above. At a high level, the resulting guarantee has the form

$$
\frac{1}{K}\sum_{k=0}^{K-1}\mathbb{E}\|\nabla f(x^{k+1})\|_\star \leq \frac{\widetilde{\mathcal{O}}\Big(\Delta^0/\gamma_0 + L\gamma_0/\alpha_0 + A_{\mathrm{eff}}\gamma_0\alpha_0 + \sqrt{C\alpha_0} + \sqrt{A_{\mathrm{eff}}\alpha_0}\sqrt{\Delta^0 + L\gamma_0^2/\alpha_0 + \gamma_0\sqrt{C\alpha_0}}\Big)}{K^{1/4}}
$$
$$
+ \frac{\widetilde{\mathcal{O}}(E_0/\alpha_0)}{K^{1/2}} + \frac{\widetilde{\mathcal{O}}(L\gamma_0)}{K^{3/4}}. \tag{31}
$$

The leading term scales as $K^{-1/4}$, which implies the familiar $\widetilde{\mathcal{O}}(\varepsilon^{-4})$ iteration complexity for finding an $\varepsilon$-stationary point in the stochastic setting, while (31) also retains the lower-order $K^{-1/2}$ and $K^{-3/4}$ contributions that are relevant outside the strict asymptotic regime.

**Summary of the Key New Ingredient.** Under bounded variance, the momentum noise term is controlled by a constant (e.g., $\sigma$), and the proof reduces to summing geometric series. Under ABC (Khaled & Richtárik, 2023), the noise level depends on the evolving gaps $\Delta^k$ and the regime $B > 1$ additionally couples to $\|\nabla f(x^k)\|^2$. The combination of the self-bounding inequality (28) (to absorb $B > 1$ into $A_{\mathrm{eff}}$) and the self-bounding closure argument (to control $S = \sum \Delta^k$ despite the coupling) is what enables a unified convergence theory for normalized/LMO-family methods with momentum under ABC.

### A.3. Detailed Proof

**Start of the Analysis: A Per-Step Descent Inequality** We work with the filtration $\mathcal{F}_k = \sigma(x^0, g^0, \ldots, x^{k-1}, g^{k-1}, x^k)$, so that $x^k$ and $m^k$ are $\mathcal{F}_k$-measurable, while $g^k$, $m^{k+1}$, $d^k$, and $x^{k+1}$ depend on the new sampling at step $k$.

**Step 1: Single-Step Descent in Terms of $\|m^{k+1}\|_\star$.** By $L$-smoothness with $y = x^{k+1} = x^k + \gamma_k d^k$ and $x = x^k$,

$$
f(x^{k+1}) \leq f(x^k) + \gamma_k \langle \nabla f(x^k), d^k \rangle + \frac{L}{2}\gamma_k^2 \|d^k\|^2. \tag{32}
$$

Insert and subtract $m^{k+1}$ inside the inner product and use $\|d^k\| \leq 1$:

$$
\langle \nabla f(x^k), d^k \rangle = \langle m^{k+1}, d^k \rangle + \langle \nabla f(x^k) - m^{k+1}, d^k \rangle \leq -\|m^{k+1}\|_\star + \|\nabla f(x^k) - m^{k+1}\|_\star,
$$

where we used $\langle m^{k+1}, d^k \rangle = -\|m^{k+1}\|_\star$ and Hölder's inequality. Plugging into (32) gives the deterministic one-step inequality

$$
f(x^{k+1}) \leq f(x^k) - \gamma_k \|m^{k+1}\|_\star + \gamma_k \|\nabla f(x^k) - m^{k+1}\|_\star + \frac{L}{2}\gamma_k^2. \tag{33}
$$

Taking $\mathbb{E}[\cdot \mid \mathcal{F}_k]$ preserves (33) by monotonicity of conditional expectation.

**Step 2: Pass from $\|m^{k+1}\|_\star$ to $\|\nabla f(x^{k+1})\|_\star$.** Apply the reverse triangle inequality,

$$
\|m^{k+1}\|_\star = \|\nabla f(x^{k+1}) - (\nabla f(x^{k+1}) - m^{k+1})\|_\star \geq \|\nabla f(x^{k+1})\|_\star - \|\nabla f(x^{k+1}) - m^{k+1}\|_\star.
$$

Substitute this lower bound for $\|m^{k+1}\|_\star$ into (33) (note the leading minus sign), and add/subtract $\nabla f(x^{k+1})$ inside the remaining error term:

$$
f(x^{k+1}) \leq f(x^k) - \gamma_k \|\nabla f(x^{k+1})\|_\star + 2\gamma_k \|\nabla f(x^{k+1}) - m^{k+1}\|_\star
$$
$$
+ \gamma_k \|\nabla f(x^{k+1}) - \nabla f(x^k)\|_\star + \frac{L}{2}\gamma_k^2. \tag{34}
$$

Next, control the drift using smoothness (Lipschitz gradient in the dual norm):

$$
\|\nabla f(x^{k+1}) - \nabla f(x^k)\|_\star \leq L\|x^{k+1} - x^k\| = L\gamma_k\|d^k\| \leq L\gamma_k.
$$

Define the one-step momentum lag

$$e^k := m^{k+1} - \nabla f(x^k), \qquad \text{so that} \qquad \|\nabla f(x^k) - m^{k+1}\|_\star = \|e^k\|_\star.$$

Then

$$\|\nabla f(x^{k+1}) - m^{k+1}\|_\star \leq \|\nabla f(x^{k+1}) - \nabla f(x^k)\|_\star + \|e^k\|_\star. \tag{35}$$

Using (35) in (34), the two middle terms jointly contribute $3\gamma_k\|\nabla f(x^{k+1}) - \nabla f(x^k)\|_\star + 2\gamma_k\|e^k\|_\star$, and hence (using $\|\nabla f(x^{k+1}) - \nabla f(x^k)\|_\star \leq L\gamma_k$) we obtain the clean per-step inequality

$$f(x^{k+1}) \leq f(x^k) - \gamma_k \|\nabla f(x^{k+1})\|_\star + 2\gamma_k \|e^k\|_\star + \frac{7L}{2}\gamma_k^2. \tag{36}$$

**Step 3: Summation and the Pre-Master Inequality.** Sum (36) for $k = 0, \ldots, K-1$ and take total expectations. The function values telescope and $f(x^K) \geq f_{\inf}$, so $\mathbb{E}[f(x^0) - f(x^K)] \leq \Delta^0$. Thus

$$\sum_{k=0}^{K-1} \gamma_k \, \mathbb{E}\|\nabla f(x^{k+1})\|_\star \leq \Delta^0 + \frac{7L}{2}\sum_{k=0}^{K-1}\gamma_k^2 + 2\sum_{k=0}^{K-1}\gamma_k \, \mathbb{E}\|e^k\|_\star.$$

$$\frac{\sum_{k=0}^{K-1}\gamma_k \, \mathbb{E}\|\nabla f(x^{k+1})\|_\star}{\sum_{k=0}^{K-1}\gamma_k} \leq \frac{\Delta^0}{\sum_{k=0}^{K-1}\gamma_k} + \frac{7L}{2}\cdot\frac{\sum_{k=0}^{K-1}\gamma_k^2}{\sum_{k=0}^{K-1}\gamma_k} + 2\cdot\frac{\sum_{k=0}^{K-1}\gamma_k \, \mathbb{E}\|e^k\|_\star}{\sum_{k=0}^{K-1}\gamma_k}. \tag{37}$$

For constant step size $\gamma_k \equiv \gamma$,

$$\frac{1}{K}\sum_{k=0}^{K-1}\mathbb{E}\|\nabla f(x^{k+1})\|_\star \leq \frac{\Delta^0}{\gamma K} + \frac{7}{2}L\gamma + \frac{2}{K}\sum_{k=0}^{K-1}\mathbb{E}\|e^k\|_\star. \tag{38}$$

**What Remains (Momentum Error).** Inequality (38) reduces the convergence analysis to bounding $\frac{1}{K}\sum_{k=0}^{K-1}\mathbb{E}\|e^k\|_\star$. The momentum recursion yields

$$e^k = (1-\alpha)\,e^{k-1} + \alpha\big(g^k - \nabla f(x^k)\big) + (1-\alpha)\big(\nabla f(x^{k-1}) - \nabla f(x^k)\big), \tag{39}$$

with $e^{-1} := m^0 - \nabla f(x^0)$. Unrolling (39) gives

$$e^k = (1-\alpha)^{k+1}e^{-1} + \underbrace{\sum_{i=0}^{k}\alpha(1-\alpha)^{k-i}\big(g^i - \nabla f(x^i)\big)}_{=:V_k} + \underbrace{\sum_{i=1}^{k}(1-\alpha)^{k-i+1}\big(\nabla f(x^{i-1}) - \nabla f(x^i)\big)}_{=:D_k}. \tag{40}$$

**Self-Bounding Inequality in the $\|\cdot\|$ Geometry.** Since $f$ is $L$-smooth with respect to $\|\cdot\|$ and bounded below by $f_{\inf}$, for all $x \in \mathbb{R}^d$,

$$\|\nabla f(x)\|_\star^2 \leq 2L\big(f(x) - f_{\inf}\big). \tag{41}$$

*Proof.* Let $d_x \in \arg\min_{\|d\|\leq 1}\langle\nabla f(x), d\rangle$, so $\langle\nabla f(x), d_x\rangle = -\|\nabla f(x)\|_\star$ and $\|d_x\| \leq 1$. Apply smoothness with $y = x + \frac{1}{L}\|\nabla f(x)\|_\star d_x$:

$$f(y) \leq f(x) + \frac{1}{L}\|\nabla f(x)\|_\star\langle\nabla f(x), d_x\rangle + \frac{L}{2}\left\|\frac{1}{L}\|\nabla f(x)\|_\star d_x\right\|^2 \leq f(x) - \frac{1}{2L}\|\nabla f(x)\|_\star^2.$$

Since $f_{\inf} \leq f(y)$, rearranging yields (41).

By norm compatibility $\|v\|_2 \leq \bar{\kappa}\,\|v\|_\star$,

$$\|\nabla f(x)\|_2^2 \leq \bar{\kappa}^2\|\nabla f(x)\|_\star^2 \leq 2L\,\bar{\kappa}^2\big(f(x) - f_{\inf}\big), \qquad \Rightarrow \qquad \mathbb{E}\|\nabla f(x^k)\|_2^2 \leq 2L\,\bar{\kappa}^2\,\Delta^k. \tag{42}$$

**Pointwise (Per-$k$) Bounds.** Using $L$-smoothness and $\|x^i - x^{i-1}\| = \gamma\|d^{i-1}\| \leq \gamma$ (unit LMO step), the drift term satisfies

$$\mathbb{E}\|D_k\|_\star \;\leq\; \sum_{i=1}^{k}(1-\alpha)^{k-i+1}\,\mathbb{E}\|\nabla f(x^{i-1}) - \nabla f(x^i)\|_\star \;\leq\; L\gamma\,\frac{1-\alpha}{\alpha}. \tag{43}$$

For the noise term, define geometric weights $w_{k,i} := \alpha(1-\alpha)^{k-i}$ (for $i \leq k$) and

$$z_k^2 \;:=\; \sum_{i=0}^{k} w_{k,i}^2 \;=\; \alpha^2 \sum_{j=0}^{k}(1-\alpha)^{2j} \;\leq\; \frac{\alpha}{2-\alpha}.$$

Let $\xi^i := g^i - \nabla f(x^i)$. Then $\mathbb{E}[\xi^i \mid x^i] = 0$, hence $\mathbb{E}[\xi^i \mid \mathcal{F}_i] = 0$, and for $i < j$, $\mathbb{E}\langle\xi^i, \xi^j\rangle = \mathbb{E}\langle\xi^i, \mathbb{E}[\xi^j \mid \mathcal{F}_j]\rangle = 0$, since $\xi^i$ is $\mathcal{F}_j$-measurable. Therefore $\mathbb{E}\big\|\sum_{i=0}^{k} w_{k,i}\xi^i\big\|_2^2 = \sum_{i=0}^{k} w_{k,i}^2\,\mathbb{E}\|\xi^i\|_2^2$. Using $\|\cdot\|_\star \leq \rho\|\cdot\|_2$ and Jensen,

$$\mathbb{E}\|V_k\|_\star \leq \rho\sqrt{\sum_{i=0}^{k} w_{k,i}^2\,\mathbb{E}\|g^i - \nabla f(x^i)\|_2^2}. \tag{44}$$

*ABC in Euclidean norm.* Assume the ABC condition in $\|\cdot\|_2$: for all $x$,

$$\mathbb{E}\big[\|g(x)\|_2^2\big] \;\leq\; 2A\big(f(x) - f_{\inf}\big) \;+\; B\,\|\nabla f(x)\|_2^2 \;+\; C. \tag{45}$$

Then, using $\mathbb{E}\|g - \nabla f\|_2^2 = \mathbb{E}\|g\|_2^2 - \mathbb{E}\|\nabla f\|_2^2$ and $(B-1) \leq (B-1)_+$,

$$\mathbb{E}\|g^i - \nabla f(x^i)\|_2^2 \leq 2A\,\Delta^i + (B-1)_+\,\mathbb{E}\|\nabla f(x^i)\|_2^2 + C.$$

Combining with (42), we obtain

$$\mathbb{E}\|g^i - \nabla f(x^i)\|_2^2 \leq 2\Big(A + L\bar{\kappa}^2(B-1)_+\Big)\Delta^i + C \;=\; 2A_{\mathrm{eff}}\,\Delta^i + C, \qquad A_{\mathrm{eff}} := A + L\bar{\kappa}^2(B-1)_+.$$

Substituting into (44) yields

$$\mathbb{E}\|V_k\|_\star \;\leq\; \rho\sqrt{\sum_{i=0}^{k} w_{k,i}^2\Big(2A_{\mathrm{eff}}\Delta^i + C\Big)}. \tag{46}$$

*Split of the $C$-term.* For $x, y \geq 0$, $\sqrt{x+y} \leq \sqrt{x} + \sqrt{y}$. Hence from (46),

$$\mathbb{E}\|V_k\|_\star \;\leq\; \rho\sqrt{\sum_{i=0}^{k} w_{k,i}^2 \cdot 2A_{\mathrm{eff}}\Delta^i} \;+\; \rho\sqrt{C}\,z_k. \tag{47}$$

**Averaging the Lag Over $k = 0, \ldots, K-1$.** By (40) and the triangle inequality,

$$\frac{1}{K}\sum_{k=0}^{K-1}\mathbb{E}\|e^k\|_\star \;\leq\; \underbrace{\frac{1}{K}\sum_{k=0}^{K-1}(1-\alpha)^{k+1}\,\mathbb{E}\|e^{-1}\|_\star}_{\leq\,\frac{1}{\alpha K}} \;+\; \underbrace{\frac{1}{K}\sum_{k=0}^{K-1}\mathbb{E}\|D_k\|_\star}_{\leq\,L\gamma\frac{1-\alpha}{\alpha}} \;+\; \frac{1}{K}\sum_{k=0}^{K-1}\mathbb{E}\|V_k\|_\star.$$

We now bound the average of the two components in (47).

*(i) The ABC-coupled part (the $A_{\mathrm{eff}}$ term).* Let $u_i := 2A_{\mathrm{eff}}\Delta^i \geq 0$ and $t_k := \sum_{i=0}^{k} w_{k,i}^2 u_i$. By Cauchy–Schwarz in $k$,

$$\sum_{k=0}^{K-1}\sqrt{t_k} \;\leq\; \sqrt{K}\sqrt{\sum_{k=0}^{K-1} t_k} \;=\; \sqrt{K}\sqrt{\sum_{i=0}^{K-1} u_i \sum_{k=i}^{K-1} w_{k,i}^2}.$$

Since $\sum_{k=i}^{\infty} w_{k,i}^2 = \alpha^2 \sum_{j=0}^{\infty}(1-\alpha)^{2j} = \dfrac{\alpha}{2-\alpha}$, we have $\sum_{k=i}^{K-1} w_{k,i}^2 \le \dfrac{\alpha}{2-\alpha}$, hence

$$\frac{1}{K} \sum_{k=0}^{K-1} \rho \sqrt{\sum_{i=0}^{k} w_{k,i}^2 u_i} \;\le\; \frac{\rho}{\sqrt{K}} \sqrt{\frac{\alpha}{2-\alpha} \sum_{i=0}^{K-1} 2A_{\mathrm{eff}} \Delta^i}. \tag{48}$$

*(ii) The constant part (the C-term).* Using $z_k \le \sqrt{\alpha/(2-\alpha)}$ for all $k$,

$$\frac{1}{K} \sum_{k=0}^{K-1} \rho \sqrt{C}\, z_k \;\le\; \rho \sqrt{C} \sqrt{\frac{\alpha}{2-\alpha}}. \tag{49}$$

**Lag Average Bound.** Collecting the pieces, we obtain

$$\frac{1}{K} \sum_{k=0}^{K-1} \mathbb{E}\|e^k\|_\star \;\le\; L\gamma \frac{1-\alpha}{\alpha} \;+\; \frac{1}{\alpha K} \mathbb{E}\|e^{-1}\|_\star \;+\; \frac{\rho}{\sqrt{K}} \sqrt{\frac{\alpha}{2-\alpha} \cdot 2A_{\mathrm{eff}} \sum_{i=0}^{K-1} \Delta^i} \;+\; \rho \sqrt{C} \sqrt{\frac{\alpha}{2-\alpha}}. \tag{50}$$

**Master Inequality (Constant $\gamma$).** Substituting (50) into (38) yields

$$\frac{1}{K} \sum_{k=0}^{K-1} \mathbb{E}\|\nabla f(x^{k+1})\|_\star \le \frac{\Delta^0}{\gamma K} \;+\; \frac{7}{2} L\gamma \;+\; 2L\gamma \frac{1-\alpha}{\alpha} \;+\; \frac{2}{\alpha K} \mathbb{E}\|e^{-1}\|_\star \tag{51}$$

$$+\; \frac{2\rho}{\sqrt{K}} \sqrt{\frac{\alpha}{2-\alpha} \cdot 2A_{\mathrm{eff}} \sum_{i=0}^{K-1} \Delta^i} \;+\; 2\rho \sqrt{C} \sqrt{\frac{\alpha}{2-\alpha}}.$$

**Closing the Loop: Bounding the History $S := \sum_{i=0}^{K-1} \Delta^i$.**

**Step 1 (Crude Running Upper Bound on the Gaps).** From (36), dropping the negative term $-\gamma\|\nabla f(x^{k+1})\|_\star$ gives, for constant $\gamma_k \equiv \gamma$,

$$\Delta^{k+1} \;\le\; \Delta^k \;+\; 2\gamma\, \mathbb{E}\|e^k\|_\star \;+\; \tfrac{7L}{2}\gamma^2.$$

Summing over $k = 0, \dots, i-1$ and then over $i = 0, \dots, K-1$ yields

$$\sum_{i=0}^{K-1} \Delta^i \;\le\; K\Delta^0 \;+\; 2\gamma K \sum_{k=0}^{K-1} \mathbb{E}\|e^k\|_\star \;+\; \frac{7L}{4} K^2 \gamma^2. \tag{52}$$

**Step 2 (Substitute the Lag Bound into (52)).** Multiply (50) by $K$ and substitute into (52). Writing $S := \sum_{i=0}^{K-1} \Delta^i$, we obtain

$$S \;\le\; K\Delta^0 \;+\; 2\gamma K \left[ K L\gamma \frac{1-\alpha}{\alpha} \;+\; \frac{1}{\alpha} \mathbb{E}\|e^{-1}\|_\star \;+\; \rho \sqrt{\frac{\alpha}{2-\alpha}} \sqrt{K} \sqrt{2A_{\mathrm{eff}} S} \;+\; K\rho \sqrt{C} \sqrt{\frac{\alpha}{2-\alpha}} \right] \;+\; \frac{7L}{4} K^2 \gamma^2. \tag{53}$$

**Step 3 (Quadratic Inequality in $\sqrt{S/K}$).** Let

$$U := \frac{S}{K}, \qquad M_\alpha := \sqrt{\tfrac{\alpha}{2-\alpha}}, \qquad Q := \rho \sqrt{2A_{\mathrm{eff}}}.$$

Divide (53) by $K$ and use $S = KU$ to get

$$U \;\le\; D + E\sqrt{U}, \qquad \text{where} \quad D := \Delta^0 + \left(2L\frac{1-\alpha}{\alpha} + \frac{7L}{4}\right) K\gamma^2 + \frac{2\gamma}{\alpha} \mathbb{E}\|e^{-1}\|_\star + 2\gamma K \rho \sqrt{C}\, M_\alpha, \quad E := 2\gamma\, Q\, M_\alpha\, K. \tag{54}$$

Since $U \ge 0$, setting $y := \sqrt{U}$ turns (54) into $y^2 - Ey - D \le 0$. Hence

$$\sqrt{\frac{S}{K}} = \sqrt{U} \;\le\; \frac{E + \sqrt{E^2 + 4D}}{2} \;\le\; E + \sqrt{D}. \tag{55}$$

**Step 4 (Closed Master Bound).**   The ABC contribution in (51) equals

$$\frac{2\rho}{\sqrt{K}}\sqrt{M_\alpha^2 \cdot 2A_{\text{eff}} S} \;=\; 2Q\,M_\alpha\,\sqrt{\frac{S}{K}}.$$

Using (55) yields

$$2Q\,M_\alpha\,\sqrt{\frac{S}{K}} \;\leq\; 2Q\,M_\alpha\,(E+\sqrt{D}) \;=\; 4\gamma\,Q^2\,M_\alpha^2\,K \;+\; 2Q\,M_\alpha\,\sqrt{D}.$$

Therefore, combining terms in (51),

$$\frac{1}{K}\sum_{k=0}^{K-1}\mathbb{E}\,\|\nabla f(x^{k+1})\|_\star \;\leq\; \underbrace{\frac{\Delta^0}{\gamma K} \;+\; \frac{7}{2}\,L\,\gamma \;+\; 2L\,\gamma\,\frac{1-\alpha}{\alpha} \;+\; \frac{2}{\alpha K}\,\mathbb{E}\|e^{-1}\|_\star}_{=:\,\mathcal{T}_0} \;+\; 4\gamma\,Q^2\,M_\alpha^2\,K + 2Q\,M_\alpha\,\sqrt{D} + \underbrace{2\rho\,\sqrt{C}\,M_\alpha}_{=:\,\mathcal{T}_C},$$

$$(56)$$

with $Q = \rho\sqrt{2A_{\text{eff}}}$, $M_\alpha = \sqrt{\alpha/(2-\alpha)}$, and $D$ as in (54).

**Step 5 (Horizon-Tuned Constant Parameters and an Explicit $K^{-1/4}$ Rate).**   Choose constant-across-iterations but $K$-dependent hyperparameters

$$\alpha = \frac{\alpha_0}{\sqrt{K}}, \qquad \gamma = \frac{\gamma_0}{K^{3/4}}, \qquad \text{with} \quad \alpha_0 \in (0, \sqrt{K}], \; \gamma_0 > 0.$$

We use the safe bounds valid for all $\alpha \in (0, 1]$:

$$M_\alpha = \sqrt{\frac{\alpha}{2-\alpha}} \leq \sqrt{\alpha}, \qquad M_\alpha^2 \leq \alpha, \qquad \frac{1-\alpha}{\alpha} \leq \frac{1}{\alpha}.$$

With $\alpha = \alpha_0/\sqrt{K}$ and $\gamma = \gamma_0/K^{3/4}$,

$$\sqrt{\alpha} = \sqrt{\alpha_0}\,K^{-1/4}, \qquad \gamma K\alpha = \gamma_0\alpha_0\,K^{-1/4}, \qquad \frac{1}{\alpha K} = \frac{1}{\alpha_0\sqrt{K}}, \qquad \gamma\cdot\frac{1}{\alpha} = \frac{\gamma_0}{\alpha_0}\,K^{-1/4}.$$

Apply these bounds in (56):

- $\mathcal{T}_0 \leq \dfrac{\Delta^0}{\gamma_0}K^{-1/4} + \dfrac{2L\gamma_0}{\alpha_0}K^{-1/4} + \dfrac{2}{\alpha_0}\,\mathbb{E}\|e^{-1}\|_\star\,K^{-1/2} + \dfrac{7}{2}L\gamma_0\,K^{-3/4}.$

- $4\gamma Q^2 M_\alpha^2 K \leq 4\gamma Q^2\alpha K = 4Q^2\gamma_0\alpha_0\,K^{-1/4} = 8\rho^2 A_{\text{eff}}\gamma_0\alpha_0\,K^{-1/4}.$

- $\mathcal{T}_C = 2\rho\sqrt{C}M_\alpha \leq 2\rho\sqrt{C\alpha} = 2\rho\sqrt{C\alpha_0}\,K^{-1/4}.$

It remains to bound the $\sqrt{D}$ term. With the same safe bounds and the above schedule, the quantity $D$ in (54) satisfies

$$D \;\leq\; D_0 \;+\; \frac{2\gamma_0}{\alpha_0}\,\mathbb{E}\|e^{-1}\|_\star\,K^{-1/4} \;+\; \frac{7L}{4}\gamma_0^2\,K^{-1/2}, \qquad D_0 := \Delta^0 + \frac{2L\gamma_0^2}{\alpha_0} + 2\rho\gamma_0\sqrt{\alpha_0 C}.$$

Since $D_0 > 0$ and $x \mapsto \sqrt{x}$ is concave, for any $y \geq 0$, $\sqrt{D_0+y} \leq \sqrt{D_0} + \dfrac{y}{2\sqrt{D_0}}$. Applying this with $y = \frac{2\gamma_0}{\alpha_0}\mathbb{E}\|e^{-1}\|_\star K^{-1/4} + \frac{7L}{4}\gamma_0^2 K^{-1/2}$, and using $2QM_\alpha \leq 2Q\sqrt{\alpha} = 2\rho\sqrt{2A_{\text{eff}}\alpha_0}\,K^{-1/4}$, we obtain

$$2QM_\alpha\sqrt{D} \;\leq\; 2\rho\sqrt{2A_{\text{eff}}\alpha_0}\,\sqrt{D_0}\,K^{-1/4} \;+\; \frac{2\gamma_0 Q}{\sqrt{\alpha_0}\sqrt{D_0}}\,\mathbb{E}\|e^{-1}\|_\star\,K^{-1/2} \;+\; \frac{7L}{4}\frac{Q\sqrt{\alpha_0}}{\sqrt{D_0}}\gamma_0^2\,K^{-3/4}.$$

Collecting all terms yields the explicit bound

$$\frac{1}{K}\sum_{k=0}^{K-1}\mathbb{E}\,\|\nabla f(x^{k+1})\|_\star \leq \frac{1}{K^{1/4}}\left[\frac{\Delta^0}{\gamma_0} + \frac{2L\gamma_0}{\alpha_0} + 8\rho^2 A_{\text{eff}}\gamma_0\alpha_0 + 2\rho\sqrt{C\alpha_0} + 2\rho\sqrt{2A_{\text{eff}}\alpha_0}\,\sqrt{D_0}\right]$$

$$+ \frac{1}{K^{1/2}}\left[\frac{2}{\alpha_0}\mathbb{E}\|e^{-1}\|_\star + \frac{2\gamma_0 Q}{\sqrt{\alpha_0}\sqrt{D_0}}\mathbb{E}\|e^{-1}\|_\star\right] + \frac{1}{K^{3/4}}\left[\frac{7}{2}L\gamma_0 + \frac{7L}{4}\frac{Q\sqrt{\alpha_0}}{\sqrt{D_0}}\gamma_0^2\right], \quad (57)$$

where

$$A_{\text{eff}} := A + L\bar\kappa^2(B-1)_+, \qquad Q := \rho\sqrt{2A_{\text{eff}}}, \qquad D_0 := \Delta^0 + \frac{2L\gamma_0^2}{\alpha_0} + 2\rho\gamma_0\sqrt{\alpha_0 C}.$$

In particular, the right-hand side is $\mathcal{O}(K^{-1/4})$, and therefore

$$\frac{1}{K}\sum_{k=0}^{K-1}\mathbb{E}\,\|\nabla f(x^{k+1})\|_\star \;\xrightarrow[K\to\infty]{}\; 0,$$

with the $C$-term decaying as $K^{-1/4}$ under this horizon-tuned choice $\alpha = \alpha_0/\sqrt{K}$ (i.e., no nonvanishing noise floor).

**Remarks.**

- The rate $\mathcal{O}(K^{-1/4})$ is the canonical envelope for normalized methods with momentum in the stochastic regime; cf. the discussion in related analyses of LMO/normalized updates.

- The role of $B$ is fully captured by $A_{\text{eff}} = A + L\bar\kappa^2(B-1)_+$: when $B \le 1$, the coupling to the true gradient norm disappears from the ABC condition and $A_{\text{eff}} = A$.

- The term $2L\gamma\frac{1-\alpha}{\alpha}$ in $\mathcal{T}_0$ is the *drift due to momentum memory*. It stays controlled under the $K^{-3/4}$–$K^{-1/2}$ horizon tuning: $2L\gamma\frac{1-\alpha}{\alpha} = \mathcal{O}(K^{-1/4})$.

- If one wishes, the leading constant in (57) can be reduced by minimizing the bracket w.r.t. $g > 0$ for a fixed $a > 0$ (a one–dimensional convex tradeoff between $\Delta^0/g$ and the terms linear in $g$).

**Theorem A.1** (LMO-based method with momentum under ABC). *Assume $f$ is $L$-smooth and bounded below by $f_{\inf}$, the stochastic oracle is unbiased, and the ABC condition holds with constants $(A, B, C)$. Let the algorithm use constant parameters $\gamma > 0$ and $\alpha \in (0,1)$, and define the one–step momentum lag $e^{-1} := m^0 - \nabla f(x^0)$. Set*

$$A_{\text{eff}} := A + L\bar\kappa^2(B-1)_+, \qquad M_\alpha := \sqrt{\frac{\alpha}{2-\alpha}}, \qquad Q := \rho\sqrt{2A_{\text{eff}}},$$

*where $\rho \ge 1$ is any norm–compatibility constant such that $\|v\|_\star \le \rho\|v\|_2$. Then for any $K \ge 1$ the iterates of the LMO-based method with momentum satisfy*

$$\frac{1}{K}\sum_{k=0}^{K-1}\mathbb{E}\,\|\nabla f(x^{k+1})\|_\star \;\le\; \underbrace{\frac{\Delta^0}{\gamma K} + \frac{7}{2}L\gamma + 2L\gamma\frac{1-\alpha}{\alpha} + \frac{2}{\alpha K}\mathbb{E}\|e^{-1}\|_\star}_{=:\,\mathcal{T}_0} + \underbrace{4\gamma Q^2 M_\alpha^2 K + 2Q M_\alpha\sqrt{D}}_{=:\,\mathcal{T}_{\text{ABC}}} + \underbrace{2\rho\sqrt{C}\,M_\alpha}_{=:\,\mathcal{T}_C},$$

(58)

*where*

$$D := \Delta^0 + \left(2L\frac{1-\alpha}{\alpha} + \frac{7L}{4}\right)K\gamma^2 + \frac{2\gamma}{\alpha}\mathbb{E}\|e^{-1}\|_\star + 2\gamma K\rho\sqrt{C}\,M_\alpha.$$

Interpretation. $\mathcal{T}_0$ contains deterministic terms (initial gap and smoothness budget) and a memory–drift penalty $2L\gamma\frac{1-\alpha}{\alpha}$. $\mathcal{T}_{\text{ABC}}$ is the ABC-coupled contribution obtained via a self-bounding closure; $\mathcal{T}_C$ is the pure-variance floor coming from the $C$–term of ABC (it disappears as $\alpha \to 0$ with the usual horizon-tuned choice).

**Corollary A.2** (Horizon-tuned constant hyperparameters; explicit $K^{-1/4}$ rate). *Fix $a > 0$ and $g > 0$, and choose constant-across-iterations (but $K$-dependent) hyperparameters*

$$\alpha = \frac{\alpha_0}{\sqrt{K}}, \qquad \gamma = \frac{\gamma_0}{K^{3/4}}, \qquad (\text{assume } K \ge \alpha_0^2 \text{ so that } \alpha \in (0,1]).$$

*Then Theorem A.1 yields*

$$\frac{1}{K}\sum_{k=0}^{K-1}\mathbb{E}\big\|\nabla f(x^{k+1})\big\|_\star \le \frac{1}{K^{1/4}}\left[\frac{\Delta^0}{\gamma_0} + \frac{2L\gamma_0}{\alpha_0} + 4\alpha_0\gamma_0\,\rho^2 A_{\text{eff}} + 2\rho\sqrt{\frac{\alpha_0}{2}C} + 2\rho\sqrt{2A_{\text{eff}}}\,\sqrt{\widetilde{D_0}}\,\sqrt{\frac{\alpha_0}{2}}\right]$$
$$+ \frac{2}{\alpha_0 K^{1/2}}\,\mathbb{E}\|e^{-1}\|_\star + \frac{7}{2}L\gamma_0\,K^{-3/4},$$

(59)

where $\widetilde{D_0} := \Delta^0 + \frac{2L\gamma_0^2}{\alpha_0} + 2\gamma_0\rho\sqrt{\frac{\alpha_0}{2}C}$. In particular, $\frac{1}{K}\sum_{k=0}^{K-1}\mathbb{E}\|\nabla f(x^{k+1})\|_\star = \mathcal{O}(K^{-1/4})$ with an explicit leading constant depending only on $(L, \rho, A_{\text{eff}}, C, \Delta^0, \alpha_0, \gamma_0)$. When $C > 0$ the contribution of the $C$-term scales as $K^{-1/4}$ (no noise floor under this horizon-tuned schedule), while for a fixed $\alpha$ independent of $K$ it becomes a $K$-independent floor $2\rho\sqrt{C}\,M_\alpha$.

## A.4. Optimizing Hyperparameters

We derive (approximately) optimal constant hyperparameters by minimizing the *leading* term in the horizon-tuned bound. Concretely, we use the standard horizon tuning

$$\alpha = \frac{\alpha_0}{\sqrt{K}}, \qquad \gamma = \frac{\gamma_0}{K^{3/4}}, \qquad \alpha_0 > 0, \; \gamma_0 > 0, \tag{60}$$

and minimize the coefficient multiplying $K^{-1/4}$ in the resulting bound (lower-order terms $K^{-1/2}$ and $K^{-3/4}$ affect constants but not the asymptotic scaling in $K$).

**BV Specialization of the Horizon-Tuned Bound.** Assume a bounded-variance oracle in Euclidean norm:

$$\mathbb{E}\|g_b(x) - \nabla f(x)\|_2^2 \leq \frac{\sigma^2}{b}, \tag{61}$$

for some $\sigma^2 > 0$ and (optional) mini-batch size $b \geq 1$. This corresponds to the ABC condition with

$$A_{\text{eff}} = 0, \qquad C = \frac{\sigma^2}{b},$$

so all ABC-coupled terms vanish ($Q = 0$). Keeping only the dominant $K^{-1/4}$ contribution in the horizon-tuned bound yields the optimization proxy

$$\frac{1}{K}\sum_{k=0}^{K-1}\mathbb{E}\|\nabla f(x^{k+1})\|_\star \lesssim \frac{1}{K^{1/4}}\,\Phi_{\text{BV}}(\alpha_0, \gamma_0), \qquad \Phi_{\text{BV}}(\alpha_0, \gamma_0) := \frac{\Delta_0}{\gamma_0} + \frac{2L\gamma_0}{\alpha_0} + 2\rho\sqrt{\frac{\sigma^2}{b}}\,\alpha_0, \tag{62}$$

where $\Delta_0 := f(x^0) - f_{\inf}$ and $\rho$ is the norm-compatibility constant used to compare $\|\cdot\|_\star$ and $\|\cdot\|_2$.

**Step 1: Optimize in $\gamma_0$ for Fixed $\alpha_0$.** For fixed $\alpha_0 > 0$, the function $\gamma_0 \mapsto \Delta_0/\gamma_0 + (2L/\alpha_0)\gamma_0$ is minimized at

$$\gamma_{0,\text{BV}}^\star(\alpha_0) = \sqrt{\frac{\Delta_0\alpha_0}{2L}}. \tag{63}$$

Substituting (63) into (62) gives a one-dimensional objective in $\alpha_0$:

$$\Phi_{\text{BV}}(\alpha_0, \gamma_{0,\text{BV}}^\star(\alpha_0)) = 2\sqrt{\frac{2L\Delta_0}{\alpha_0}} + 2\rho\sqrt{\frac{\sigma^2}{b}}\,\sqrt{\alpha_0}. \tag{64}$$

**Step 2: Optimize in $\alpha_0$.** The right-hand side of (64) has the standard form $p/\sqrt{\alpha_0} + q\sqrt{\alpha_0}$ with

$$p := 2\sqrt{2L\Delta_0}, \qquad q := 2\rho\sqrt{\frac{\sigma^2}{b}}.$$

Its minimizer satisfies $p/\sqrt{\alpha_0} = q\sqrt{\alpha_0}$, hence

$$\alpha_{0,\text{BV}}^\star = \frac{p}{q} = \frac{\sqrt{2L\Delta_0}}{\rho} \cdot \frac{\sqrt{b}}{\sigma}. \tag{65}$$

**Resulting Optimal BV Parameters.** Combining (60), (63), and (65) yields

$$\boxed{\alpha_{\mathrm{BV}}^{\star} = \frac{\alpha_{0,\mathrm{BV}}^{\star}}{\sqrt{K}} = \frac{\sqrt{2bL\Delta_0/K}}{\rho\sigma}, \qquad \gamma_{\mathrm{BV}}^{\star} = \frac{\gamma_{0,\mathrm{BV}}^{\star}(\alpha_{0,\mathrm{BV}}^{\star})}{K^{3/4}} = \frac{1}{2^{1/4}}\frac{\Delta_0^{3/4}\,b^{1/4}}{L^{1/4}\,(\rho\sigma)^{1/2}}\frac{1}{K^{3/4}}.} \tag{66}$$

Thus, the optimized bound (up to universal constants) becomes

$$\frac{1}{K}\sum_{k=0}^{K-1}\mathbb{E}\|\nabla f(x^{k+1})\|_{\star} \;\lesssim\; \frac{(L\Delta_0)^{1/4}\,(\rho^2\sigma^2/b)^{1/4}}{K^{1/4}}. \tag{67}$$

**Comparison to Prior Work.** In (Shulgin et al., 2026), Corollary 3 gives (for the stochastic setting) the parameter choices

$$\gamma^{\star} = \left(\frac{\Delta_0}{K}\right)^{3/4}\frac{1}{(\sigma^2 L(1+\delta))^{1/4}}, \qquad \alpha^{\star} = \sqrt{\frac{\Delta_0 L(1+\delta)}{K\sigma^2}},$$

where $\delta$ quantifies LMO inexactness. In the *exact* case $\delta = 0$ and after accounting for (i) the norm-compatibility factor $\rho$ in our analysis and (ii) mini-batching via $\sigma^2 \mapsto \sigma^2/b$, the dependence on the core quantities $(L, \sigma, K, \Delta_0, b)$ in (66) matches (Shulgin et al., 2026) up to universal constant factors and notational differences.

**A.5. $\tau$-Nice Mini-Batching (Finite-Sum ABC): Optimal $(\alpha, \gamma)$ and Effect of $(A_b, B_b, C_b)$**

**ABC Constants for $\tau$-Nice Sampling.** Under $\tau$-nice sampling without replacement with batch size $b$ in the finite-sum model, the ABC constants can be taken as

$$A_b = \frac{n-b}{b(n-1)}L_{\max}, \qquad B_b = \frac{n(b-1)}{b(n-1)} \le 1, \qquad C_b = 2A_b\Delta_{\inf}.$$

Since $B_b \le 1$, we have $(B_b - 1)_+ = 0$ and hence $A_{\mathrm{eff}} = A_b$ in our master bound.

Define the associated ABC scale

$$Q_b := \rho\sqrt{2A_b}, \qquad \text{so that} \qquad \rho\sqrt{C_b} = \rho\sqrt{2A_b\Delta_{\inf}} = Q_b\sqrt{\Delta_{\inf}}. \tag{68}$$

**Dominant Horizon-Tuned Objective (Scaling Proxy).** Plugging $A_{\mathrm{eff}} = A_b$ and $C = C_b$ into the leading $K^{-1/4}$ coefficient (and suppressing only universal constants) yields a proxy objective of the form

$$\frac{1}{K}\sum_{k=0}^{K-1}\mathbb{E}\|\nabla f(x^{k+1})\|_{\star} \;\lesssim\; \frac{1}{K^{1/4}}\,\Phi_{\tau}(\alpha_0,\gamma_0), \qquad \Phi_{\tau}(\alpha_0,\gamma_0) \approx \frac{\Delta_0}{\gamma_0} + \gamma_0\left(\frac{2L}{\alpha_0} + c_1\rho^2 A_b\,\alpha_0\right) + c_2\,\rho\sqrt{A_b}\,(\sqrt{\Delta_0} + \sqrt{\Delta_{\inf}})\,\sqrt{\alpha_0}, \tag{69}$$

where $c_1, c_2 > 0$ are universal constants. The last term aggregates the pure-noise contribution ($\propto \rho\sqrt{C_b\alpha_0}$) and the leading part of the ABC "square-root" term ($\propto Q_b\sqrt{\alpha_0}\sqrt{\Delta_0}$); both scale as $\rho\sqrt{A_b}\sqrt{\alpha_0}$, and the dependence on $\Delta_0$ and $\Delta_{\inf}$ appears only through $\sqrt{\Delta_0}$ and $\sqrt{\Delta_{\inf}}$ thanks to (68).

**Step 1: Optimize in $\gamma_0$ for Fixed $\alpha_0$.** Ignoring the (additive) $\sqrt{\alpha_0}$ term (which does not depend on $\gamma_0$), the minimizer of $\Delta_0/\gamma_0 + \gamma_0(\frac{2L}{\alpha_0} + c_1\rho^2 A_b\alpha_0)$ is

$$\gamma_{0,\tau}^{\star}(\alpha_0) = \sqrt{\frac{\Delta_0}{\frac{2L}{\alpha_0} + c_1\rho^2 A_b\alpha_0}}. \tag{70}$$

Substituting (70) into (69) shows that the dependence on $A_b$ enters through the "effective" coefficient

$$\frac{2L}{\alpha_0} + c_1\rho^2 A_b\alpha_0,$$

which is minimized (in $\alpha_0$) by balancing the memory and ABC-growth contributions.

**Step 2: ABC-Balanced Choice of $\alpha_0$.** Balancing $\frac{2L}{\alpha_0}$ and $c_1\rho^2 A_b\alpha_0$ yields the canonical ABC-balanced scaling

$$\alpha_{0,\tau}^{\mathrm{bal}} \asymp \sqrt{\frac{L}{\rho^2 A_b}}. \tag{71}$$

With (71), (70) gives

$$\gamma_{0,\tau}^{\mathrm{bal}} \asymp \frac{\sqrt{\Delta_0}}{\rho^{1/2}(LA_b)^{1/4}}. \tag{72}$$

Plugging (71)–(72) into the $\sqrt{\alpha_0}$-term in (69) shows it scales as

$$\rho\sqrt{A_b}\sqrt{\alpha_{0,\tau}^{\mathrm{bal}}} \asymp \rho\sqrt{A_b}\left(\frac{L}{\rho^2 A_b}\right)^{1/4} = \rho^{1/2}(LA_b)^{1/4},$$

i.e., it matches the same $(LA_b)^{1/4}$ scaling as the optimized "linear" part. Hence (71)–(72) is stable: the $\sqrt{\alpha_0}$ terms affect constants but not the scaling with $A_b$.

**Optional Refinement When $\Delta_{\mathrm{inf}} \gg \Delta_0$.** If $\Delta_{\mathrm{inf}}$ dominates $\Delta_0$, the $\sqrt{\alpha_0}$ penalty becomes relatively more important and it can be beneficial (in the bound) to reduce $\alpha_0$ by the factor

$$r := \frac{\sqrt{\Delta_0}}{\sqrt{\Delta_0} + \sqrt{\Delta_{\mathrm{inf}}}} \in (0,1].$$

A convenient refinement is $\alpha_{0,\tau}^{\mathrm{ref}} \asymp \alpha_{0,\tau}^{\mathrm{bal}} r$, which preserves the same $A_b^{-1/2}$ scaling while accounting for heterogeneity.

**Resulting $\tau$-Nice Parameter Scalings.** Using (60) and (71)–(72) gives the canonical $\tau$-nice tuning

$$\boxed{\alpha_\tau^\star \asymp \min\left\{1, \frac{1}{\sqrt{K}}\sqrt{\frac{L}{\rho^2 A_b}}\right\}, \qquad \gamma_\tau^\star \asymp \frac{1}{K^{3/4}}\cdot\frac{\sqrt{\Delta_0}}{\rho^{1/2}(LA_b)^{1/4}}.} \tag{73}$$

Substituting $A_b = \frac{n-b}{b(n-1)}L_{\max}$ makes the batch-size effect explicit:

$$\boxed{\alpha_\tau^\star(b) \asymp \min\left\{1, \frac{1}{\sqrt{K}}\sqrt{\frac{L}{\rho^2 L_{\max}}}\sqrt{\frac{b(n-1)}{n-b}}\right\}, \qquad \gamma_\tau^\star(b) \asymp \frac{1}{K^{3/4}}\frac{\sqrt{\Delta_0}}{\rho^{1/2}(LL_{\max})^{1/4}}\left(\frac{b(n-1)}{n-b}\right)^{1/4}.} \tag{74}$$

**Comparison: BV vs. $\tau$-Nice.** The BV optimum (66) has the form

$$\alpha_{\mathrm{BV}}^\star \asymp \frac{1}{\sqrt{K}}\frac{\sqrt{L\Delta_0}}{\rho}\sqrt{\frac{b}{\sigma^2}}, \qquad \gamma_{\mathrm{BV}}^\star \asymp \frac{1}{K^{3/4}}\frac{\Delta_0^{3/4}}{\rho^{1/2}L^{1/4}}\left(\frac{b}{\sigma^2}\right)^{1/4}.$$

In contrast, $\tau$-nice sampling induces an ABC constant $A_b$ (and hence $Q_b = \rho\sqrt{2A_b}$), and the optimized $\tau$-nice parameters (74) depend on $b$ through the finite-population correction $\frac{n-b}{b(n-1)}$. In the common regime $b \ll n$ this correction satisfies $\frac{n-b}{n-1} \approx 1$, and the $\tau$-nice dependence on $b$ reduces to the familiar BV scaling $\alpha^\star \propto \sqrt{b}$ and $\gamma^\star \propto b^{1/4}$, but with $\sigma$ effectively replaced by the ABC-induced scale (through $A_b$ and $\Delta_{\mathrm{inf}}$).

## A.6. Iteration Complexity Analysis

Fix a target accuracy $\varepsilon \in (0,1]$, and let

$$\Delta_0 := f(x^0) - f_{\mathrm{inf}}, \qquad E_0 := \mathbb{E}\|e^{-1}\|_\star = \mathbb{E}\|m^0 - \nabla f(x^0)\|_\star.$$

Throughout we start from the (closed) master bound of Theorem A.1:

$$\frac{1}{K}\sum_{k=0}^{K-1}\mathbb{E}\|\nabla f(x^{k+1})\|_\star \leq \underbrace{\frac{\Delta_0}{\gamma K} + \frac{2E_0}{\alpha K}}_{\texttt{init+burn-in}} + \underbrace{L\gamma\left(\frac{7}{2} + 2\frac{1-\alpha}{\alpha}\right)}_{\texttt{smooth+memory}} + \underbrace{2\rho\sqrt{C}\,M_\alpha}_{\texttt{noise}} + \underbrace{4\gamma Q^2 M_\alpha^2 K + 2Q\,M_\alpha\sqrt{D}}_{\texttt{ABC-coupled}}, \tag{75}$$

where

$$M_\alpha := \sqrt{\frac{\alpha}{2-\alpha}}, \qquad Q := \rho\sqrt{2A_{\text{eff}}}, \qquad A_{\text{eff}} := A + L\bar{\kappa}^2(B-1)_+,$$

and

$$D := \Delta_0 + \left(2L\frac{1-\alpha}{\alpha} + \frac{7L}{4}\right)K\gamma^2 + \frac{2\gamma}{\alpha}E_0 + 2\gamma K\rho\sqrt{C}\,M_\alpha.$$

**A convenient relaxation (removing the $C$-term from inside $\sqrt{D}$ when $Q > 0$).** Assume $Q > 0$ (equivalently $A_{\text{eff}} > 0$). Write

$$D = \widetilde{D} + D_C, \qquad \widetilde{D} := \Delta_0 + \left(2L\frac{1-\alpha}{\alpha} + \frac{7L}{4}\right)K\gamma^2 + \frac{2\gamma}{\alpha}E_0, \qquad D_C := 2\gamma K\rho\sqrt{C}\,M_\alpha.$$

Then by $\sqrt{x+y} \le \sqrt{x} + \sqrt{y}$,

$$2QM_\alpha\sqrt{D} \le 2QM_\alpha\sqrt{\widetilde{D}} + 2QM_\alpha\sqrt{D_C}.$$

Moreover,

$$2QM_\alpha\sqrt{D_C} = 2QM_\alpha\sqrt{2\gamma K\rho\sqrt{C}\,M_\alpha} = \sqrt{\left(4\gamma Q^2 M_\alpha^2 K\right)\left(2\rho\sqrt{C}\,M_\alpha\right)} \le \frac{1}{2}\left(4\gamma Q^2 M_\alpha^2 K + 2\rho\sqrt{C}\,M_\alpha\right),$$

where the last step is AM–GM. Substituting into (75) yields the relaxed bound

$$\frac{1}{K}\sum_{k=0}^{K-1}\mathbb{E}\|\nabla f(x^{k+1})\|_\star \le \underbrace{\frac{\Delta_0}{\gamma K}}_{\text{init}} + \underbrace{\frac{2E_0}{\alpha K}}_{\text{burn-in}} + \underbrace{L\gamma\left(\frac{7}{2} + 2\frac{1-\alpha}{\alpha}\right)}_{\text{smooth+memory}} + \underbrace{3\rho\sqrt{C}\,M_\alpha}_{\text{noise}} + \underbrace{6\gamma Q^2 M_\alpha^2 K}_{\text{ABC-grow}} + \underbrace{2Q\,M_\alpha\sqrt{\widetilde{D}}}_{\text{ABC-sqrt}}. \tag{76}$$

**Budget split.** We enforce each of the six groups in (76),

$$\texttt{init, burn-in, smooth+memory, noise, ABC-grow, ABC-sqrt,}$$

to be $\le \varepsilon/6$. We repeatedly use, for $\alpha \in (0,1]$,

$$M_\alpha \le \sqrt{\alpha}, \qquad M_\alpha^2 \le \alpha.$$

We also split $\widetilde{D}$ into three nonnegative pieces and use $\sqrt{x+y+z} \le \sqrt{x} + \sqrt{y} + \sqrt{z}$:

$$\widetilde{D} = \underbrace{\Delta_0}_{\text{(i)}} + \underbrace{\left(2L\frac{1-\alpha}{\alpha} + \frac{7L}{4}\right)K\gamma^2}_{\text{(ii)}} + \underbrace{\frac{2\gamma}{\alpha}E_0}_{\text{(iii)}}.$$

To ensure $\texttt{ABC-sqrt} \le \varepsilon/6$, we allocate $\varepsilon/18$ to each of the three contributions coming from (i)–(iii).

**Step 1 (Momentum).** Impose

$$\texttt{noise} \le \varepsilon/6, \qquad \texttt{ABC-sqrt's } \Delta_0\text{-piece} \le \varepsilon/18.$$

Using $M_\alpha \le \sqrt{\alpha}$, this is guaranteed by

$$\boxed{\alpha \le \min\left\{\left(\frac{\varepsilon}{18\,\rho\sqrt{C}}\right)^2, \left(\frac{\varepsilon}{36\,Q\,\sqrt{\Delta_0}}\right)^2, 1\right\}.} \tag{77}$$

**Step 2 (Stepsize).** With $\alpha$ as in (77), imposing

$$\texttt{smooth+memory, ABC-grow, ABC-sqrt's } K\gamma^2\text{-piece, ABC-sqrt's } E_0\text{-piece} \leq \varepsilon/6$$

gives the caps

$$\gamma \;\leq\; \min\left\{ \frac{\varepsilon}{6L\left(\frac{7}{2} + 2\frac{1-\alpha}{\alpha}\right)},\; \frac{\varepsilon}{36\,Q^2\alpha\,K},\; \frac{\varepsilon}{36\,Q\,\sqrt{2L}\,\sqrt{K}},\; \frac{\varepsilon^2}{2592\,Q^2\,E_0} \right\}. \tag{78}$$

*Derivation of the last two caps.* For the $K\gamma^2$-piece,

$$2QM_\alpha\sqrt{\left(2L\frac{1-\alpha}{\alpha} + \frac{7L}{4}\right)K\gamma^2} \;\leq\; 2Q\gamma\sqrt{LK}\,\sqrt{2(1-\alpha) + \frac{7}{4}\alpha} \;\leq\; 2Q\gamma\sqrt{2LK},$$

so requiring it $\leq \varepsilon/18$ gives $\gamma \leq \varepsilon/(36Q\sqrt{2L}\sqrt{K})$. For the $E_0$-piece,

$$2QM_\alpha\sqrt{\tfrac{2\gamma}{\alpha}E_0} = 2Q\sqrt{2\gamma E_0},$$

so requiring it $\leq \varepsilon/18$ gives $\gamma \leq \varepsilon^2/(2592Q^2E_0)$.

**Step 3 (Iteration Budget).** From (76), the decaying groups give

$$\texttt{init} \leq \varepsilon/6 \quad\Rightarrow\quad K \geq \frac{6\,\Delta_0}{\varepsilon\,\gamma}, \qquad \texttt{burn-in} \leq \varepsilon/6 \quad\Rightarrow\quad K \geq \frac{12\,E_0}{\alpha\,\varepsilon}. \tag{79}$$

Because $\gamma$ is itself upper-bounded by (78), determining a sufficient $K$ is a fixed-point calculation. The favorable asymptotic regime (small $\varepsilon$) is when the $K$-dependent cap $\gamma \leq \varepsilon/(36Q^2\alpha K)$ is active. In that regime,

$$\gamma K = \frac{\varepsilon}{36Q^2\alpha} \qquad\Rightarrow\qquad \texttt{init} = \frac{\Delta_0}{\gamma K} = \frac{36Q^2\alpha\Delta_0}{\varepsilon} \leq \varepsilon/6 \iff \alpha \leq \frac{\varepsilon^2}{216Q^2\Delta_0},$$

which is implied by the second condition in (77). Thus, in the active-$K$-cap regime, the only remaining decay constraint is

$$K \;\geq\; \frac{12\,E_0}{\alpha\,\varepsilon}. \tag{80}$$

To guarantee that the $K$-dependent cap is indeed the active one in (78), it suffices to ensure

$$K \;\geq\; \max\left\{ \frac{2L}{Q^2\alpha^2},\; \frac{72\,E_0}{\alpha\,\varepsilon} \right\}, \tag{81}$$

obtained by comparing $\varepsilon/(36Q^2\alpha K)$ with the other caps in (78). (The second condition ensures $\varepsilon/(36Q^2\alpha K) \leq \varepsilon^2/(2592Q^2E_0)$, and it dominates the burn-in lower bound (80) up to an absolute constant.)

**Condensed Stochastic Scales.** Define

$$S_C := \rho\sqrt{C}, \qquad S_A := Q\sqrt{\Delta_0} = \rho\sqrt{2A_{\mathrm{eff}}\Delta_0}, \qquad S := \max\{S_C, S_A\}.$$

Then (77) implies $\alpha = \Theta\big(\min\{1, \varepsilon^2/S^2\}\big)$ (up to absolute constants). Plugging this into (80)–(81) yields explicit sufficient choices.

**Theorem A.3** (Iteration complexity of normalized momentum under ABC via $\varepsilon$-splitting ($Q > 0$)). *Assume the setup of Theorem A.1 and that $Q > 0$ (equivalently $A_{\mathrm{eff}} > 0$). Fix $\varepsilon \in (0,1]$, let $S := \max\{\rho\sqrt{C}, \rho\sqrt{2A_{\mathrm{eff}}\Delta_0}\}$, and set*

$$\alpha \;=\; \min\left\{1, \frac{\varepsilon^2}{c_\alpha\,S^2}\right\}, \qquad \gamma \;=\; \min\left\{ \frac{\varepsilon}{c_1 L\left(\frac{7}{2} + 2\frac{1-\alpha}{\alpha}\right)},\; \frac{\varepsilon}{c_2 Q^2\alpha K} \right\},$$

*with absolute constants $c_\alpha, c_1, c_2 > 0$ (e.g., $c_\alpha = 1296$, $c_1 = 6$, $c_2 = 36$ are admissible for (77)–(78) up to harmless constant-factor slack). Then a sufficient iteration budget that guarantees*

$$\frac{1}{K} \sum_{k=0}^{K-1} \mathbb{E}\,\|\nabla f(x^{k+1})\|_\star \;\leq\; \varepsilon$$

*is*

$$K \;=\; \mathcal{O}\left(\underbrace{\frac{L\,\Delta_0}{\varepsilon^2}}_{\text{deterministic}} \;\vee\; \underbrace{\frac{E_0}{\varepsilon}}_{\text{linear burn-in}} \;\vee\; \underbrace{\frac{E_0\,S^2}{\varepsilon^3}}_{\text{cubic burn-in}} \;\vee\; \underbrace{\frac{L}{Q^2}\frac{S^4}{\varepsilon^4}}_{\text{ABC-coupled quartic}}\right), \tag{82}$$

*where $\vee$ denotes the maximum.*

**Remarks.**

- The first term is the classical deterministic baseline $\mathcal{O}(L\Delta_0/\varepsilon^2)$.

- The burn-in constraint is $K \gtrsim E_0/(\alpha\varepsilon)$. When $\alpha$ saturates at 1 (i.e., $S \lesssim \varepsilon$), this yields the linear term $\mathcal{O}(E_0/\varepsilon)$; when $\alpha = \Theta(\varepsilon^2/S^2)$ (the stochastic regime $S \gtrsim \varepsilon$), it yields the cubic term $\mathcal{O}(E_0 S^2/\varepsilon^3)$.

- The ABC-coupled quartic term is present only when $Q > 0$ (i.e., $A_{\text{eff}} > 0$); it enforces the active-$K$-cap regime (81) and is the analogue of the classical $\mathcal{O}(\cdot/\varepsilon^4)$ stochastic envelope for normalized methods.

- When $Q = 0$ (no history coupling), the ABC-coupled terms vanish and the stochastic quartic complexity is governed by the bounded-variance specialization below.

**Corollary A.4** (Bounded-variance specialization; recovery of Kovalev-type bounds). *Suppose $A = 0$ and $B \leq 1$ in ABC, so $A_{\text{eff}} = 0$ and hence $Q = 0$. Assume a bounded-variance noise model $\mathbb{E}\|g^k - \nabla f(x^k)\|_2^2 \leq \sigma^2$ (equivalently, in (45) one may take $C = \sigma^2$). Then with*

$$\alpha = \min\left\{1, \frac{\varepsilon^2}{c\,\rho^2\sigma^2}\right\}, \qquad \gamma = \min\left\{\frac{\varepsilon}{L}, \frac{\varepsilon^3}{c'\,\rho^2\sigma^2\,L}\right\},$$

*for universal constants $c, c' > 0$, a sufficient iteration budget is*

$$K = \mathcal{O}\left(\underbrace{\frac{L\,\Delta_0}{\varepsilon^2}}_{\text{deterministic}} \;\vee\; \underbrace{\frac{E_0}{\varepsilon}}_{\text{linear}} \;\vee\; \underbrace{\frac{E_0\,\rho^2\sigma^2}{\varepsilon^3}}_{\text{cubic}} \;\vee\; \underbrace{\frac{L\,\Delta_0\,\rho^2\sigma^2}{\varepsilon^4}}_{\text{quartic}}\right). \tag{83}$$

*In particular, for the standard initialization $m^0 = g^0$ and $\mathbb{E}\|g^0 - \nabla f(x^0)\|_2^2 \leq \sigma^2$, one has $E_0 \leq \rho\,\sigma$, and (83) implies the norm-compatible structure*

$$K \;=\; \mathcal{O}\left(\frac{L\,\Delta_0}{\varepsilon^2} \;\vee\; \frac{\rho\sigma}{\varepsilon} \;\vee\; \frac{\rho^3\sigma^3}{\varepsilon^3} \;\vee\; \frac{L\,\Delta_0\,\rho^2\sigma^2}{\varepsilon^4}\right),$$

*matching the complexity envelopes appearing in (Khaled & Richtárik, 2023)-style bounds (and Kovalev-type analyses for ideal LMO steps) when written in the norm-compatible form with $\rho$.*

### A.7. Mini-Batch $\tau$–Nice Sampling Without Replacement

We now specialize Theorem A.3 to the finite-sum setting with $\tau$–nice (without replacement) mini-batching.

Consider

$$f(x) \;=\; \frac{1}{n} \sum_{i=1}^{n} f_i(x),$$

where each $f_i$ is $L_i$–smooth under $\|\cdot\|$, and set $L_{\max} := \max_{1\le i\le n} L_i$ and $L$ the smoothness constant of $f$ (e.g. $L \le L_{\max}$). For a mini-batch of size $b \in \{1, \ldots, n\}$ drawn uniformly without replacement ($\tau$–nice sampling) and the usual stochastic gradient $g(x) = \frac{1}{b}\sum_{i\in B} \nabla f_i(x)$, the ABC condition holds in the Euclidean norm with constants

$$A(b) \;=\; \frac{n-b}{b(n-1)}\, L_{\max}, \qquad B(b) \;=\; \frac{n(b-1)}{b(n-1)}, \qquad C(b) \;=\; 2A(b)\,\Delta_{\inf}, \tag{84}$$

where $\Delta_{\inf} \ge 0$ is the usual "finite-sum bias" constant (see, e.g., the SGD analysis under ABC). Note that $B(b) \le 1$ for all $b \le n$, so $(B(b) - 1)_+ = 0$ and hence

$$A_{\mathrm{eff}}(b) \;=\; A(b).$$

Recall from Theorem A.1 and Theorem A.3 that

$$Q \;=\; \rho\sqrt{2A_{\mathrm{eff}}}, \qquad S^2 \;:=\; \max\{\rho^2 C,\; Q^2\Delta_0\},$$

and that for any target $\varepsilon \in (0,1]$ there exist choices of constant $\alpha, \gamma$ and an iteration budget $K$ such that $\frac{1}{K}\sum_{k=0}^{K-1} \mathbb{E}\|\nabla f(x^{k+1})\|_\star \le \varepsilon$ whenever

$$K \;=\; \mathcal{O}\!\left(\frac{L\Delta_0}{\varepsilon^2} \;\vee\; \frac{E_0 S^2}{\varepsilon^3} \;\vee\; \frac{LS^4}{Q^2\varepsilon^4}\right), \tag{85}$$

where $E_0 := \mathbb{E}\|e^{-1}\|_\star$ is the initial momentum lag.

**Plugging in the $\tau$–Nice ABC Constants.** For (84) we have, for every $b$,

$$A_{\mathrm{eff}}(b) = A(b), \qquad Q^2(b) = 2\rho^2 A(b), \qquad \rho^2 C(b) = 2\rho^2 A(b)\Delta_{\inf}.$$

Hence

$$S^2(b) = \max\{2\rho^2 A(b)\Delta_{\inf},\; 2\rho^2 A(b)\Delta_0\} = 2\rho^2 A(b)\,\overline{\Delta}, \qquad \overline{\Delta} := \max\{\Delta_0, \Delta_{\inf}\}.$$

Consequently,

$$\frac{E_0 S^2(b)}{\varepsilon^3} = \frac{2E_0\rho^2 A(b)\overline{\Delta}}{\varepsilon^3}, \qquad \frac{LS^4(b)}{Q^2(b)\varepsilon^4} = \frac{L \cdot 4\rho^4 A(b)^2\overline{\Delta}^2}{2\rho^2 A(b)\varepsilon^4} = \frac{2L\rho^2 A(b)\overline{\Delta}^2}{\varepsilon^4}.$$

Using $A(b)$ from (84),

$$A(b) \;=\; \frac{n-b}{b(n-1)}\, L_{\max},$$

we obtain

$$\frac{E_0 S^2(b)}{\varepsilon^3} = \mathcal{O}\!\left(\frac{E_0\rho^2 L_{\max}\overline{\Delta}}{\varepsilon^3} \cdot \frac{n-b}{b(n-1)}\right), \qquad \frac{LS^4(b)}{Q^2(b)\varepsilon^4} = \mathcal{O}\!\left(\frac{L\rho^2 L_{\max}\overline{\Delta}^2}{\varepsilon^4} \cdot \frac{n-b}{b(n-1)}\right). \tag{86}$$

**Corollary A.5** (Normalized method with momentum under $\tau$–nice mini-batching). *Assume the setting of Theorem A.1 and the finite-sum model with $\tau$–nice mini-batching of size $b \in \{1, \ldots, n\}$. Let $\Delta_0 = f(x^0) - f_{\inf}$, $\Delta_{\inf}$ be as in (84), and $\overline{\Delta} = \max\{\Delta_0, \Delta_{\inf}\}$. Then for any target accuracy $\varepsilon \in (0,1]$ there exist constant parameters $\alpha \in (0,1)$ and $\gamma > 0$ and an iteration budget $K$ such that*

$$\frac{1}{K}\sum_{k=0}^{K-1} \mathbb{E}\|\nabla f(x^{k+1})\|_\star \;\le\; \varepsilon$$

*whenever*

$$K(b,\varepsilon) \;=\; \mathcal{O}\!\left(\frac{L\Delta_0}{\varepsilon^2} \;\vee\; \frac{E_0\rho^2 L_{\max}\overline{\Delta}}{\varepsilon^3} \cdot \frac{n-b}{b(n-1)} \;\vee\; \frac{L\rho^2 L_{\max}\overline{\Delta}^2}{\varepsilon^4} \cdot \frac{n-b}{b(n-1)}\right), \tag{87}$$

*where $E_0 := \mathbb{E}\|e^{-1}\|_\star$ is the initial momentum lag. The associated* sample complexity *(total number of stochastic gradients) $T(b,\varepsilon) = b\, K(b,\varepsilon)$ satisfies*

$$T(b,\varepsilon) \;=\; \mathcal{O}\!\left(\frac{bL\Delta_0}{\varepsilon^2} \;\vee\; \frac{E_0\rho^2 L_{\max}\overline{\Delta}}{\varepsilon^3} \cdot \frac{n-b}{n-1} \;\vee\; \frac{L\rho^2 L_{\max}\overline{\Delta}^2}{\varepsilon^4} \cdot \frac{n-b}{n-1}\right). \tag{88}$$

**Dependence on the Batch Size.**  The three terms in (88) correspond to:

- A *deterministic* baseline $\mathcal{O}(bL\Delta_0/\varepsilon^2)$, present even in the absence of stochasticity; it grows linearly with $b$.

- A *burn-in* term $\mathcal{O}\left(E_0\rho^2 L_{\max}\overline{\Delta}/\varepsilon^3 \cdot \frac{n-b}{n-1}\right)$, which is essentially independent of $b$ for moderate mini-batches $b \ll n$ (since $\frac{n-b}{n-1} \approx 1$), and vanishes as $b \to n$.

- A *stochastic* term $\mathcal{O}\left(L\rho^2 L_{\max}\overline{\Delta}^2/\varepsilon^4 \cdot \frac{n-b}{n-1}\right)$, which dominates in the small-$\varepsilon$ regime. For $b \ll n$ this term is again essentially independent of $b$, while it decays linearly to 0 as $b \to n$.

Thus, for mini-batch sizes $b$ much smaller than $n$, the normalized method with momentum has *sample complexity* in the stochastic regime that is (up to constants) *essentially independent of* $b$: increasing the batch size trades iterations for per-iteration cost without changing the total number of stochastic gradients. Only when $b$ becomes comparable to $n$ does the stochastic contribution significantly decrease (as the noise vanishes and the method approaches deterministic gradient descent), at which point the deterministic term $bL\Delta_0/\varepsilon^2$ becomes the dominant one.

### A.8. Optimal Mini-Batch Size for the LMO-Based Method Under $\tau$–Nice Sampling

In this subsection we specialize the $\varepsilon$–complexity result of Theorem A.3 to the case of $\tau$–nice sampling without replacement with mini-batch size $b \in \{1, \ldots, n\}$, and study how the total number of stochastic gradients

$$T(b) := K(b) \cdot b$$

needed to reach an $\varepsilon$–stationary point depends on $b$. We focus on the *LMO-based method with momentum*.

**ABC Constants for $\tau$–Nice Sampling.**  Recall that we consider a finite–sum objective

$$f(x) = \frac{1}{n}\sum_{i=1}^{n} f_i(x),$$

where each $f_i$ is $L_i$–smooth and bounded below by $f_i^{\inf}$. Let $b$ denote the mini-batch size and assume $\tau$–nice sampling *without replacement*, i.e., each subset of indices of cardinality $b$ is equally likely.

As discussed earlier, the ABC constants under this sampling are (with $L_{\max} := \max_i L_i$ and $\Delta_{\inf} := \frac{1}{n}\sum_{i=1}^{n}(f_{\inf} - f_i^{\inf})$)

$$A(b) = \frac{n-b}{b\,(n-1)}\,L_{\max}, \qquad B(b) = \frac{n(b-1)}{b\,(n-1)} \in [0,1], \qquad C(b) = 2A(b)\,\Delta_{\inf}. \tag{89}$$

Since $B(b) \leq 1$ for all $b \leq n$, we have $(B(b) - 1)_+ = 0$ and therefore

$$A_{\mathrm{eff}}(b) = A(b), \qquad Q^2(b) = 2\rho^2 A_{\mathrm{eff}}(b) = 2\rho^2 A(b).$$

**Plugging into the General Complexity Bound.**  Let $\Delta_0 = f(x^0) - f_{\inf}$ and $\Delta_* := \max\{\Delta_0, \Delta_{\inf}\}$. From Theorem A.3, there exist universal constants $c_1, c_2, c_3 > 0$ such that for any $\varepsilon \in (0, 1]$ one can choose $(\alpha, \gamma)$ so that the iteration complexity of the LMO-based method satisfies

$$K(\varepsilon) \leq \mathcal{O}\left(\frac{L\,\Delta_0}{\varepsilon^2} \vee \frac{E_0\,S^2}{\varepsilon^3} \vee \frac{L\,S^4}{Q^2\,\varepsilon^4}\right), \tag{90}$$

where

$$S^2 := \max\{\rho^2 C, \, Q^2\Delta_0\}, \qquad E_0 := \mathbb{E}\|e^{-1}\|_\star.$$

For the $\tau$–nice mini-batching (89), we have

$$\rho^2 C(b) = 2\rho^2 A(b)\,\Delta_{\inf}, \qquad Q^2(b)\Delta_0 = 2\rho^2 A(b)\Delta_0,$$

so

$$S^2(b) \;=\; \max\{\rho^2 C(b),\, Q^2(b)\Delta_0\} \;=\; 2\rho^2 A(b)\,\Delta_* \tag{91}$$

and hence

$$\frac{S^4(b)}{Q^2(b)} \;=\; \frac{\left(2\rho^2 A(b)\Delta_*\right)^2}{2\rho^2 A(b)} \;=\; 2\rho^2 A(b)\,\Delta_*^2. \tag{92}$$

Substituting (91) and (92) into (90) yields the *iteration complexity* under $\tau$–nice sampling:

$$K_b(\varepsilon) \;:=\; K(\varepsilon,b) \;=\; \mathcal{O}\left(\underbrace{\frac{L\,\Delta_0}{\varepsilon^2}}_{\text{deterministic}} \;\vee\; \underbrace{\frac{E_0\,\rho^2 A(b)\,\Delta_*}{\varepsilon^3}}_{\text{burn-in}} \;\vee\; \underbrace{\frac{L\,\rho^2 A(b)\,\Delta_*^2}{\varepsilon^4}}_{\text{stochastic/ABC}}\right). \tag{93}$$

Using the explicit form of $A(b)$ from (89),

$$A(b) \;=\; \frac{n-b}{b\,(n-1)}\,L_{\max},$$

we see that both the burn-in and the stochastic term are proportional to $\frac{n-b}{b}$.

**Sample Complexity and Its Dependence on $b$.** The total number of stochastic gradients used is

$$T(b,\varepsilon) \;:=\; b \cdot K_b(\varepsilon).$$

Using (93), this yields

$$T(b,\varepsilon) \;=\; \mathcal{O}\left(\underbrace{\frac{L\,\Delta_0}{\varepsilon^2}\,b}_{\text{deterministic part}} \;\vee\; \underbrace{\frac{E_0\,\rho^2 L_{\max}\,\Delta_*}{\varepsilon^3}\cdot\frac{n-b}{n-1}}_{\text{burn-in part}} \;\vee\; \underbrace{\frac{L\,\rho^2 L_{\max}\,\Delta_*^2}{\varepsilon^4}\cdot\frac{n-b}{n-1}}_{\text{stochastic/ABC part}}\right). \tag{94}$$

The dependence on $b$ is therefore as follows:

- The *deterministic* term grows linearly in $b$ (more gradients per iteration).

- The two *stochastic* terms decrease linearly in $b$, via the factor $\frac{n-b}{n-1}$ inherited from $A(b)$.

For moderate accuracy, the burn-in term may be relevant; however, in the high-accuracy regime $\varepsilon \to 0$ the last (stochastic/ABC) term with $\varepsilon^{-4}$ dominates the stochastic behavior. In that regime, (94) can be summarized as

$$T(b,\varepsilon) \;=\; \mathcal{O}\left(\max\left\{\frac{L\,\Delta_0}{\varepsilon^2}\,b,\; \frac{L\,\rho^2 L_{\max}\,\Delta_*^2}{\varepsilon^4}\cdot\frac{n-b}{n-1}\right\}\right), \tag{95}$$

up to constants and lower-order terms in $\varepsilon$.

**Critical Mini-Batch Size for the Normalized Method.** The right-hand side of (95) is the maximum of

- an *increasing* function of $b$: $T_{\det}(b) \propto b$,

- a *decreasing* function of $b$: $T_{\text{stoch}}(b) \propto (n-b)$.

Hence, $T(b,\varepsilon)$ is minimized (up to constants) when these two contributions are of the same order. Ignoring universal constants, the *critical* batch size $b^\star(\varepsilon)$ is obtained by solving

$$\frac{L\Delta_0}{\varepsilon^2}\,b \;\approx\; \frac{L\,\rho^2 L_{\max}\,\Delta_*^2}{\varepsilon^4}\cdot\frac{n-b}{n-1}.$$

Cancelling $L$ and rearranging, this is

$$\Delta_0\,b\,\varepsilon^2(n-1) \;\approx\; \rho^2 L_{\max}\,\Delta_*^2\,(n-b).$$

Define the dimensionless ratio

$$H_{\mathrm{norm}}(\varepsilon) \;:=\; \frac{\rho^2 L_{\max} \Delta_*^2}{\Delta_0 (n-1) \varepsilon^2}.$$

Then the balance condition becomes $b \approx H_{\mathrm{norm}}(\varepsilon)(n-b)$, which yields

$$b^\star(\varepsilon) \;\approx\; \frac{H_{\mathrm{norm}}(\varepsilon)}{1 + H_{\mathrm{norm}}(\varepsilon)}\, n. \tag{96}$$

Equivalently:

- For $b < b^\star(\varepsilon)$, the LMO-based method is in the *stochastic* regime: the $\varepsilon^{-4}$ term dominates and $T(b,\varepsilon)$ *decreases* with $b$.

- For $b > b^\star(\varepsilon)$, the method is in the *deterministic* regime: the $\varepsilon^{-2}$ term dominates and $T(b,\varepsilon)$ *increases* with $b$.

Thus $b^\star(\varepsilon)$ plays the role of a *critical mini-batch size* for the normalized method with momentum under $\tau$–nice sampling.

**Asymptotic Behavior of $b^\star(\varepsilon)$.** From the definition of $H_{\mathrm{norm}}(\varepsilon)$ we see that

$$H_{\mathrm{norm}}(\varepsilon) \;=\; \Theta\!\Big(\frac{1}{\varepsilon^2}\Big) \qquad \Longrightarrow \qquad b^\star(\varepsilon) \to n \quad \text{as} \quad \varepsilon \to 0.$$

That is, as we require higher accuracy, the *optimal* mini-batch size for the LMO-based method moves closer to full-batch, reflecting the increasingly dominant role of the ABC–driven stochastic term.

For moderate accuracies (larger $\varepsilon$), $H_{\mathrm{norm}}(\varepsilon)$ may be of order one, and (96) predicts a nontrivial interior optimum $b^\star(\varepsilon) \in (1, n)$, exactly mirroring the critical batch size phenomenon for SGD under ABC, but with a *different* effective heterogeneity parameter $H_{\mathrm{norm}}(\varepsilon)$ that encodes both the problem data $(L_{\max}, \Delta_0, \Delta_{\inf})$ and the geometry via $\rho$.

### A.9. Bounded-Variance vs. $\tau$-Nice Subsampling: What Changes for the LMO-Based Method?

This subsection contrasts the implications of two standard stochastic models for mini-batching: (i) the *bounded variance* (BV) oracle, and (ii) $\tau$-*nice sampling without replacement* in the finite-sum setting.

We focus on the *LMO-based method with momentum* (Theorem A.1) and highlight how the predicted batch-size dependence differs across models, both in *iteration complexity* $K(b,\varepsilon)$ and in *sample complexity* $T(b,\varepsilon) := b \cdot K(b,\varepsilon)$. Throughout, $\Delta_0 := f(x^0) - f_{\inf}$, $E_0 := \mathbb{E}\|e^{-1}\|_*$ denotes the initial momentum lag, $\rho \geq 1$ is the norm-compatibility constant (so $\|\cdot\|_* \leq \rho\|\cdot\|_2$), and $L_{\max} := \max_i L_i$ in the finite-sum model.

We also write $\overline{\Delta} := \max\{\Delta_0, \Delta_{\inf}\}$, where $\Delta_{\inf}$ is the constant appearing in the $\tau$-nice ABC specialization (cf. (84)).

**1. Bounded-Variance (BV) Specialization and Its Batch-Size Implications.** Consider the BV oracle model with mini-batch size $b$, where $g^k$ is unbiased and satisfies the Euclidean variance bound

$$\mathbb{E}\left[\|g(x) - \nabla f(x)\|_2^2\right] \;\leq\; \frac{\sigma^2}{b} \qquad \text{for all } x, \tag{97}$$

for some per-sample variance proxy $\sigma^2$.

This corresponds to the ABC parameters

$$A = 0, \qquad B = 1, \qquad C = \frac{\sigma^2}{b},$$

so that $A_{\mathrm{eff}} = 0$ and hence $Q = 0$ in Theorem A.1.

In this case, the master bound reduces to

$$\frac{1}{K}\sum_{k=0}^{K-1} \mathbb{E}\|\nabla f(x^{k+1})\|_* \;\lesssim\; \frac{\Delta_0}{\gamma K} \;+\; L\gamma\Big(1 + \frac{1-\alpha}{\alpha}\Big) \;+\; \frac{E_0}{\alpha K} \;+\; \rho\sqrt{\frac{\sigma^2}{b}}\,\sqrt{\alpha},$$

(up to absolute constants), and a standard $\varepsilon$-splitting yields the *iteration complexity*

$$K_{\mathrm{BV}}(b, \varepsilon) \;=\; \mathcal{O}\left(\frac{L\Delta_0}{\varepsilon^2} \;\vee\; \frac{E_0}{\varepsilon} \;\vee\; \frac{E_0\rho^2\sigma^3}{b^{3/2}\varepsilon^3} \;\vee\; \frac{L\Delta_0\rho^2\sigma^2}{b\varepsilon^4}\right). \tag{98}$$

Multiplying by $b$ gives the corresponding *sample complexity*

$$T_{\mathrm{BV}}(b, \varepsilon) \;:=\; b \cdot K_{\mathrm{BV}}(b, \varepsilon) \;=\; \mathcal{O}\left(\frac{bL\Delta_0}{\varepsilon^2} \;\vee\; \frac{bE_0}{\varepsilon} \;\vee\; \frac{E_0\rho^2\sigma^2}{\sqrt{b}\varepsilon^3} \;\vee\; \frac{L\Delta_0\rho^2\sigma^2}{\varepsilon^4}\right). \tag{99}$$

**Key BV message.**

In the BV model (97), the *high-accuracy stochastic terms* (the $\varepsilon^{-3}$ and $\varepsilon^{-4}$ contributions in (99)) are *independent of $b$* at the level of sample complexity because the $1/b$ variance reduction cancels with the $b$ samples used per iteration. Thus, (99) predicts that mini-batching does *not* improve the dominant stochastic sample complexity in the small-$\varepsilon$ regime. Any *nontrivial* batch-size effect in BV must therefore come from (i) the *deterministic* cost $bL\Delta_0/\varepsilon^2$, or (ii) the *initialization-dependent* lag $E_0$.

**Optional refinement (only if one relates $E_0$ to $b$).**

If one initializes with a stochastic mini-batch momentum $m^0 = g^0$ of batch size $b$, then $E_0 = \mathbb{E}\|g^0 - \nabla f(x^0)\|_* \leq \rho\sigma/\sqrt{b}$ by Jensen and norm compatibility. Plugging this into (99) yields the more explicit $b$-dependence

$$T_{\mathrm{BV}}(b, \varepsilon) \;=\; \mathcal{O}\left(\frac{bL\Delta_0}{\varepsilon^2} \;\vee\; \frac{\rho\sigma}{\varepsilon}\sqrt{b} \;\vee\; \frac{\rho^3\sigma^3}{\varepsilon^3\sqrt{b}} \;\vee\; \frac{L\Delta_0\rho^2\sigma^2}{\varepsilon^4}\right), \tag{100}$$

which indeed exhibits a *genuine trade-off* between a term increasing as $\sqrt{b}$ and one decreasing as $1/\sqrt{b}$. Balancing these two gives a BV "critical batch size" $b_{\mathrm{BV}}^\star \asymp (\rho\sigma/\varepsilon)^2$ (up to clipping to $[1, n]$), while the $\varepsilon^{-4}$ term remains batch-invariant.

**2. $\tau$-Nice Sampling Without Replacement: The Finite-Population Correction Survives in Sample Complexity.** Now consider the finite-sum model $f(x) = \frac{1}{n}\sum_{i=1}^n f_i(x)$ with $\tau$-nice mini-batching of size $b \in \{1, \ldots, n\}$.

The ABC specialization takes the form

$$A(b) = \frac{n-b}{b(n-1)}L_{\max}, \qquad B(b) = \frac{n(b-1)}{b(n-1)} \in [0, 1], \qquad C(b) = 2A(b)\Delta_{\inf}. \tag{101}$$

Since $B(b) \leq 1$, we have $A_{\mathrm{eff}} = A(b)$, hence

$$Q^2 = 2\rho^2 A(b), \qquad \rho^2 C(b) = 2\rho^2 A(b)\Delta_{\inf}.$$

Using $\overline{\Delta} = \max\{\Delta_0, \Delta_{\inf}\}$, Corollary A.5 can be read as the (sufficient) bound

$$K_\tau(b, \varepsilon) \;=\; \mathcal{O}\left(\frac{L\Delta_0}{\varepsilon^2} \;\vee\; \frac{E_0\rho^2 L_{\max}\overline{\Delta}}{\varepsilon^3} \cdot \frac{n-b}{b(n-1)} \;\vee\; \frac{L\rho^2 L_{\max}\overline{\Delta}^2}{\varepsilon^4} \cdot \frac{n-b}{b(n-1)}\right), \tag{102}$$

and hence the sample complexity becomes

$$T_\tau(b, \varepsilon) \;:=\; b \cdot K_\tau(b, \varepsilon) \;=\; \mathcal{O}\left(\frac{bL\Delta_0}{\varepsilon^2} \;\vee\; \frac{E_0\rho^2 L_{\max}\overline{\Delta}}{\varepsilon^3} \cdot \frac{n-b}{n-1} \;\vee\; \frac{L\rho^2 L_{\max}\overline{\Delta}^2}{\varepsilon^4} \cdot \frac{n-b}{n-1}\right). \tag{103}$$

**Key $\tau$-nice message.**

In contrast to BV, the stochastic terms in (103) retain the *finite-population correction*

$$\frac{n-b}{n-1},$$

which *does not cancel* in sample complexity. As a result, even the leading $\varepsilon^{-4}$ term decreases with batch size and vanishes as $b \to n$ (where the method becomes deterministic since $A(b) = C(b) = 0$). This is precisely the qualitative behavior that the BV model cannot express: in a true finite-sum problem, going to larger batches *can* reduce sample complexity because sampling without replacement becomes progressively closer to exact/full-gradient information.

**3. BV vs. $\tau$-Nice: What Do the Two Models Predict Differently?**

- **Does the leading stochastic sample complexity depend on $b$?**

  From (99), under BV the dominant stochastic sample complexity terms scale as $\varepsilon^{-3}$ and $\varepsilon^{-4}$ and are *independent of $b$*. From (103), under $\tau$-nice the corresponding terms are multiplied by $(n-b)/(n-1)$ and hence *decrease with $b$*. This is the most direct theoretical distinction between the two models.

- **Where can a nontrivial "critical batch size" come from?**

  In the BV model, if one keeps $E_0$ abstract, then $b$ only appears in the increasing terms $bL\Delta_0/\varepsilon^2$ and $bE_0/\varepsilon$ in (99), so minimizing $T_{\mathrm{BV}}(b, \varepsilon)$ over $b$ is essentially trivial (favoring the smallest $b$). A nontrivial BV critical batch size only emerges after modeling how $E_0$ scales with $b$ (e.g., (100)). In contrast, under $\tau$-nice, even with $E_0$ treated as an abstract constant, the stochastic terms in (103) decrease with $b$ through $(n-b)/(n-1)$, so the theory predicts a genuine trade-off (and hence a critical batch size) at the level of sample complexity.

- **Large-$n$ limit (expectation-like regime).**

  If $n \to \infty$ with $b$ fixed, then $(n-b)/(n-1) \to 1$ and $A(b) \approx L_{\max}/b$ in (101). Consequently, the $\tau$-nice sample complexity (103) approaches a BV-like behavior in which the stochastic terms become effectively batch-independent. In other words, the *difference* between BV and $\tau$-nice is most pronounced precisely when $b$ is a non-negligible fraction of $n$ (finite-population regime), whereas for $n \gg b$ the two models become close.

- **Deterministic limit as $b \to n$.**

  Under $\tau$-nice, (103) correctly predicts that the stochastic contribution vanishes as $b \to n$ (since $A(b), C(b) \to 0$), recovering deterministic behavior. The BV model does not encode this limit unless one explicitly enforces $\sigma^2 \to 0$ with $b$, which is one reason why ABC/$\tau$-nice is a more faithful abstraction of subsampling in finite sums.

**4. Comparison to SGD Under ABC/$\tau$-Nice (Khaled & Richtárik, 2023).** For SGD under $\tau$-nice sampling, the sample complexity derived from the ABC analysis takes the form

$$T_{\mathrm{SGD}}(b, \varepsilon) \geq \frac{12L\Delta_0}{\varepsilon^2} \max\left\{ \frac{n(b-1)}{n-1}; \frac{H(n-b)}{n-1} \right\}, \qquad H = \frac{L_{\max}}{\varepsilon^2} \max\{12\Delta_0, 4\Delta_{\inf}\}. \tag{104}$$

This has a clean "increasing vs. decreasing" structure and yields a critical batch size $b^{\star}_{\mathrm{SGD}} = \frac{n(H+1)}{n+H}$ (cf. the discussion around (111)–(113)). For the LMO-based method under $\tau$-nice, (103) has a similar qualitative shape:

- an *increasing* branch $\frac{bL\Delta_0}{\varepsilon^2}$ (deterministic cost of using $b$ gradients per step),

- and *decreasing* branches proportional to $\frac{n-b}{n-1}$ (stochasticity shrinks as $b$ increases).

However, the constants and the accuracy-dependence differ: the dominant decreasing branch for the LMO-based method scales as $\varepsilon^{-4}$ with coefficient $L\rho^2 L_{\max} \overline{\Delta}^2$, whereas for SGD the corresponding decreasing branch coefficient in (104) scales as $L\Delta_0 \cdot (L_{\max} \overline{\Delta})$ (up to constants), and the *increasing branch* differs slightly as well ($n(b-1)$ vs. $b$). As a result, both methods admit a critical batch size under $\tau$-nice sampling, but the *location* of the critical batch (and the sharpness of the trade-off) depends differently on the problem constants. In particular, because the LMO-based method

carries additional geometric factors (notably $\rho^2$) and involves $\overline{\Delta}^2$ in the leading stochastic term, its theory can favor larger batches more strongly in regimes where this stochastic branch dominates, which is consistent with the empirical lore that LMO/orthogonalization-based optimizers are most effective in large-batch training.

**Takeaway.**

Both SGD and the LMO-based method exhibit a meaningful critical batch phenomenon under $\tau$-nice sampling (finite sums), but the BV model largely suppresses this phenomenon at the level of sample complexity unless one explicitly models how initialization-related quantities (e.g., $E_0$) scale with $b$. This distinction supports using ABC/$\tau$-nice as the default model when the goal is to understand and predict the effect of subsampling and mini-batching in finite-sum training.

### A.10. Expected Strong Growth as a Special ABC Regime

We now consider a standard "strong growth"–type assumption on the stochastic gradients and show how it simplifies Theorem A.1 and the resulting complexity.

**Assumption A.6** (Expected strong growth (ESG)). There exists a constant $\kappa \geq 1$ such that for all $x$,

$$\mathbb{E}\big[\|g(x)\|_2^2 \,\big|\, x\big] \;\leq\; \kappa \,\|\nabla f(x)\|_2^2,$$

and the oracle is unbiased, $\mathbb{E}[g(x) \mid x] = \nabla f(x)$.

**Lemma A.7** (ESG as a special case of ABC). *Under Assumption A.6, the stochastic gradient $g(x)$ satisfies ABC condition*

$$\mathbb{E}\big[\|g(x)\|_2^2\big] \;\leq\; 2A\,\Delta(x) + B\,\|\nabla f(x)\|_2^2 + C$$

*with*

$$A = 0, \qquad B = \kappa, \qquad C = 0.$$

*Consequently,*

$$A_{\mathrm{eff}} \;=\; A + L(B-1)_+ \;=\; L\,(\kappa-1)_+, \qquad Q \;=\; \rho\,\sqrt{2A_{\mathrm{eff}}} = \rho\,\sqrt{2L\,(\kappa-1)_+},$$

*and the pure-variance term vanishes: $C = 0 \Rightarrow \mathcal{T}_C = 0$ in (58).*

*Proof.* The proof follows from the definition of ESG. $\qquad\square$

## B. SGD with Mini-Batching Analysis

We specialize the ABC constants to $\tau$–nice sampling without replacement with mini-batch size $b \in \{1, \ldots, n\}$. For this scheme the exact constants are (here $L_{\max} := \max_i L_i$)

$$A(b) = \frac{n - b}{b\,(n - 1)}\,L_{\max}, \qquad B(b) = \frac{n(b - 1)}{b\,(n - 1)}, \qquad C(b) = 2\,A(b)\,\Delta^{\inf}, \tag{105}$$

where $\Delta^{\inf} \geq 0$ is the (problem-dependent) constant entering the ABC condition.[2]

**Admissible Stepsize (from (Khaled & Richtárik, 2023)).**  Let $L$ denote the smoothness constant of $f$ in the (primal) norm used to state the descent lemma (and the bound). For any iteration budget $K \geq 1$ and target tolerance $\varepsilon > 0$, choose the constant stepsize

$$\gamma = \min\left\{ \underbrace{\frac{1}{\sqrt{L\,A(b)\,K}}}_{\gamma_A(b, K)}, \ \underbrace{\frac{1}{L\,B(b)}}_{\gamma_B(b)}, \ \underbrace{\frac{\varepsilon}{2\,L\,C(b)}}_{\gamma_C(b,\varepsilon)} \right\}. \tag{106}$$

With the concrete constants (105) this becomes

$$\gamma_A(b, K) = \sqrt{\frac{b\,(n - 1)}{(n - b)\,L\,L_{\max}\,K}}, \qquad \gamma_B(b) = \frac{b\,(n - 1)}{L\,n\,(b - 1)}, \qquad \gamma_C(b, \varepsilon) = \frac{\varepsilon}{4L\,\Delta^{\inf}} \cdot \frac{b\,(n - 1)}{(n - b)\,L_{\max}}. \tag{107}$$

*Dependence on $b$.*  Both $\gamma_A$ and $\gamma_C$ scale as $b/(n - b)$ and hence increase monotonically with $b$ (blowing up as $b \to n$), while $\gamma_B$ grows like $b/(b - 1)$ and quickly saturates at $1/L$ as $b$ increases. Consequently, for small $b$ the active cap in (106) is typically $\gamma_A$ (or $\gamma_C$ when $\varepsilon$ is very small), whereas for sufficiently large $b$ the admissible stepsize saturates at $\gamma_B \approx 1/L$ and no longer benefits from increasing $b$.

**Iteration and Sample Complexity (from (Khaled & Richtárik, 2023)).**  Let $\Delta^0 = f(x^0) - f_{\inf} \geq 0$. With the stepsize (106), SGD returns an $\varepsilon$–stationary point in the sense that $\min_{0 \leq k \leq K-1} \mathbb{E}\,\|\nabla f(x^k)\|_\star \leq \varepsilon$ provided

$$K \geq \frac{12\,L\,\Delta^0}{\varepsilon^2}\,\max\left\{ B(b),\ \underbrace{\frac{12\,\Delta^0}{\varepsilon^2}\,A(b)}_{\text{from } A},\ \underbrace{\frac{2\,C(b)}{\varepsilon^2}}_{\text{from } C} \right\}. \tag{108}$$

Substituting (105) and collecting the terms that depend on $b$ gives

$$K(b) \geq \frac{12\,L\,\Delta^0}{\varepsilon^2}\,\max\left\{ \frac{n(b - 1)}{b\,(n - 1)},\ \frac{n - b}{b\,(n - 1)} \cdot \frac{L_{\max}}{\varepsilon^2}\,\max\{12\,\Delta^0,\ 4\,\Delta^{\inf}\} \right\} \tag{109}$$

$$= \frac{12\,L\,\Delta^0}{\varepsilon^2}\,\max\left\{ \frac{n(b - 1)}{b\,(n - 1)},\ H \cdot \frac{n - b}{b\,(n - 1)} \right\}, \qquad H := \frac{L_{\max}}{\varepsilon^2}\,\max\{12\,\Delta^0,\ 4\,\Delta^{\inf}\}. \tag{110}$$

The *sample complexity* (number of stochastic gradients) is $T(b) := b\,K(b)$. Multiplying (109) by $b$ yields the remarkably simple piecewise-linear dependence

$$T(b) \geq \frac{12\,L\,\Delta^0}{\varepsilon^2}\,\max\left\{ \underbrace{\frac{n(b - 1)}{n - 1}}_{\uparrow\text{ in } b},\ \underbrace{\frac{H(n - b)}{n - 1}}_{\downarrow\text{ in } b} \right\}. \tag{111}$$

**Critical Batch Size and Optimal Sample Complexity.**  The right-hand side of (111) is the maximum of an increasing and a decreasing affine function of $b$, hence it is minimized at the (unique) intersection. Solving

$$\frac{n(b^\star - 1)}{n - 1} = \frac{H(n - b^\star)}{n - 1} \quad \Longleftrightarrow \quad b^\star = \frac{n\,(H + 1)}{n + H},$$

---

[2]We only use that $\Delta^{\inf}$ is finite and nonnegative. No additional structure is needed here.

we obtain the *critical batch size*

$$b^\star = \frac{n(H+1)}{n+H} \in (1,n), \qquad \text{and one may take} \quad \widehat{b} \in \{\lfloor b^\star \rfloor, \lceil b^\star \rceil\}. \tag{112}$$

Evaluating (111) at $b^\star$ (the two branches coincide there) gives

$$T(b^\star) = \frac{12\, L\, \Delta^0}{\varepsilon^2} \cdot \frac{n\, H}{n+H}. \tag{113}$$

For the two extremes,

$$T(1) = \frac{12\, L\, \Delta^0}{\varepsilon^2}\, H, \qquad T(n) = \frac{12\, L\, \Delta^0}{\varepsilon^2}\, n, \qquad \text{and} \qquad T(b^\star) = \min_{1 \le b \le n} T(b) \le \min\{T(1), T(n)\}. \tag{114}$$

**Interpretation (Existence and Size of the Critical Batch).** The dimensionless parameter $H$ in (110) aggregates the data-dependent quantities $(L_{\max}, \Delta^0, \Delta^{\inf})$ and the target accuracy $\varepsilon$:

$$H = \frac{L_{\max}}{\varepsilon^2} \cdot \max\{12\,\Delta^0,\, 4\,\Delta^{\inf}\}.$$

- **When $H \ll n$** (coarse accuracy, small $L_{\max}$, or small gaps), the critical batch is near *single-sample*: $b^\star = \frac{n(1+H)}{n+H} = 1 + \mathcal{O}(H/n)$ and $T(b^\star) \approx T(1) = \frac{12\, L\, \Delta^0}{\varepsilon^2}\, H$. Larger batches do not reduce sample complexity.

- **When $H \gg n$** (stringent accuracy or large $L_{\max}/\Delta^{\inf}$), the critical batch moves towards *full batch*: $b^\star = \frac{n(H+1)}{n+H} = n - \mathcal{O}(n^2/H)$ and $T(b^\star) \approx T(n) = \frac{12\, L\, \Delta^0}{\varepsilon^2}\, n$. In this regime, increasing $b$ significantly reduces the dominant $A/C$–driven part, so large batches are sample-efficient.

- **Phase transition (nontrivial optimum).** For intermediate $H$ the optimal batch is interior: $b^\star \in (1,n)$ as in (112), and the minimal sample complexity $T(b^\star)$ in (113) improves over both extremes by the factors

$$\frac{T(b^\star)}{T(1)} = \frac{n}{n+H} \in (0,1), \qquad \frac{T(b^\star)}{T(n)} = \frac{H}{n+H} \in (0,1).$$

**Which Stepsize Cap Is Active at the Optimum?** For $b$ near $b^\star$, the two *structural* contributions inside the max in (109) are equal, so the iteration budget is tuned to balance the $B$–term ("bias-like", increasing with $b$) and the $A/C$–terms ("variance-like", decreasing with $b$). On the stepsize side, $\gamma_A$ and $\gamma_C$ behave like $b/(n-b)$ while $\gamma_B$ saturates to $1/L$. Thus, as $b$ increases from 1 to $b^\star$, the active cap typically transitions from $\gamma_A$ (or $\gamma_C$ when $\varepsilon$ is very small) to $\gamma_B$; beyond this point the admissible stepsize no longer grows with $b$. This is the standard *stepsize saturation* phenomenon under mini-batching.

**Summary.** Under $\tau$–nice sampling without replacement, SGD admits an explicit *critical batch size* (112) that minimizes the sample complexity (111). The optimal $b$ depends on $H = (L_{\max}/\varepsilon^2) \max\{12\Delta^0,\, 4\Delta^{\inf}\}$: small $H$ favors $b = 1$, large $H$ favors $b = n$, and otherwise a nontrivial interior optimum exists. The corresponding minimal sample complexity is given by (113), and the admissible constant stepsize is the minimum of the three explicit caps in (107).

### B.1. Empirical Small-Batch Results for LLMs vs. ABC Theory (SGD)

Recent empirical studies investigate language–model pretraining and fine-tuning with *small* batch sizes and report that carefully tuned SGD can be competitive with Adam/AdamW, and that gradient accumulation is often wasteful at fixed compute budgets.[3] We summarize their main observations before mapping them to our ABC analysis.

---

[3] Marek et al. (2025) and Srećković–Geiping–Orvieto (2025); see discussion below.

**What the Experiments Find.**

1. **Small batches are stable, robust, and per-FLOP efficient.** Across models up to $\sim 1.3$B parameters and token budgets in the billions, batch sizes down to 1–16 train stably and show broader "good" hyperparameter regions (compared to large batches). Per FLOP, small batches achieve equal or better loss, and vanilla SGD (no momentum, no state) can train LMs stably.[4] (Marek et al., 2025)

2. **SGD vs. Adam: the batch size decides the gap.** Under a *fixed token budget* (fixed compute), the Adam–SGD gap *widens* as batch size grows; for *small* batches and adequate tuning (momentum, clipping), SGD matches Adam (even at 410M–1B scale). With a *fixed number of steps*, larger batches reduce the gap because SGD simply needs more steps; momentum and clipping become crucial as batch size grows.[5] (Srećković et al., 2025)

3. **Why small batches help in practice.** (i) They allow many more parameter updates for a given FLOP budget, (ii) they are more robust to hyperparameter misspecification, and (iii) they obviate the need for sophisticated optimizer state.[6] (Marek et al., 2025)

**Mapping to ABC Theory with $\tau$-Nice Sampling.** Under $\tau$-nice sampling without replacement (global batch $b = \tau$),

$$A(b) = \frac{n-b}{b(n-1)} \max_i L_i, \qquad B(b) = \frac{n(b-1)}{b(n-1)}, \qquad C(b) = 2A(b)\,\Delta_{\inf}.$$

From Corollary 1 of Khaled & Richtárik (2023) (specialized to SGD), choosing $\gamma = \min\left\{\frac{1}{\sqrt{LA(b)K}},\ \frac{1}{LB(b)},\ \frac{\varepsilon}{2LC(b)}\right\}$, it suffices to take

$$K \ \gtrsim\ \frac{12\,\Delta^0\,L}{\varepsilon^2}\ \max\left\{B(b),\ \frac{12\,\Delta^0}{\varepsilon^2}A(b),\ \frac{2\,C(b)}{\varepsilon^2}\right\}.$$

Equivalently, the *sample* (token) complexity $T(b) = K(b)\,b$ obeys (up to absolute constants)

$$T(b) \ \propto\ b \cdot \max\left\{ \underbrace{B(b)}_{\text{increasing in } b},\ \underbrace{\frac{12\,\Delta^0}{\varepsilon^2}A(b)}_{\text{decreasing in } b},\ \underbrace{\frac{2\,C(b)}{\varepsilon^2}}_{\text{scales like } A(b)} \right\}.$$

In the *interpolation* regime ($C \approx 0$), the tradeoff is between $B(b) = \frac{n(b-1)}{b(n-1)}$ (increasing in $b$) and $\frac{12\,\Delta^0}{\varepsilon^2}A(b) = \frac{12\,\Delta^0}{\varepsilon^2}\frac{n-b}{b(n-1)}L_{\max}$ (decreasing in $b$). As a function of $b$, $T(b)$ is the pointwise maximum of an increasing and a decreasing line; hence it is minimized at their intersection. Setting the two terms equal yields the *critical batch size*

$$b^\star \ =\ \frac{n\,(c+1)}{n+c}, \qquad c \ :=\ \frac{12\,\Delta^0\,L_{\max}}{\varepsilon^2}.$$

Two regimes are particularly informative:

- **Far from stationarity / modest target $\varepsilon$ ($c \ll n$).** Then $b^\star \approx c + 1$. If $c \lesssim 1$, the optimum is essentially $b^\star \simeq 1$: small batches minimize tokens (FLOPs) to reach $\varepsilon$. This matches the finding that under fixed token budgets, SGD improves as $b$ decreases and can match Adam at small $b$. (Srećković et al., 2025)

- **Extreme accuracy target ($c \gg n$).** Then $b^\star \to n$ (full batch): when chasing very small $\varepsilon$, the $A$-term dominates and larger batches reduce the required *steps*. This regime is atypical for compute-optimal LLM training, which rarely runs to stationarity.[7] (Marek et al., 2025)

---

[4]See abstract and Figs. 1–5 in (Marek et al., 2025); the paper also argues against gradient accumulation unless one replicates models across devices.

[5]See Sec. 2.2–2.3 and Figs. 1–5 in (Srećković et al., 2025).

[6]See the recommendations section and Fig. 1 in (Marek et al., 2025).

[7]See the "far-from-convergence" discussion and compute-optimal training rationale in (Marek et al., 2025).

**Reconciling Practice and Theory.**

1. *Why gradient accumulation can be wasteful.* At fixed tokens/FLOPs, increasing $b$ via accumulation reduces the number of steps $K = T/b$ but typically *increases* the token complexity $T(b)$ whenever $b > b^\star$. Hence small $b$ is the compute-optimal choice unless one targets a very small $\varepsilon$ (large $c$). (Marek et al., 2025)

2. *Why SGD catches up to Adam at small $b$.* Our bound predicts $K(b)$ is controlled by $\max\{B(b), cA(b)\}$. For large $n$, $B(b) \approx 1 - \frac{1}{b}$ and does not penalize small $b$, while $A(b) \propto \frac{1}{b}$ *rewards* small $b$ in the compute-limited view. Empirically, under fixed token budgets, the Adam–SGD gap shrinks as $b$ decreases (and vanishes with enough steps), matching this prediction. (Srećković et al., 2025)

3. *Role of momentum and clipping.* The empirical studies show momentum and (global-norm) clipping are increasingly important as $b$ grows, while for small $b$ even vanilla SGD can be stable.[8] In our ABC bounds, such practices effectively reduce the constants coupled to $A$ and can shrink the radius where the $A$-term dominates, pushing $b^\star$ smaller—again favoring small $b$. (Srećković et al., 2025)

**Takeaways.** (i) In the practically relevant, far-from-stationary LM pretraining regime ($c$ not huge), the ABC theory predicts a critical batch size $b^\star$ near 1, so *small batches minimize the token (compute) budget* to reach an $\varepsilon$-stationary point; this aligns with the recent LLM experiments.[9] (ii) Only when targeting extremely small $\varepsilon$ (or if $L_{\max}\Delta^0$ is massive) does the optimal $b^\star$ become large; this is atypical under compute-optimal training schedules.

---

[8]See Fig. 2 and Sec. 2.2–2.3 in (Srećković et al., 2025).

[9]Small $b$ also improves hyperparameter robustness and reduces optimizer state (Marek et al., 2025).

## B.2. Optimal Batch Size for the LMO-Based Method and Comparison to SGD

We now compare the mini-batch dependence of SGD and the LMO-based method with momentum under $\tau$–nice sampling without replacement (batch size $b$).

**SGD Recap.** From the ABC constants for $\tau$–nice sampling,

$$A(b) \;=\; \frac{n-b}{b\,(n-1)}\,L_{\max}, \qquad B(b) \;=\; \frac{n(b-1)}{b\,(n-1)}, \qquad C(b) \;=\; 2A(b)\,\Delta^{\inf},$$

and the nonconvex SGD rate under ABC (Khaled & Richtárik), one obtains the iteration complexity bound

$$K_{\mathrm{SGD}}(b) \;\geq\; \frac{12\,L\,\Delta^0}{\varepsilon^2}\;\max\left\{\frac{n(b-1)}{b\,(n-1)},\; \frac{n-b}{b\,(n-1)}\cdot\frac{L_{\max}}{\varepsilon^2}\,\max\{\,12\,\Delta^0,\;4\,\Delta^{\inf}\,\}\right\},$$

and thus the sample complexity $T_{\mathrm{SGD}}(b) := b\,K_{\mathrm{SGD}}(b)$ satisfies

$$T_{\mathrm{SGD}}(b) \;\geq\; \frac{12\,L\,\Delta^0}{\varepsilon^2}\;\max\left\{\underbrace{\frac{n(b-1)}{n-1}}_{\uparrow\,\text{in}\,b},\; \underbrace{\frac{H(n-b)}{n-1}}_{\downarrow\,\text{in}\,b}\right\}, \tag{115}$$

where

$$H \;:=\; \frac{L_{\max}}{\varepsilon^2}\,\max\{\,12\,\Delta^0,\;4\,\Delta^{\inf}\,\}.$$

Thus $T_{\mathrm{SGD}}(b)$ is the maximum of an increasing affine function and a decreasing affine function of $b$. The unique minimizer is the *critical batch size*

$$b^\star_{\mathrm{SGD}} \;=\; \frac{n\,(H+1)}{n+H} \in (1,n),$$

and the optimal sample complexity at this batch size is

$$T_{\mathrm{SGD}}(b^\star_{\mathrm{SGD}}) \;=\; \frac{12\,L\,\Delta^0}{\varepsilon^2}\cdot\frac{n\,H}{n+H}.$$

**LMO-Based Method Recap.** For the LMO-based method with momentum, Corollary A.5 (specializing Theorem A.3 to $\tau$–nice sampling) shows that, for sufficiently small $\varepsilon$, the iteration complexity can be written in the same *two–branch* form:

$$K_{\mathrm{LMO}}(b) \;\geq\; \frac{c_0\,L\,\Delta^0}{\varepsilon^2}\;\max\left\{\frac{n(b-1)}{b\,(n-1)},\; \frac{n-b}{b\,(n-1)}\,H_{\mathrm{LMO}}\right\},$$

for some absolute constant $c_0 > 0$ (coming from the detailed $\varepsilon$–splitting in Theorem A.3) and with

$$H_{\mathrm{LMO}} \;=\; \Theta\left(\frac{L_{\max}}{L}\cdot\frac{(\max\{\Delta^0,\Delta^{\inf}\})^2}{\Delta^0\,\varepsilon^2}\right),$$

up to universal numerical constants.[10] The corresponding sample complexity $T_{\mathrm{LMO}}(b) := b\,K_{\mathrm{LMO}}(b)$ therefore satisfies

$$T_{\mathrm{LMO}}(b) \;\geq\; \frac{c_0\,L\,\Delta^0}{\varepsilon^2}\;\max\left\{\underbrace{\frac{n(b-1)}{n-1}}_{\uparrow\,\text{in}\,b},\; \underbrace{H_{\mathrm{LMO}}\frac{(n-b)}{n-1}}_{\downarrow\,\text{in}\,b}\right\}. \tag{116}$$

Thus, *up to a different numerical constant and a different "noise parameter"* $H_{\mathrm{LMO}}$, the LMO-based method has the *same* piecewise–linear structure in $b$ as SGD.

---

[10]The exact expression of $H_{\mathrm{LMO}}$ follows from substituting the $\tau$–nice ABC constants $A(b) = \frac{n-b}{b(n-1)}L_{\max}$ and $C(b) = 2A(b)\Delta^{\inf}$ into Theorem A.3, and then isolating the dominant (stochastic) branch in the $\varepsilon \to 0$ regime. For the batch–size discussion, only the $b$–dependence and the fact that $H_{\mathrm{LMO}}$ does not depend on $b$ are important.

Solving the intersection

$$\frac{n(b-1)}{n-1} = H_{\text{LMO}} \frac{n-b}{n-1} \iff b^{\star}_{\text{LMO}} = \frac{n(H_{\text{LMO}}+1)}{n+H_{\text{LMO}}},$$

we obtain the critical batch size for the LMO-based method, with optimal sample complexity

$$T_{\text{LMO}}(b^{\star}_{\text{LMO}}) = \frac{c_0 L \Delta^0}{\varepsilon^2} \cdot \frac{n H_{\text{LMO}}}{n+H_{\text{LMO}}}.$$

**Qualitative Comparison and Insights.** The formulas (115) and (116) share the same structure:

$$T(b) \propto \max\{ f_{\uparrow}(b), f_{\downarrow}(b) \}, \qquad f_{\uparrow}(b) = \frac{n(b-1)}{n-1}, \quad f_{\downarrow}(b) = H_{\bullet} \frac{n-b}{n-1},$$

where $H_{\bullet}$ stands for $H$ (SGD) or $H_{\text{LMO}}$ (LMO-based method).

This yields several common and method–specific insights:

- **Existence of a unique critical batch size.** For both methods, the sample complexity is the maximum of one term that increases linearly in $b$ ($f_{\uparrow}$, driven by the deterministic gap $L\Delta^0$) and one that decreases linearly in $b$ ($f_{\downarrow}$, driven by variance/ABC constants). Therefore, in both cases there is a unique *critical batch size* $b^{\star}$ where these two effects balance. Increasing $b$ beyond $b^{\star}$ does not reduce the *total number of stochastic gradients*; it only trades more work per iteration for fewer iterations.

- **Regimes $H_{\bullet} \ll n$ and $H_{\bullet} \gg n$.** Using $b^{\star} = n(H_{\bullet}+1)/(n+H_{\bullet})$:

    - If $H_{\bullet} \ll n$, then
      $$b^{\star} \approx 1 + \frac{H_{\bullet}}{n} \approx 1.$$
      In this regime the problem is "easy" relative to the target accuracy $\varepsilon$ (small effective noise / heterogeneity), so the optimal batch size is very small: stochasticity is not the bottleneck, and larger mini-batches only increase the cost per iteration without improving sample efficiency.

    - If $H_{\bullet} \gg n$, then
      $$b^{\star} \approx n\left(1 - \frac{n}{H_{\bullet}}\right) \approx n.$$
      Here the variance/ABC part dominates; one is in a noise–limited regime. Both methods prefer large batches: it is sample–efficient to reduce stochastic noise by using many samples per iteration.

- **Where the methods differ: $H$ versus $H_{\text{LMO}}$.** For SGD,

  $$H = \frac{L_{\max}}{\varepsilon^2} \max\{ 12\Delta^0, 4\Delta^{\inf} \},$$

  depends linearly on $L_{\max}$ and on the initial/suboptimality gaps, and scales as $1/\varepsilon^2$.

  For the LMO-based method,

  $$H_{\text{LMO}} = \Theta\left( \frac{L_{\max}}{L} \cdot \frac{\left(\max\{\Delta^0, \Delta^{\inf}\}\right)^2}{\Delta^0 \varepsilon^2} \right),$$

  which still grows like $1/\varepsilon^2$ but with a different dependence on the geometry ($L_{\max}/L$) and the gaps. In regimes where the effective expected–smoothness constant $A_{\text{eff}}$ is significantly smaller than the global Lipschitz $L$ (e.g., gradients along the trajectory are much "flatter" than the worst–case), one can have $H_{\text{LMO}} < H$. In that case:

  $$b^{\star}_{\text{LMO}} < b^{\star}_{\text{SGD}}, \qquad T_{\text{LMO}}(b^{\star}_{\text{LMO}}) < T_{\text{SGD}}(b^{\star}_{\text{SGD}}) \quad \text{(up to constants)},$$

  meaning that the LMO-based method achieves its optimal sample efficiency with *smaller* mini-batches than SGD and can be more sample–efficient overall.

  Conversely, if $H_{\text{LMO}} > H$ (for example, if the ABC constants are unfavorable for the LMO-based geometry), the critical batch size for the LMO-based method shifts closer to $n$ and its sample complexity may become worse than that of SGD in the high–precision regime.

- **Practical takeaway.** The "critical batch size" phenomenon is *shared* by both SGD and the LMO-based method:

  - For $b < b^\star$, increasing $b$ is beneficial: it reduces the variance–dominated part of the bound faster than it increases the deterministic part, so $T(b)$ decreases.
  - For $b > b^\star$, increasing $b$ is neutral or harmful from a sample–complexity perspective: $T(b)$ does not improve (and may deteriorate), although wall–clock time can still benefit from parallelism.

The LMO-based method changes the *location* of $b^\star$ (through $H_{\text{LMO}}$) and therefore can shift the "sweet spot" of batch sizes compared to SGD, especially in regimes where the ABC constants are more favorable under the LMO-based geometry than under the Euclidean one.

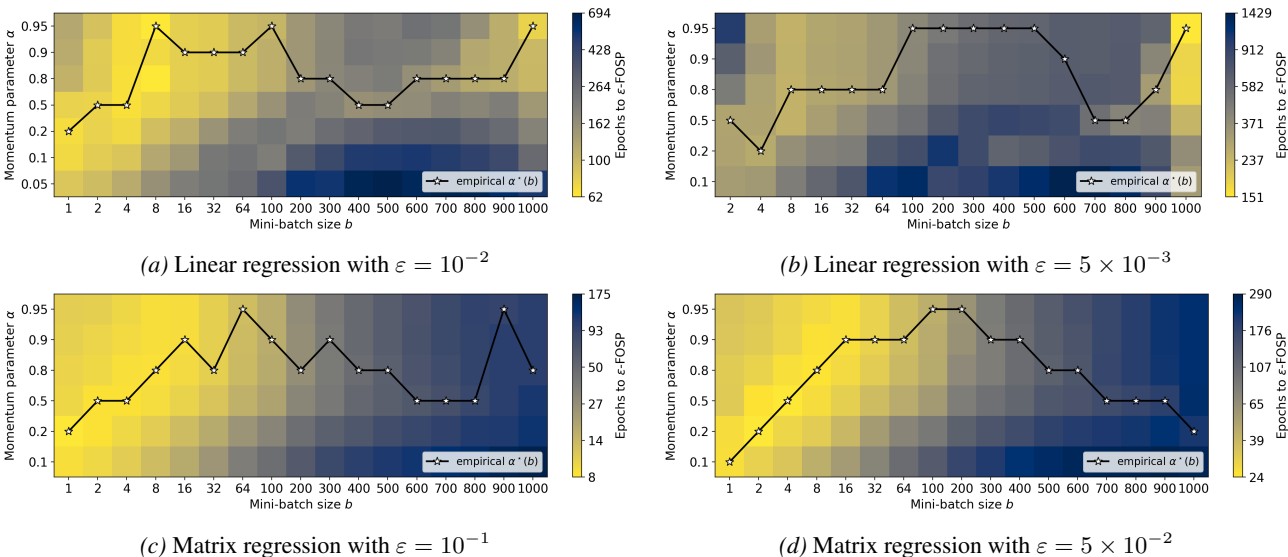

*(a)* Linear regression with $\varepsilon = 10^{-2}$

*(b)* Linear regression with $\varepsilon = 5 \times 10^{-3}$

*(c)* Matrix regression with $\varepsilon = 10^{-1}$

*(d)* Matrix regression with $\varepsilon = 5 \times 10^{-2}$

*Figure 5.* Heat maps for the optimal number of epochs achieved for each $(\alpha, b)$ pair after tuning $\gamma$

## C. Experiments

### C.1. Regression

The experiment focus on a standard matrix regression model,

$$Y = XW + \xi, \qquad \xi \sim \mathcal{N}\left(0, \sigma^2 I_n\right),$$

where $X \in \mathbb{R}^{n \times d_1}$, $Y \in \mathbb{R}^{n \times d_2}$, and $W \in \mathbb{R}^{d_1 \times d_2}$. The special case $d_2 = 1$ reduces to ordinary linear regression. In our synthetic setup, the design matrix $X$ and ground-truth parameter $W_{\text{true}}$ are sampled i.i.d. from a standard multivariate normal distribution. We use $n = 1000$, $d_1 = 50$, $d_2 = 5$, and $\sigma = 0.05$.

We consider the least-squares objective

$$f(W) \;=\; \|XW - Y\|_F^2,$$

and aim to compute an $\varepsilon$-first-order stationary point $W^\star$

$$\|\nabla f(W^\star)\|_F \leq \varepsilon.$$

Equivalently, $W^\star$ approximately minimizes $f$ up to the prescribed first-order optimality tolerance. Throughout, we use $\tau$-nice sampling for mini-batch selection to match the assumptions in our theoretical analysis.

For every $(\alpha, b)$ pair, we tune the stepsize $\gamma$ via a grid search. Specifically, we evaluate $\gamma \in \{10^{-5}, 2 \times 10^{-5}, \dots\}$ with grid spacing $10^{-5}$ with exception for results that have high standard deviation we used finer spacing $10^{-6}$ around the suspicious points, then select the $\gamma$ that minimizes the number of iterations needed to satisfy the stopping criterion. We consider the stepsize to be optimal once the subsequent consecutive grid points fail to reach the required stationary point with the current

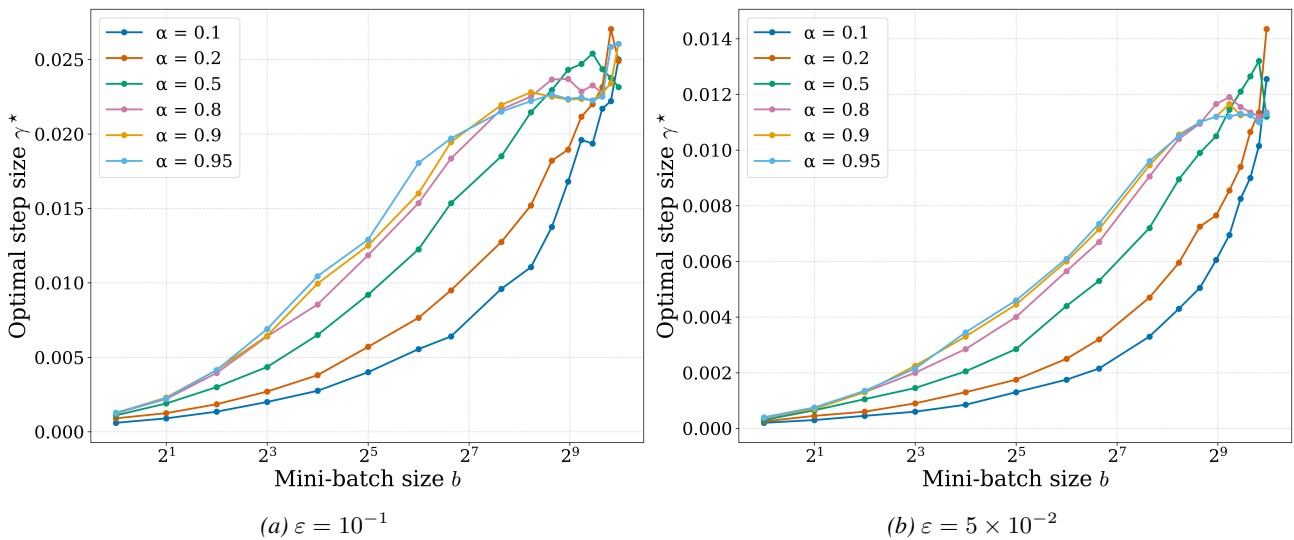

*(a)* $\varepsilon = 10^{-1}$            *(b)* $\varepsilon = 5 \times 10^{-2}$

*Figure 6.* Optimal step size for matrix regression via Muon.

number of iterations. For each hyperparameter configuration, we repeat the run three times with independent randomness and report the mean and standard deviation for number of iterations. For fair comparison, the number of iterations is converted into the total gradient evaluation done as the number of epochs.

For completeness, we also report the step sizes tuned to minimize the number of epochs required for Muon to reach an $\varepsilon$-first-order stationary point in matrix regression, as shown in Fig. 6. As in NSGD, the optimal step size increases with the batch size $b$. Moreover, for larger momentum values $\alpha$, the optimal step size $\gamma^\star$ is also larger for fixed $b$. The only notable difference from the behavior of NSGD is that $\gamma^\star$ remains nearly constant for a large batch size near full batch size and high $\alpha \in 0.8, 0.9, 0.95$.

Another helpful visualization is the heatmaps 5 to see the interaction between $\alpha$ and $b$ more clearly. In our experiments, linear regression is optimized using NSGD with momentum, whereas matrix regression is optimized using Muon. In the easier cases (Figures 5a and 5c), Muon and NSGD with momentum exhibit noticeably greater robustness with respect to the choice of the momentum parameter $\alpha$ compared to their counterparts in hard cases: 5b and 5d, which is consistent with our expectations.

Moreover, the dependence of the optimal momentum on the batch size is visually apparent: $\alpha^\star$ increases with $b$. This trend is particularly pronounced as the problem gets harder, where the heatmap becomes more green along the main diagonal (increasing $b$ and $\alpha$ jointly), while it shifts toward red along the anti-diagonal (increasing $b$ while decreasing $\alpha$).

### C.2. Text Prediction

In our setup, we use the WikiText-103 corpus (Merity et al., 2017), which is first tokenized and then partitioned into input-target pairs using a sliding window of fixed length $L$. Specifically, for a sequence of tokens starting at position $s$, the input-target pair is defined as

$$x = (t_s, t_s + 1, \ldots, t_{x+L-1}) \qquad y = (t_{s+1}, t_{s+2}, \ldots, t_{x+L})$$

We set $L = 128$ and use a stride of 128, so consecutive text blocks do not overlap.

In our implementation, we configure Nano-GPT with 10 transformer layers, 3 attention heads, and an embedding dimension of 96, resulting in approximately 6M parameters in total. The model is trained using the token-level cross-entropy loss to find $p_\theta$ the predictive distribution parametrized by $\theta$

$$\mathcal{L}(\theta) = -\frac{1}{L} \sum_{i=1}^{L} \log p_\theta(t_i | t_{<i})$$

The Muon implementation used in this experiment differs slightly from that in our earlier setup. In particular, we use

the PyTorch implementation, which is parameterized by the momentum coefficient $\beta$. This parameter is related to the momentum parameter $\alpha$ in our standard notation through the relation $\beta = 1 - \alpha$. For consistency, we report the results with $\alpha$.

In particular, Muon cannot be applied to all layers: some components, such as the embedding layer, are not compatible with Muon and are therefore optimized using AdamW instead. We use a shared fixed learning rate for both AdamW and Muon, which is made possible by matching the root-mean-square magnitude of Muon's update to that of AdamW, following the Moonshot implementation (Liu et al., 2025). AdamW has default values for the momentum ($\beta_1 = 0.9$ and $\beta_2 = 0.95$) and the weight decay for both optimizers is zero. Unlike regression experiment, we use Muon with five steps of the Newton-Schulz iteration to approximately orthogonalize the update matrix, since exact orthogonalization is computationally prohibitive. Finally, we use standard epoch-based training rather than $\tau$-nice sampling, as our goal is to examine whether our theoretical insights carry over to a practical training setting.

