# OpenReview forum: "General Analysis of LMO-based Optimizers: Beyond Bounded Variance"
_ICML.cc/2026/Conference — ICML 2026 regular_

### Official Review · Reviewer_tZdi · 2026-03-05

**Soundness:** 4
**Presentation:** 3
**Significance:** 3
**Originality:** 3
**Overall Recommendation:** 4
**Confidence:** 4

**Summary:**

This paper studies the convergence and sample complexity of linear minimization oracle (LMO)-based optimizers under the ABC (expected smoothness) condition, which handles stochastic gradients with potentially unbounded variances. The effects of parameters such as batch sizes are carefully analyzed. Numerical experiments on matrix regression are conducted to verify their theoretical findings.

**Compliance With Llm Reviewing Policy:**

Affirmed.

**Final Justification:**

The authors addressed all my concerns in my rebuttal, and I thus kept my original postive assessment.

**Key Questions For Authors:**

1. What are the other practical examples where the ABC assumption is satisfied but not the bounded-variance assumption?

2. Are the self-bounding closure arguments required for the analysis of existing literature that studies the convergence of other algorithms under the ABC assumption, or is this difficulty uniquely related to LMO optimizers?

3. Do your convergence guarantees subsume the existing guarantees of SGD-type algorithms under the ABC assumption? If not, what are the key differences? It would also be good to have a table summarizing the results from the literature and the results in the current work.

**Limitations:**

A discussion on the limitations seems to be missing.

**Strengths And Weaknesses:**

Strengths:

1. This paper presents technically sound convergence guarantees of momentum-based LMO optimizers under the ABC assumption, which is more general than the bounded variance assumption. The results are timely due to the recent success of Muon and other LMO-based optimizers. Also, posing the ABC assumption allows analysis of more practical problems.

2. Novel proof techniques referred to as "self-bounding closure" are used to handle the analysis for LMO optimizers.

3. The analysis on critical batch size and momentum scaling is detailed and provides practical insights.

Weaknesses:

1. The validity of the ABC assumption should be better justified, with examples beyond quadratics.

2. Larger-scale numerical experiments (with nonconvex objectives) should be conducted to better correspond to the theoretical results, which do not require convexity.

---

> ### Author Rebuttal · Authors · 2026-03-31
>
> Thank you for the very positive assessment and for the constructive questions.
>
> ---
>
> 1. **Why ABC is meaningful beyond quadratics.** The derivation in Section 3 is not restricted to quadratics. More generally, if $f(x)=\frac1n\sum_i f_i(x)$ with each $f_i$ being $L_i$-smooth and bounded below, then the standard self-bounding inequality gives
>
> $$
> \|\nabla f_i(x)\|_2^2 \le 2L_i\big(f_i(x)-f_i^{\inf}\big).
> $$
>
> Under uniform/$\tau$-nice subsampling this yields an ABC bound of the same form as in our paper; this is precisely the expected-smoothness perspective developed in [1-3]. Thus the class includes many smooth finite-sum models beyond quadratics, e.g. least-squares, regularized generalized linear models, and more generally smooth bounded-below finite-sum objectives. Related expected-smoothness/ABC bounds also appear in compressed and distributed SGD analyses [2,3]. In contrast, uniformly bounded variance can still fail on unbounded domains for heterogeneous smooth components; the quadratic construction in Section 3 is the cleanest illustrative example of this broader phenomenon.
>
> ---
>
> 2. **Is the self-bounding closure specific to LMO methods?** The difficulty is not ABC alone; it is ABC combined with the momentum/LMO tracking recursion. In standard SGD-under-ABC analyses [2], the noise enters the descent recursion directly and the proof closes without this extra step. Here the geometrically weighted momentum-lag term depends on the whole optimization history, which is why we need the closure $U\le D+E\sqrt U$. So this ingredient is specific to the momentum-LMO setting studied here, rather than a generic requirement of all ABC analyses.
>
> ---
>
> 3. **Relation to SGD-type results under ABC.** Our guarantees are **complementary** to, rather than subsuming, SGD-under-ABC results. Under the Euclidean norm and $\alpha=1$, our method reduces to **normalized** SGD (unit-length steps), not plain SGD. The key distinction is therefore algorithmic: SGD analyses such as [2] study magnitude-sensitive gradient steps under ABC, whereas we study norm-constrained LMO directions (including normalized/sign/Muon-style updates) under ABC. For convenience, the comparison is: **SGD under ABC** [2]: plain SGD, ABC, $\widetilde O(\varepsilon^{-4})$; **prior LMO/Muon results** [4-7]: normalized/sign/Muon/LMO under bounded variance or bounded moments; **this paper**: momentum LMO family under ABC, with explicit critical-batch and momentum-scaling laws. We will add a short comparison table to make this distinction immediate.
>
> ---
>
> 4. **Larger-scale nonconvex experiment / limitations.** We agree with both points. We have additionally run a larger-scale nonconvex next-token-prediction experiment (nanoGPT-style, 6M parameters, WikiText), using the practical Newton--Schulz Muon approximation (https://osf.io/5u7bx/files/x6jvh?view_only=8ccbbc87d936466eb678b3f72e660631). It shows the same qualitative critical-batch behavior, with a clear optimum around $b\approx 128$ across momentum values. We will add this experiment in the final version. We will also add a limitations paragraph making explicit that the current theory assumes smooth objectives, target/fixed-horizon tuning, and a Euclidean ABC model, and that the current camera-ready should include broader empirical validation.
>
> ---
>
> **References**
>
> 1. R. M. Gower, N. Loizou, X. Qian, A. Sailanbayev, E. Shulgin, and P. Richtárik. *SGD: General analysis and improved rates*. ICML, 2019.
> 2. A. Khaled and P. Richtárik. *Better theory for SGD in the nonconvex world*. arXiv:2002.03329, 2020.
> 3. Z. Li and P. Richtárik. *A unified analysis of stochastic gradient methods for nonconvex federated optimization*. arXiv:2006.07013, 2020.
> 4. A. Cutkosky and H. Mehta. *Momentum improves normalized SGD*. ICML, 2020.
> 5. J. Li and M. Hong. *A note on the convergence of Muon*. arXiv:2502.02900, 2025.
> 6. T. Pethick, W. Xie, K. Antonakopoulos, Z. Zhu, A. Silveti-Falls, and V. Cevher. *Training deep learning models with norm-constrained LMOs*. ICML, 2025.
> 7. D. Kovalev. *Understanding gradient orthogonalization for deep learning via non-Euclidean trust-region optimization*. arXiv:2503.12645, 2025.

---

> > ### Author Rebuttal · Reviewer_tZdi · 2026-04-01
> >
> > I would like to thank the authors for their detailed response. I will keep my positive assessment of the paper.

---

### Official Review · Reviewer_tke6 · 2026-03-09

**Soundness:** 3
**Presentation:** 3
**Significance:** 3
**Originality:** 2
**Overall Recommendation:** 4
**Confidence:** 3

**Summary:**

This work studies Linear Minimization Oracle (LMO) methods under expected smoothness. In particular, the authors provide unified convergence guarantees for a family of LMO methods in non-convex problems under the expected smoothness assumption (ABC condition). Further, the authors investigate the relationship between batch size and sample complexity in the cases of bounded-variance and $\tau$-nice sampling. Specifically, in the case of $\tau$-nice sampling, they determine the critical batch size that minimizes the theoretical sample complexity by identifying three different regimes based on the tolerance $\epsilon$. Finally, the authors perform numerical experiments using NSGD and Muon in linear and matrix regression, respectively, where they confirm their theoretical results on the optimal batch-size.

**Compliance With Llm Reviewing Policy:**

Affirmed.

**Final Justification:**

The authors have addressed my questions and concerns in their rebuttal. Given the few revisions I have highlighted in my review, I will keep my recommendation as a weak accept.

**Key Questions For Authors:**

- In the experiments with matrix regression, have the authors tried experimenting with a smaller $\epsilon$? Would the "climax" pattern be more pronounced in that case? From the current plots it seems like for larger values of $\alpha$, the deterministic setting does not show any improvement in the number of epochs, i.e., the curvature-dominated regime is not very obvious.

**Limitations:**

Yes.

**Strengths And Weaknesses:**

**Strengths**:
- This work extends the convergence guarantees of LMO methods to the case of expected smoothness, which is more general than bounded variance.
- The theory provided also covers the case of bounded variance.
- The authors provide nice explanations of the theoretical results and the specific terms involved, which makes the reading of the paper more enjoyable.
- The theoretical results accommodate different choices of momentum initialization.
- The conducted numerical experiments nicely complement the theory and provide further insights into the interplay between batch size and momentum parameter.

**Weaknesses**:
- A limitation of providing convergence results in terms of the dual norm is the appearance of the norm-equivalency constants in the bounds. These can introduce an explicit dependency on the problem dimension, which may be problematic in high-dimensional settings.
- In the experimental results, it would be nice to include a plot of step size vs mini-batch size for matrix regression, similar to linear regression, for completeness.
- Minor points on the notation: I may have missed it, but $e^{-1}$ was not introduced before the statement of Theorem 5.1. Also, in step 3 of the proof sketch for the same theorem, $D$ and $E$ are a little unclear. Maybe the dependence of $D$ and $E$ on $K$ could be made more explicit.

---

> ### Author Rebuttal · Authors · 2026-03-31
>
> Thank you for the encouraging assessment and for highlighting both the theoretical and empirical strengths.
>
> ---
>
> 1. **Norm-equivalence constants / dimension dependence.** We agree this is an important limitation to state clearly. Two remarks are relevant. First, this issue is not specific to our paper; it is standard whenever the stochastic model is stated in Euclidean norm while the update geometry is non-Euclidean, including in prior LMO analyses [2,3]. Second, Appendix C shows that if one has a **geometry-matched non-Euclidean ABC condition**, then the same proof removes the norm-compatibility factor $\rho$ altogether while preserving the same asymptotic $\varepsilon$-rates. So the dimension dependence comes from the Euclidean/non-Euclidean mismatch in the noise model, not from the LMO analysis per se. We will emphasize this limitation and the Appendix C remedy more clearly.
>
> ---
>
> 2. **Notation around Theorem 5.1.** Thank you; we will fix this. In particular, we will define the relevant quantities before the theorem, and in the proof sketch we will replace the abstract $D,E$ notation by explicit expressions (or at least explicit $K,\alpha,\gamma$ dependence) so the closure step is easier to follow.
>
> ---
>
> 3. **Step-size-vs-batch-size plot for matrix regression.** We agree this would improve completeness. We have generated the corresponding Muon plot (https://osf.io/5u7bx/files/hce2j?view_only=8ccbbc87d936466eb678b3f72e660631), and it shows the same qualitative trend as in linear regression: the empirically optimal step size increases with batch size. We will add this figure in the revision.
>
> ---
>
> 4. **Smaller $\varepsilon$ / "climax" pattern.** Yes -- with a smaller target tolerance, the non-monotone pattern becomes more pronounced, and the curvature-dominated regime is easier to see (especially for smaller momentum values such as $\alpha=0.1$). We have run this additional sweep (https://osf.io/5u7bx/files/463mg?view_only=8ccbbc87d936466eb678b3f72e660631) and will add it to the revised version.
>
> ---
>
> **References**
>
> 1. A. Khaled and P. Richtárik. *Better theory for SGD in the nonconvex world*. arXiv:2002.03329, 2020.
> 2. T. Pethick, W. Xie, K. Antonakopoulos, Z. Zhu, A. Silveti-Falls, and V. Cevher. *Training deep learning models with norm-constrained LMOs*. ICML, 2025.
> 3. D. Kovalev. *Understanding gradient orthogonalization for deep learning via non-Euclidean trust-region optimization*. arXiv:2503.12645, 2025.

---

> > ### Author Rebuttal · Reviewer_tke6 · 2026-04-03
> >
> > I thank the authors for their response and the additional experimental results. I would suggest that the authors add the experimental results with the smaller values of $\epsilon$ for the matrix regression case and revise the corresponding text to explain the shown pattern. I will maintain my recommendation.

---

### Official Review · Reviewer_9msF · 2026-03-16

**Soundness:** 3
**Presentation:** 3
**Significance:** 3
**Originality:** 2
**Overall Recommendation:** 4
**Confidence:** 3

**Summary:**

This paper studies the convergence of momentum-based Linear Minimization Oracle (LMO) methods, including normalized SGD, SignSGD, and Muon. The main motivation is that the bounded-variance assumption used in prior works can fail even for simple finite-sum problems. To address this, the paper analyzes these methods under the ABC assumption, which controls the second moment of the stochastic gradient using function suboptimality, gradient norm, and a residual term. Based on this framework, the paper derives a unified nonconvex convergence result. In particular, under $\tau$-nice sampling, the paper identifies a critical batch size that minimizes sample complexity, and it also derives how the optimal momentum and step size scale with the batch size. Finally, the paper supports the theory with experiments on linear regression and matrix regression.

**Compliance With Llm Reviewing Policy:**

Affirmed.

**Key Questions For Authors:**

Q1. Many recent optimization papers generalize results from classical smoothness to $(L_0,L_1)$-smoothness setting. Would it be possible to extend the analysis in this paper to the $(L_0,L_1)$-smoothness setting as well?

Q2. In convergence analyses for LMO-based optimizers, is it common to use $K$-dependent choices of $\alpha$ and $\gamma$? I initially thought this could be a weakness, since the result requires knowing $K$ in advance.

**Limitations:**

yes

**Strengths And Weaknesses:**

## Strength

- The proof sketch is easy to follow, and the full proof is clearly written. I only checked Theorem 5.1 in some detail, but it looked correct to me.
- The experiments are well-conducted. While many optimization papers describe experiments only roughly, this paper describes the experimental setup in detail, and the empirical findings are well aligned with the theory.

## Weakness

I think this paper is polished overall, but my main concern is that the contribution does not feel very large.

- The paper has a single main technical result, namely Theorem 5.1. As I understand it, under the ABC assumption, the main new difficulty in the proof comes from the extra term involving $A_{eff}$. The paper handles this by deriving an inequality of the form $U\le D+E\sqrt{U}$ and then solving this quadratic inequality to upper bound $U$. This argument seems interesting, but it does not look like a broadly reusable proof technique. For this reason, the technical novelty of the proof itself seems limited, and the contribution of the paper seems to be primarily an extension of existing results to the ABC-assumption setting.

Below are some minor weaknesses.

- In the statement of Theorem 5.1, the quantity $e^{-1}$ appears before it is defined.
- In Section 5.2, the authors state the optimal parameters under $\tau$-nice sampling in equation 17. While the min in $\alpha^\star$ prevents blowup as $b$ goes to $n$, $\gamma^\star$ seems to go to infinity as written. This part requires clarification.
- In the matrix regression experiment, the stopping criterion is based on the Frobenius norm of the gradient. Since Muon is defined through the spectral norm geometry, I would have expected the nuclear norm to be the more natural choice here, as it is the dual norm of the spectral norm.

---

> ### Author Rebuttal · Authors · 2026-03-31
>
> Thank you for the positive assessment of the proof and experiments. We are glad the main theorem and empirical section came across clearly.
>
> Regarding the concern that the contribution may feel limited, we respectfully view the paper as more than a routine extension. Existing LMO/Muon analyses are largely under bounded variance or bounded moments [2-4,6], which can fail under finite-sum subsampling and also suppress the batch-size effects that motivate the work. Under ABC [1], the main obstruction is not just an extra term in the algebra: the momentum-lag recursion produces a geometrically weighted noise term that is no longer bounded by a constant, but instead scales like
> $\sqrt{A_{\mathrm{eff}}\alpha}\,\Big(\tfrac1K\sum_i \Delta_i\Big)^{1/2},$
>
> which couples the proof to the entire optimization history. The self-bounding closure is what closes this recursion. This yields, to our knowledge, the **first** subsampling-faithful nonconvex guarantees for the momentum LMO family under ABC, together with explicit momentum/batch-size scaling laws and the critical-batch prediction. We therefore see the contribution as both technically new and practically motivated.
>
>
> On the minor points:
>
> 1. **Quantity defined after first use.** Thank you; we will fix this in the revision.
>
>
> 2. **Equation (17) and the $b\to n$ regime.** This is a very helpful comment. The scaling in (17) is stated for the stochastic $\tau$-nice regime $b\in\{1,\dots,n-1\}$ and should not be extrapolated to the full-batch endpoint. Because $b$ is discrete, the largest stochastic batch is $n-1$, so the displayed factor is finite; at $b=n$, the oracle becomes deterministic and $A_b=C_b=0$, so the analysis transitions to the deterministic/full-gradient regime rather than using the stochastic scaling formula. We will make this transition explicit right after (17) to avoid the impression of a blow-up.
>
>
> 3. **Frobenius vs nuclear norm in the Muon experiment.** We used the Frobenius norm only as a practical stopping rule, not as part of the Muon update. It is much cheaper to evaluate than the nuclear norm, while $\|G\|_\star\ge \|G\|_F$, so replacing $\|\cdot\|_F$ by $\|\cdot\|_{\star}$ at the same tolerance would only make the criterion stricter and typically increase runtime. In our tests, the qualitative batch-size trend was unchanged, so we kept the cheaper criterion.
>
>
> 4. **$(L_0,L_1)$-smoothness.** We think this is a promising and largely orthogonal direction. The closure argument itself is driven by the ABC-induced history coupling; if the drift term can be controlled under $(L_0,L_1)$-smoothness, the same proof strategy should plausibly extend. We will mention this as future work; this is also closely related to the broader non-Euclidean smoothness viewpoint in [5].
>
>
> 5. **$K$-dependent $\alpha,\gamma$.** Horizon-tuned choices are standard when one wants a clean finite-horizon nonconvex rate, including in prior normalized/LMO analyses [2-4,6]. Our Theorem 5.1 follows this route: for any chosen $\alpha_0,\gamma_0>0$, the schedule $\alpha=\alpha_0/\sqrt K$, $\gamma=\gamma_0/K^{3/4}$ yields the stated $\widetilde O(K^{-1/4})$ envelope. We agree that this should be explained more clearly, and we will also mention standard multistage/doubling-trick conversions when an anytime implementation is desired.
>
>
> **References**
>
> 1. A. Khaled and P. Richtárik. *Better theory for SGD in the nonconvex world*. arXiv:2002.03329, 2020.
> 2. A. Cutkosky and H. Mehta. *Momentum improves normalized SGD*. ICML, 2020.
> 3. T. Sun, Q. Wang, D. Li, and B. Wang. *Momentum ensures convergence of signsgd under weaker assumptions*. ICML, 2023.
> 4. T. Pethick, W. Xie, K. Antonakopoulos, Z. Zhu, A. Silveti-Falls, and V. Cevher. *Training deep learning models with norm-constrained LMOs*. ICML, 2025.
> 5. T. Pethick, W. Xie, M. Erdogan, K. Antonakopoulos, T. Silveti-Falls, and V. Cevher. *Generalized gradient norm clipping and non-Euclidean $(L_0,L_1)$-smoothness*. arXiv:2506.01913, 2025.
> 6. D. Kovalev. *Understanding gradient orthogonalization for deep learning via non-Euclidean trust-region optimization*. arXiv:2503.12645, 2025.

---

> > ### Author Rebuttal · Reviewer_9msF · 2026-04-03
> >
> > I thank the authors for their response. I will maintain my original recommendation.

---

### Official Review · Reviewer_cFzF · 2026-03-18

**Soundness:** 2
**Presentation:** 2
**Significance:** 2
**Originality:** 2
**Overall Recommendation:** 3
**Confidence:** 4

**Summary:**

This paper studies stochastic steepest descent / LMO-based methods with momentum for unconstrained non-convex minimization problems with unbounded variance gradient oracles. Specifically, the authors consider general non-convex minimization problems of the form
$$\min_{x\in\mathbb{R}^d} f(x)$$
for some $L$-smooth function $f \colon \mathbb{R}^d \to \mathbb{R}$ with $\inf f > -\infty$. The authors assume access to a stochastic first-order oracle providing stochastic gradient information on $f$, that is, a random vector $g(x)$ such that
$$\mathbb{E}[g(x)] = \nabla f(x)$$
and moments satisfying the so-called ABC condition
$$\mathbb{E}[\|g(x)\|_2^2] \leq A[f(x) - \inf f] + B \|\nabla f(x)\|_2^2 + C \tag{ABC}$$
instead of the standard bounded variance assumption $\mathrm{Var}[g(x)] \leq \sigma^2$ (which corresponds to $B=1$, $C=\sigma^2$ in ABC).

The authors consider the family of momentum-based stochastic steepest descent methods
\begin{align}
m_{t+1} &= (1-\alpha) m_t + \alpha g(x_t)
\\\\
x_{t+1} &= x_t + \gamma_t \mathrm{LMO}(m_{t+1})
\end{align}
where $\alpha\in(0,1)$ is a momentum parameter and $\mathrm{LMO}(v) = \arg\min_{\|z\| \leq 1} \langle v,z\rangle$ for some underlying norm $\|\cdot\|$ (the $L^2$ norm for normalized updates, and the $L^\infty$ norm for sign updates).

The authors' main results is that, if the algorithm above is run for a number of iterations $T$ (fixed in advance) with parameters $\alpha\propto1/\sqrt{T}$ and $\gamma\propto1/T^{3/4}$ chosen as a function of $T$, the algorithm's iterates enjoy a guarantee of the form
$$\\mathbb{E}[(1/T) \\sum_{t=1}^T \\|\nabla f(x_t)\\|_\ast] = \mathcal{\\tilde O}(1/T^{1/4})$$

The authors actually provide a more detailed expression for this bound (including subleading terms and bounds for the various constants involved. The authors also provide some ancillary results for finite-sum / minibatch oracles, and a range of numerical experiments validating their theoretical analysis.

**Compliance With Llm Reviewing Policy:**

Affirmed.

**Final Justification:**

As I stated in my original review, I find the technical contribution solid for ICML, and while I did not check the proofs in detail, I didn't spot any red flags in terms of correctness.

The technical part of my discussion with the authors focused on the practicality of the paper's guarantees (some of which are not immediately implementable as stated in the paper). While I was not fully satisfied by the authors' responses, again, from a technical standpoint, I believe that the paper clears the bar.

The presentation is another issue. After Section 5.1 or so, the quality of the presentation drops quite sharply, and I am not comfortable making an "accept" recommendation without a second look at a revised version. For this reason, I maintain my "weak reject" recommendation: I believe this can be a nice paper, but it's not yet there.

**Key Questions For Authors:**

See above.

**Limitations:**

See above.

**Strengths And Weaknesses:**

I enjoyed the premise of the paper, and the main result and assumptions were, for the most part, clearly stated.

On the other hand, the quality of the presentation dropped significantly after Section 5.1, to the extent that I was not sure what I was reading when I got to Section 6 (where Theorem 6.1 was presented without any context or explanations of its relation to the previous section, etc). With a lot of back-and-forth between the main text and the appendix, one can understand what's going on, but, if I'm honest, the write-up and presentation after page 5 is—at best—unpolished.

As for the authors' results per se, I have the following comments:
1. The paper's results do not cover the case $\alpha=1$ which corresponds to the standard signed / normalized SGD and Muon methods. Going into the paper—that is, from the title and the abstract—I expected to see new results about these algorithms, but the paper is exclusively about momentum-based methods, something which the authors did not clarify.
2. It is not exactly clear how to use the authors' convergence rate result.
    - First, in order to be meaningful, an algorithm should provide a specific output point, which the authors' analysis does not. The standard "random stopping" trick of Ghadimi-Lan seems to apply in the case of Theorem 5.1, in which case this issue would be painlessly resolved; however, the authors do not discuss this issue at all, so I was left wondering if there is an issue.
    - Second, in order to be practical, any stopping conditions / tuning requirements should not involve quantities that are only possible to estimate if one has solved the minimization problem under study in the first place. Specifically, in order to apply Theorems 5.1/6.1, the algorithm must be run with $T$ tuned as a function of $\epsilon$, and $\alpha,\gamma$ tuned as a function of $T$. The problem here is that the tuning of $T$ with respect to $\epsilon$ involves the starting gap $\Delta_0 = f(x_0) - \inf f$ which, in general, cannot be estimated. [The situation is not as bad as if the bound depended on the initial distance to a minimizer, but it is still unclear how to use this bound in practice. There are also other constants that would need to be known in advance, but I am less concerned about those.]

The second issue is not unresolvable, and I can think of use cases where it can be completely circumvented. However, the complete lack of discussion of this issue—as well as the lack of anytime guarantees with variable $\alpha$ and $\gamma$ depending on $t$ instead of $T$—significantly limits the applicability of the authors' results and analysis.

To conclude, even though I liked the paper's premise, I believe it requires a major revision along the lines indicated above—that is, fixing the presentation and providing implementable guarantees—before being considered for publication at a top-tier venue. My "weak rejection" rating should be interpreted in this light: the current manuscript does not clear the (admittedly high) bar for acceptance, but a drastically revised version might; however, this is not something that can be reasonably undertaken in a week-long discussion and it would most likely require a fresh set of reviews, so I am compelled to make a "reject" recommendation at this stage.

---

> ### Author Rebuttal · Authors · 2026-03-31
>
> Thank you for the careful reading and for identifying the main issues as scope, practical use of the guarantees, and presentation. We agree that the manuscript should make these points much more explicit.
>
> 1. **Scope / no-memory case.** The intended object is the **momentum LMO family**. We agree that the current title/abstract could more clearly foreground this. That said, the theory does include the no-memory boundary: in Theorem 5.1, taking $\alpha_0=\sqrt{K}$ gives $\alpha=1$, hence $m_{k+1}=g_k$, i.e., the standard memoryless LMO step; likewise Theorem 6.1 allows $\alpha=1$ through the $\min\lbrace 1,\cdot\rbrace$ choice. So the boundary case is covered, but the **main technical focus** is momentum, because the new difficulty is precisely the momentum-induced history coupling under ABC. This focus is also consistent with prior analyses of normalized/sign/LMO methods with momentum [3-6]. We will revise the title/abstract/Section 4 so this scope is unmistakable.
>
> 2. **Meaningful output point.** You are right that this should have been stated. Theorem 5.1 bounds
> $\frac1K\sum_{k=0}^{K-1}\mathbb E\|\nabla f(x_{k+1})\|_\star.$
>
> Therefore the standard Ghadimi-Lan random-output trick [1] applies immediately: if $R\sim \mathrm{Unif}\\{0,\dots,K-1\\}$, then
>
> $\mathbb{E}|\nabla f(x_{R+1})|\_{\star}
> = \frac{1}{K} \sum\_{k=0}^{K-1 }\mathbb{E} \|\nabla f(x_{k+1})\|\_{\star}.$
>
> There is no hidden obstruction here; we simply omitted this standard corollary. We will add it explicitly.
>
>
> 3. **Practical tuning / role of $\Delta_0$.** There are two distinct layers of results. Theorem 5.1 / Cor. A.2 are **fixed-horizon guarantees**: for any chosen $\alpha_0,\gamma_0>0$, running $\alpha=\alpha_0/\sqrt K$, $\gamma=\gamma_0/K^{3/4}$ yields the stated $\widetilde O(K^{-1/4})$ envelope, so the method can be run without knowing $\Delta_0$. The $\varepsilon$-targeted statements then convert this master bound into sufficient parameter choices and budgets, which is standard in nonconvex stochastic optimization [1,2]. We agree that the role of $\Delta_0$ should have been explained more clearly: only an **upper bound** is needed there, and when a lower bound on the objective is known one may use $\Delta_0\le f(x_0)-f_{\inf}$. We will add this discussion and also state more clearly that our current theory is fixed-horizon / target-tuned rather than anytime; developing anytime schedules is an important next step.
>
> ---
>
> 4. **Presentation after Section 5.1.** We agree. Section 6 is a specialization of the master bound: Section 6.1 recovers the bounded-variance regime, while Section 6.2 plugs in the $\tau$-nice ABC constants to obtain the batch-size tradeoff, critical-batch result, and parameter scalings. We will rewrite the transition so this dependency is explicit, and define the key quantities before they are used in the proof sketch.
>
> ---
>
> 5. **Why this is not a routine extension of SGD-under-ABC.** The proof does not follow by directly importing SGD-under-ABC analyses. Under bounded variance, the geometrically weighted noise term in the momentum-lag recursion is controlled by a constant and the proof closes by summing a geometric series, as in prior normalized/LMO analyses [3-6]. Under ABC, that same term becomes coupled to the full optimization history through $\frac1K\sum_i(f(x_i)-f_{\inf})$, exactly as in the ABC framework of [2]. The self-bounding closure is what converts this history-coupled term into an explicit rate. This is the main technical novelty of the paper.
>
> ---
>
> **References**
>
> 1. S. Ghadimi and G. Lan. *Stochastic first- and zeroth-order methods for nonconvex stochastic programming*. SIAM Journal on Optimization, 2013.
> 2. A. Khaled and P. Richtárik. *Better theory for SGD in the nonconvex world*. arXiv:2002.03329, 2020.
> 3. A. Cutkosky and H. Mehta. *Momentum improves normalized SGD*. ICML, 2020.
> 4. T. Sun, Q. Wang, D. Li, and B. Wang. *Momentum ensures convergence of signsgd under weaker assumptions*. ICML, 2023.
> 5. T. Pethick, W. Xie, K. Antonakopoulos, Z. Zhu, A. Silveti-Falls, and V. Cevher. *Training deep learning models with norm-constrained LMOs*. ICML, 2025.
> 6. D. Kovalev. *Understanding gradient orthogonalization for deep learning via non-Euclidean trust-region optimization*. arXiv:2503.12645, 2025.

---

> > ### Author Rebuttal · Reviewer_cFzF · 2026-04-01
> >
> > Dear authors,
> >
> > Thank you for your replies. I am happy to see that there is no obstruction to the random stopping trick, but I remain unconvinced about the rest of your replies.
> > - Insofar as the presentation is concerned, a major revision is required, and this is not something that can be evaluated without a fresh look at the paper. [And the discussion phase is intended to iron out mostly minor issues; major issues requiring a drastic rewrite are beyond the scope of the rebuttal mechanism]
> > - I am also unconvinced by your reply regarding the algorithm's anytime guarantees. When running an optimization algorithm, one either wants a guarantee of the form "we get ε accuracy if the algorithm is run for this many iterations" or, in an agnostic setting, "if I let the algorithm run over time, the output keeps getting better and better". If you fix the algorithm's hyperparameters as a function of the number of iterations $K$, then you only get a guarantee for a single iteration, and this satisfies neither of the above criteria: it cannot be tuned to ε (if $Δ_0$ is not known), and it does not get better over time (since it only applies to a very specific output point). In order for such a guarantee to be truly anytime, you would need to run the method with variable parameters $\propto 1/k$ not $\propto 1/K$.
> >
> > I do not believe that either of the above issues would be a roadblock for the ultimate acceptance of your paper after a major revision. In a journal, I would likely make a "revise and resubmit" recommendation; however, without the possibility of further review, I have to remain on the "reject" side of the things.
> >
> > All my very best,
> >
> > Reviewer cFzF
> >
> >
> > ---
> >
> > Edit: I wanted to add a follow-up message, but the ICML setup this year makes it impossible. I therefore edited my message here, and I include my follow-up below:
> >
> > > As a follow-up to my rebuttal acknowledgment, I should point out that a finite-horizon guarantee can be turned into an anytime one via a doubling trick. In this regard, my second point above is not a show-stopper; however, the presentation in the paper—as well as your rebuttal—is not very clear about these issues, hence my concern.
> >
> > ---
> >
> > Edit2: I also changed my acknowledgment from (c) to (b), to make sure that the authors have the opportunity to respond (I do not know what the system allows at this stage). [However, my assessment remains at (c), as I believe that Section 6 (and part of Section 5) requires a major rewrite]

---

> > > ### Author Response · Authors · 2026-04-03
> > >
> > > Thank you for the follow-up. We appreciate your clarification that (i) the random-stopping point is not an obstruction, and (ii) a fixed-horizon guarantee can be converted into an anytime one via a standard doubling trick. In light of this, we understand your remaining concern as being primarily about exposition/signposting rather than soundness.
> > >
> > > We would, however, like to push back carefully on the claim that the manuscript requires a *major rewrite*. At present, this concern is still difficult for us to act on concretely. In your original review, the premise, assumptions, and main result were described as mostly clearly stated, and the only specific presentation example identified was the transition after Section 5.1, especially the role of Theorem 6.1 relative to the preceding material. We agree that this bridge can be made clearer. But beyond that, we would be very grateful for one or two **specific** places in Sections 5-6 (a definition, transition, or theorem statement) that you found most problematic. As things stand, the critique remains too vague for us to treat it as evidence of a paper-wide presentation failure.
> > >
> > > We also want to stress that the *type* of guarantee used in our paper is standard. The submission already contains both:
> > > 1. a **fixed-horizon master bound** (Theorem 5.1 / Cor. A.2), and
> > > 2. **$\varepsilon$-targeted/sample-complexity statements** (Section 6).
> > >
> > > This fixed-horizon $\rightarrow$ complexity-corollary structure is standard in nonconvex stochastic optimization, including prior SGD under ABC (Khaled and Richtarik, 2020) and normalized/LMO analyses (Kovalev, 2025). Likewise, horizon-tuned constant parameters are standard in this literature (Cutkosky and H. Mehta, 2021; Sun et. al, 2023; Khaled and Richtarik, 2020; Kovalev, 2025) rather than something peculiar to our paper. This is also why we do not view the use of a finite-horizon master theorem as a scientific weakness by itself.
> > >
> > > Relatedly, now that you explicitly note that the doubling trick resolves the anytime issue in principle, we believe the remaining question is *how clearly this is explained*, not whether the paper lacks an implementable guarantee. Section 6 already provides the first type of guarantee you mention, namely, an explicit $\varepsilon$-targeted/sample-complexity statement, while the second type (anytime) follows from the same fixed-horizon result via a standard multistage/doubling conversion.
> > >
> > > At the same time, we now also have a clear route to a **genuine one-run variable-parameter extension**, and we mention this not as a new claim of the submitted paper, but to clarify that there is no conceptual roadblock here. The natural schedule is
> > > $\alpha_k \asymp (k+1)^{-1/2}$ and $\gamma_k \asymp (k+1)^{-3/4}$,
> > > with the output index sampled proportionally to $\gamma_k$. Appendix A.3 already writes the descent inequality with variable $\gamma_k$; the remaining modification is in the momentum-lag analysis. There, instead of the current unweighted history average used in the self-bounding closure, one uses the kernel-weighted history quantity induced by the variable momentum coefficients. This is precisely in the spirit of Hubler et. al (2024) Parameter-Agnostic Optimization under Relaxed Smoothness, which studies NSGD-M with the same $(k^{-1/2},k^{-3/4})$ schedule. The resulting rate naturally incurs logarithmic losses, which is also consistent with the broader recent anytime literature, e.g. Kornowski and Shamir (2025). So while the submitted paper focuses on the fixed-horizon theory, we do not view the variable-parameter anytime version as a major conceptual gap.
> > >
> > > Finally, we note that the other reviewers assessed the presentation as **good** and raised only localized notation/signposting issues rather than a need for a major rewrite. This is why we currently view the remaining issue as a camera-ready-level expository refinement, not a fundamental obstacle to the paper's scientific contribution.
> > >
> > > If you are willing to point us to one or two concrete places in Sections 5-6 that most affected your reading, we would sincerely appreciate it and will target the revision accordingly.
> > >
> > > **References**
> > > 1. A. Khaled and P. Richtarik. *Better Theory for SGD in the Nonconvex World*. Transactions on Machine Learning Research, 2023.
> > > 2. A. Cutkosky and H. Mehta. *Momentum Improves Normalized SGD*. ICML, 2020.
> > > 3. T. Sun, Q. Wang, D. Li, and B. Wang. *Momentum Ensures Convergence of SignSGD under Weaker Assumptions*. ICML, 2023.
> > > 4. D. Kovalev. *Understanding Gradient Orthogonalization for Deep Learning via Non-Euclidean Trust-Region Optimization*. arXiv:2503.12645, 2025.
> > > 5. F. Hubler, J. Yang, X. Li, and N. He. *Parameter-Agnostic Optimization under Relaxed Smoothness*. AISTATS, 2024.
> > > 6. G. Kornowski and O. Shamir. *Gradient Descent's Last Iterate is Often (slightly) Suboptimal*. NeurIPS 2025 OPT-ML Workshop.

---

### Decision · Program_Chairs · 2026-04-30

**Decision:**

Accept (regular)

**Comment:**

**Summary:** This paper studies the convergence of momentum-based Linear Minimization Oracle (LMO) methods, including normalized SGD, SignSGD, and Muon. The main motivation is that the bounded-variance assumption used in prior works can fail even for simple finite-sum problems. To address this, the paper analyzes these methods under the ABC assumption and derives a unified non-convex convergence result. In particular, under $\tau$-nice sampling, the paper identifies a critical batch size that minimizes sample complexity, and also derives how the optimal momentum and step size scale with the batch size. Finally, the paper supports the theory with experiments on linear regression and matrix regression.

**Strengths:**
- This work extends the convergence guarantees of LMO methods to the case of expected smoothness, which is more practical than bounded variance, and introduces some new proof techniques.
- Well-written paper with a clear theoretical result and supporting experiments.

**Weaknesses:**
- Lack of justification for the ABC assumption
- Minor theoretical discrepancies in the fixed-horizon guarantees.

**Decision and Suggested Changes:** The reviewers unanimously agree that the paper's contributions merit acceptance. Addressing the following concerns will help strengthen the current version of the paper:
- Clearly stating the scope of the paper to be momentum LMO methods (Rev. cFzF)
- Clarifying the main technical novelty, and expanding the discussion about the practical tuning (Rev. cFzF, Rev. 9msF)
- Improve the presentation, especially in Section 5 and beyond (Rev. cFzF)
- Adding the comment about the dimension dependence and the geometry-matched non-Euclidean ABC condition (Rev. tke6)
- Additional experiments and analysis plots (Rev. tke6, Rev. Rev. tZdi)